# Robust Mean Estimation Without Moments for Symmetric Distributions

**Gleb Novikov**
Department of Computer Science
ETH Zurich

**David Steurer**
Department of Computer Science
ETH Zurich

**Stefan Tiegel**
Department of Computer Science
ETH Zurich

## Abstract

We study the problem of robustly estimating the mean or location parameter without moment assumptions. Known computationally efficient algorithms rely on strong distributional assumptions, such as sub-Gaussianity, or (certifiably) bounded moments. Moreover, the guarantees that they achieve in the heavy-tailed setting are weaker than those for sub-Gaussian distributions with known covariance. In this work, we show that such a tradeoff, between error guarantees and heavy-tails, is not necessary for symmetric distributions. We show that for a large class of symmetric distributions, the same error as in the Gaussian setting can be achieved efficiently. The distributions we study include products of arbitrary symmetric one-dimensional distributions, such as product Cauchy distributions, as well as elliptical distributions, a vast generalization of the Gaussian distribution.

For product distributions and elliptical distributions with known scatter (covariance) matrix, we show that given an $\varepsilon$-corrupted sample, we can with probability at least $1 - \delta$ estimate its location up to error $O(\varepsilon\sqrt{\log(1/\varepsilon)})$ using $\frac{d\log(d)+\log(1/\delta)}{\varepsilon^2\log(1/\varepsilon)}$ samples. This result matches the best-known guarantees for the Gaussian distribution and known SQ lower bounds (up to the $\log(d)$ factor). For elliptical distributions with unknown scatter (covariance) matrix, we propose a sequence of efficient algorithms that approaches this optimal error. Specifically, for every $k \in \mathbb{N}$, we design an estimator using time and samples $\tilde{O}(d^k)$ achieving error $O(\varepsilon^{1-\frac{1}{2k}})$. This matches the error and running time guarantees when assuming certifiably bounded moments of order up to $k$. For unknown covariance, such error bounds of $o(\sqrt{\varepsilon})$ are not even known for (general) sub-Gaussian distributions.

Our algorithms are based on a generalization of the well-known filtering technique [DK22]. More specifically, we show how this machinery can be combined with Huber-loss-based techniques to work with projections of the noise that behave more nicely than the initial noise. Moreover, we show how sum-of-squares proofs can be used to obtain algorithmic guarantees even for distributions without a first moment. We believe that this approach may find other applications in future works.

## 1 Introduction

Robust statistics [Hub11, RHRS11] is a central field in statistics with the goal of designing algorithms for statistical problems that are robust to a small amount of outliers, for example caused by measurement errors or corrupted data. We model this as follows: Samples are generated from an unknown

37th Conference on Neural Information Processing Systems (NeurIPS 2023).

distribution $\mathcal{D}$ and then an $\varepsilon$-fraction of them is arbitrarily corrupted by an adversary with full knowledge of the underlying model and our algorithm. We only have access to the corrupted samples. In this work we focus on the canonical task of estimating the mean of $\mathcal{D}$. Traditionally, estimators robust to such corruptions have been computationally inefficient, requiring time exponential in the ambient dimension [Ber06], while computationally efficient estimators have incurred error scaling with the ambient dimension thus rendering them unsuitable for today's high-dimensional statistical tasks. Recently however, a flurry of efficient estimators emerged achieving guarantees without this prohibitive dependence on the dimension and with error rates approaching those of computationally inefficient ones [DKK+19, LRV16, CSV17, DKK+17, HL18, KSS18, DK22].

A textbook example of this development is when $\mathcal{D}$ is the multi-variate Gaussian distribution (or sub-Gaussian distributions with *known* covariance). In this setting, efficient algorithms can estimate the mean to within error $O(\varepsilon\sqrt{\log(1/\varepsilon)})$, i.e., nearly linear in $\varepsilon$ [DKK+19, DKK+17, KMZ22]. The statiscally optimal error rate is $O(\varepsilon)$ [CGR18]. That is, efficient algorithms are optimal up to the additional factor of $\sqrt{\log(1/\varepsilon)}$. Further, it is conjectured that this factor is inherent for efficient algorithms, i.e., that there is a *computational-statistical gap*. This is evidenced by known lower bounds for statistical query algorithms [DKS17]. However, there are two drawbacks with this textbook example: If the covariance matrix is unknown, it is not known how to efficiently achieve the same error for sub-Gaussian distributions, in fact, in full generality, it is not even known how to achieve error $o(\sqrt{\varepsilon})$ in this setting. This is because known algorithms rely heavily on the algebraic structure of Gaussian moments. Second, the assumption that the uncorrupted data belongs to a (sub)-Gaussian distribution is arguably very strong. A more natural setting is when $\mathcal{D}$ is allowed to have heavier tails. In the setting that $\mathcal{D}$ has bounded second moments it is known how to achieve error $O(\sqrt{\varepsilon})$ efficiently [DKS19, LRV16, CSV17, DKK+17]. Interestingly, $O(\sqrt{\varepsilon})$ seems to constitute a barrier for efficient algorithms. In particular, while error $o(\sqrt{\varepsilon})$ is possible information-theoretically if we assume that higher-order moments are bounded, [HL19] show that bounded moments alone are likely not enough for efficient algorithms, and that additional assumptions are required. Currently, these manifest as either assuming that there is a certificate, in sum-of-squares, for the boundedness of the moments [HL18, KSS18] or assuming the covariance is the identity (or known) [DKP20]. While these approaches indeed break the $O(\sqrt{\varepsilon})$ barrier, they in many cases fall short of the $O(\varepsilon\sqrt{\log(1/\varepsilon)})$ error possible in the Gaussian setting. In particular, they only achieve error comparable to the Gaussian case, when using $O(\log(1/\varepsilon))$ many moments. Unfortunately, it is necessary to use this many moments, even when assuming they are certifiably bounded [KSS18]. Thus, the tails of these distributions are already much lighter than, say, in the bounded covariance case. Therefore, a natural question is:

*Do there exist classes of heavy-tailed distributions for which we can
(efficiently) achieve the same robust error as for the Gaussian
distribution?*

In this work, we answer this question by giving algorithms that use a (quasi-)polynomial number of samples (in the dimension and $1/\varepsilon$) and time polynomial in the number of samples for the broad class of symmetric product and elliptical distributions. Moreover, in many cases, we can achieve this using (nearly) the same amount of samples as for the Gaussian distribution, recovering, e.g., the optimal dependence on the failure probability. We remark that the distributions we consider might have arbitrarily heavy tails and we do not make any assumptions related to sum-of-squares.

**Product and Elliptical Distributions** Symmetry is a natural assumption in statistics with many applications to, e.g., mathematical finance and risk management [Lin75, OR83, Cha83]. In this work, we consider the following two types of symmetric distributions that are of particular interest [Kel70, CHS81, Fan18]: First, product distributions of symmetric one-dimensional distributions and spherically symmetric distributions. These correspond to a generalization of the standard Gaussian distribution. Examples are product Cauchy distributions and the multi-variate Student $t$-distribution with identity scatter matrix. Second, elliptical distributions. These correspond to a generalization of Gaussians with arbitrary (unknown) covariance matrix. Examples include the multi-variate Student $t$-distribution, symmetric multivariate stable distributions and multivariate Laplace distributions. For both classes, it is information-theoretically possible to obtain robust error $O(\varepsilon)$ [CGR18, PBR20], i.e., matching that of a Gaussian. These approaches are not known to run in faster than exponential time and prior work [PBR20] asked whether a similar error can be achieved efficiently. We respond to this question, by designing algorithms that nearly, in some cases exactly, match this error in

polynomial (in the number of samples) time, if the number of samples is polynomial (in the cases of symmetric product or spherically symmetric distribution) or quasi-polynomial (in the case of elliptical distributions) in the dimension and $1/\varepsilon$.

## 1.1 Problem Set-Ups and Results

Next, we give formal definitions of the corruption model and distributions considered and state our results. We use the following standard model for corruptions – often referred to as the strong contaminiation model.

**Definition 1.1.** Let $\boldsymbol{X}_1, \ldots, \boldsymbol{X}_n$ be i.i.d.samples from a distribution $\mathcal{D}$ and let $\varepsilon > 0$. We say that $Z_1, \ldots, Z_n$ are an $\varepsilon$-*corruption* of $\boldsymbol{X}_1, \ldots, \boldsymbol{X}_n$, if they agree on at least an $(1 - \varepsilon)$-fraction of the points. The remaining $\varepsilon$-fraction can be arbitrary and in particular, can be corrupted by an adversary with full knowledge of the model, our algorithm, and all problem parameters.

We consider the following two classes of distributions.

**Definition 1.2** (Semi-Product Distributions)**.** Let $\rho > 0$. We say a distribution $\mathcal{D}$ over $\mathbb{R}^d$ is an $\rho$-*semi-product* distribution, if for $\boldsymbol{\eta} \sim \mathcal{D}$ it holds that

1. For all $j \in [d]$, the distribution of $\boldsymbol{\eta}_j$ is symmetric about 0,

2. For all $j \in [d]$, $\mathbb{P}\left(|\boldsymbol{\eta}_j| \leqslant \rho\right) \geqslant \frac{1}{100}$,

3. The random vectors $(\mathrm{sign}(\boldsymbol{\eta}_j))_{j=1}^d$ and $(|\boldsymbol{\eta}_j|)_{j=1}^d$ are independent, and the random variables $\mathrm{sign}\left(\boldsymbol{\eta}_1\right), \ldots, \mathrm{sign}\left(\boldsymbol{\eta}_d\right)$ are mutually independent.

Some remarks about the definition are in order. First, notice that the Gaussian distribution $N(0, \rho^2 \cdot I)$ is $\Theta(\rho)$-semi-product. Similarly, every symmetric product or spherically symmetric distribution that has covariance bounded by $\rho^2 \cdot \mathrm{Id}_d$ is $\Theta(\rho)$-semi-product. In particular, the coordinates $\boldsymbol{\eta}_j$ need not be independent. However, the definition allows for much heavier tails, the only requirement is that at least a $1/100$-fraction of the probability mass lies in an interval of length $\rho$ around $0$[1]. In particular, it captures all spherically symmetric distributions, e.g. multi-variate $t$-distributions with identity scatter matrix, and all symmetric product distributions, e.g. the product Cauchy distribution. Notice that in the non-robust setting the first two properties are enough to accurately estimate $\mu^*$ from samples $\mu^* + \boldsymbol{\eta}_1, \ldots, \mu^* + \boldsymbol{\eta}_n$, via adding symmetric mean zero noise with tiny variance to each coordinate and computing the entry-wise median. We expect that *some* additional assumption is necessary for efficient algorithms in the robust setting. We show that Property 3 above is sufficient. A different sufficient condition would be assuming that the distribution is elliptical, which we discuss next.

The second one is the class of elliptical distributions [Kel70, CHS81, Fan18] a generalization of spherically symmetric distribution, which in particular can have more complex dependency structures.

**Definition 1.3.** A distribution $\mathcal{D}$ over $\mathbb{R}^d$ is *elliptical*, if for $\boldsymbol{\eta} \sim \mathcal{D}$ the following holds: Let $\boldsymbol{U}$ be uniformly distributed over the $d$-dimensional unit sphere. There exists a positive random variable $\boldsymbol{R}$, independent of $\boldsymbol{U}$, and a positive semi-definite matrix $\Sigma$, such that

$$\boldsymbol{\eta} = \boldsymbol{R}\Sigma^{1/2}\boldsymbol{U} \quad \text{and} \quad \mathbb{P}\left(\boldsymbol{R} \leqslant \sqrt{2d}\right) \geqslant \frac{1}{100} \, .$$

We call $\Sigma$ the *scatter* matrix of the distribution (sometimes also referred to as the dispersion matrix). In particular, the Gaussian distribution $N(0, \Sigma)$ with arbitrary $\Sigma$ is elliptical. Further, we can choose its covariance matrix as the scatter matrix. Indeed, we can decompose $\boldsymbol{X} \sim N(0, \Sigma)$ as $\boldsymbol{X} = \boldsymbol{R}\Sigma^{1/2}\boldsymbol{U}$, where $\boldsymbol{R}$ is distributed as the square root of a $\chi^2$-distribution with $d$ degrees of freedom and $\boldsymbol{U}$ is independent of $\boldsymbol{R}$ and distributed uniformly over the unit sphere. Then by Markov's Inequality, it holds that $\mathbb{P}(\boldsymbol{X} \leqslant \sqrt{2d}) = \mathbb{P}(\boldsymbol{X}^2 \leqslant 2d) \geqslant \frac{1}{2}$. Elliptical distributions with scatter matrix identity correspond to spherically symmetric distributions, a special case of semi-product distributions. Note that that elliptical distributions *do not* capture product distributions except for the Gaussian case – e.g., the product Cauchy distribution is not elliptical.

---

[1]This can even further be relaxed to any $\alpha > 0$ fraction. The sample complexity and error then scale naturally with $\alpha$. See the appendices for more details.

Extending the above two defintions, we say that a distribution $\mathcal{D}$ is $\rho$-semi-product (resp. elliptical) *with location* $\mu^* \in \mathbb{R}^d$ if samples of $\mathcal{D}$ take the form $\mu^* + \boldsymbol{\eta}$, where $\mu^* \in \mathbb{R}^d$ is deterministic and $\boldsymbol{\eta}$ is $\rho$-semi-product (resp. elliptical). Note that the location takes the place of the mean, as this might not exist for distributions of the above form.

**Results**   Our main result for semi-product distributions is the following: Note that we can reduce the case of elliptical distributions with known scatter matrix to the $\Theta(1)$-semi-product case, hence the theorems below also apply to this setting. We also remark that the algorithm only receives the corrupted samples as input. In particular, $\rho$ need not be known (and can be estimated from the corrupted samples). See the appendices for a proof.

**Theorem 1.4.** *Let* $\mu^* \in \mathbb{R}^d, \varepsilon, \rho > 0$ *and* $\mathcal{D}$ *be a* $\rho$-*semi-product distribution with location* $\mu^*$. *Let* $C > 0$ *be a large enough absolute constant and assume that* $\varepsilon \leqslant 1/C$ *and* $n \geqslant C \cdot \frac{d \log(d) + \log(1/\delta)}{\varepsilon^2 \log(1/\varepsilon)}$. *Then, there exists an algorithm that, given an* $\varepsilon$-*corrupted sample from* $\mathcal{D}$, *runs in time* $n^{O(1)}$ *and outputs* $\hat{\mu} \in \mathbb{R}^d$ *such that with probability at least* $1 - \delta$ *it holds that*

$$\|\hat{\mu} - \mu^*\| \leqslant O\left(\rho \cdot \left[\sqrt{\frac{d \log(d) + \log(1/\delta)}{n}} + \varepsilon\sqrt{\log(1/\varepsilon)}\right]\right) .$$

Our sample complexity is nearly optimal in the dependence of the error on $\varepsilon$ and $d$ and optimal in the dependence on the failure probability $\delta$, all up to constant factors. Recall that $N(0, \rho^2 \cdot I_d)$ is a $\Theta(\rho)$-semi-product distribution and hence lower bounds for this setting apply. The statistically optimal error in this setting is $\Omega(\rho \cdot \varepsilon)$ [CGR18] and can be achieved using $n \geqslant \frac{d + \log(1/\delta)}{\varepsilon^2}$ samples. Note that we match this error up to the $\sqrt{\log(1/\varepsilon)}$ term, using only slightly more samples ($d$ vs $d \log d$). It is conjectured that the larger error is necessary for efficient algorithms, as it is necessary for all efficient SQ algorithms [DKS17]. It is an interesting open question to remove the additional factor of $\log d$ in our sample complexity. Further, our algorithm nearly matches results known for the standard Gaussian distribution (up to the $\log(d)$ factor in error and sample complexity) [DKK+19, DKK+17, KMZ22]. We expect our algorithm to be practical, since it only uses one-dimensional smooth convex optimization ($O(nd)$ times), the top eigenvector computation ($O(n)$ times) and arithmetic operations.

Interestingly, we show how to achieve error scaling only with $O(\rho \cdot \varepsilon)$ using quasi-polynomially many samples:

**Theorem 1.5.** *Let* $\mu^* \in \mathbb{R}^d, \varepsilon, \rho > 0$ *and* $\mathcal{D}$ *be a* $\rho$-*semi-product distribution with location* $\mu^*$. *Let* $C > 0$ *be a large enough absolute constant and assume that* $\varepsilon \leqslant 1/C$ *and* $n \geqslant C \cdot d^{C \log(1/\varepsilon)}$. *Then, there exists an algorithm that, given an* $\varepsilon$-*corrupted sample from* $\mathcal{D}$, *runs in time* $n^{O(1)}$ *and outputs* $\hat{\mu} \in \mathbb{R}^d$ *such that with probability at least* $1 - \delta$ *it holds that*

$$\|\hat{\mu} - \mu^*\| \leqslant O\left(\rho \cdot \left[\sqrt{\frac{d + \log(1/\delta)}{n}} + \varepsilon\right]\right) .$$

In order to state our results for elliptical distributions, we need to introduce the notion of the *effective rank* of a matrix $\Sigma$. This is defined as $r(\Sigma) := \mathrm{Tr}\,\Sigma/\|\Sigma\|$ and captures the intrinsic dimensionality of the data. Note that it is always at most $d$, but can also be significantly smaller. We remark that we do not assume that the scatter matrix $\Sigma$ is known to the algorithm.

**Theorem 1.6.** *Let* $C > 0$ *be a large enough absolute constant. Let* $k \in \mathbb{N}, \varepsilon, \delta > 0, \mu^* \in \mathbb{R}^d$ *such that* $\varepsilon \leqslant 1/C$ *and assume* $\mathcal{D}$ *is an elliptical distribution with location* $\mu^*$ *and scatter matrix* $\Sigma$ *satisfying* $r(\Sigma) \geqslant C \cdot k \log d^2$. *Also, let* $n \geqslant C \cdot (r(\Sigma)/k)^k \log(d/\delta)$. *There is an algorithm that, given an* $\varepsilon$-*corrupted sample from* $\mathcal{D}$, *runs in time* $n^{O(1)} d^{O(k)}$ *and with probability at least* $1 - \delta$ *outputs* $\hat{\mu} \in \mathbb{R}^d$ *satisfying*

$$\|\hat{\mu} - \mu^*\| \leqslant O\left(\sqrt{\|\Sigma\|} \cdot \left[\sqrt{\frac{r(\Sigma) \cdot \log(d/\delta)}{n}} + \sqrt{k} \cdot \varepsilon^{1 - 1/(2k)}\right]\right) .$$

---

[2]A slightly weaker condition suffices, see the appendices for all details. Note that as long as $k \leqslant O(\log(1/\varepsilon)) \leqslant O(\log d)$, this assumption is mild and in particular is always satisfied if the condition number of $\Sigma$ is $O(1)$ (as $d \to \infty$).

Note that in the special case when $\mathcal{D} = N(0, \Sigma)$, the information-theoretically optimal error is $\Omega(\sqrt{\|\Sigma\|} \cdot \varepsilon)$ and can be achieved using $n \geqslant \frac{r(\Sigma) + \log(1/\delta)}{\varepsilon^2}$ samples [CGR18, MZ23]. We give a sequence of algorithms (nearly) approaching this error using an increasing number of samples and time. In fact, for $k = O(\log(1/\varepsilon))$, we achieve error $O(\varepsilon\sqrt{\log(1/\varepsilon)} \cdot \sqrt{\|\Sigma\|})$ using at most $\frac{r(\Sigma)^{\log(1/\varepsilon)} \log(d/\delta)}{\varepsilon^2}$ samples and time $d^k$. Similar to the bounded moment setting [KSS18, HL18] the parameter $k$ can be thought of as a way to trade between sample/time complexity and accuracy guarantees. Also note that already for $k = 2$ we achieve error $O(\varepsilon^{3/4}\sqrt{\|\Sigma\|})$ for elliptical distributions with unknown covariance/scatter matrix in polynomial time and sample complexity, while even for (general) sub-Gaussian distributions with unknown covariance it is not known if it is possible to achieve error $o(\sqrt{\varepsilon} \cdot \sqrt{\|\Sigma\|})$ in polynomial time and sample complexity. In addition, if $r(\Sigma)$ is much smaller than $d$, for example, $r(\Sigma) \leqslant d^{0.1}$, then we achieve error $O(\varepsilon^{3/4}\sqrt{\|\Sigma\|})$ with a *sub-linear* number of samples $n = \tilde{O}(d^{0.2})$ in time $d^{O(1)}$.

**Previous Algorithms and Possible Extensions**    Previously, to the best of our knowledge, only exponential time computable estimators were known, for both semi-product and elliptical distributions. The results [CGR18, PBR20] imply that there exist estimators that achieve error $O(\sqrt{\frac{d + \log(1/\delta)}{n}} + \varepsilon)$ (assuming $\|\Sigma\| \leqslant O(1)$ for elliptical distributions). Both algorithms rely on brute force techniques for which it is unclear how to make them efficient.

A natural question is if we can achieve (nearly) optimal error in polynomial time (analogous to the semi-product case). Indeed, for *non-spherical* Gaussian distributions, it is known how to do this [DKK+19, DKK+17]. These algorithms crucially rely on robustly estimating the covariance matrix first. Similarly for elliptical distributions, it seems necessary to first robustly estimate the scatter matrix and then apply our results for the semi-product case. However, current covariance estimation algorithms rely heavily on the algebraic relations between the Gaussian moments, and it is not known how to achieve this for other distributions where moments do not have these specific relations. Thus, we expect this to be an interesting but challenging task.

A different direction we believe is interesting to explore is the following: We gave algorithmic results matching in many cases what is known for the Gaussian distribution. In the non-robust setting, there is in principle hope to go beyond this: There are known estimators that are asymptotically normal with variance scaling with the Fisher information of the distribution [Sto75]. So there is hope for "better-than-Gaussian error" on distributions with small Fisher information. However, these results are purely asymptotic and there are (symmetric) distributions for which we cannot hope to achieve even finite sample results scaling with the Fisher information (see, e.g., the discussions in [GLP23]). To get around this issue, the recent work of [GLP23] introduced a related notion, called "smoothed Fisher Information" and achieved finite-sample (non-robust) error guarantees scaling with the smoothed Fisher Information in the one-dimensional case, also for symmetric distributions. It would be interesting to see, if similar guarantees can be achieved in the robust and high-dimensional setting, too.

We further believe that similar guarantees can be achieved based on $\ell_1$-minimization techniques (instead of Huber-loss) under slightly different assumptions. In particular, even in the non-robust setting, we need to require a small amount of density at (or sufficiently close to) the location, else any point in the region around the location with zero density, would be an $\ell_1$-minimizer. In the case of semi-product distributions, this is without of loss of generality, since we could add Gaussian noise of the appropriate variance to each coordinate. For elliptical distribution, this approach does not work, since the resulting distribution might no longer be elliptical.

## 1.2   Related Work

We only list the works most closely related to this work, we refer to [DK22] for a survey on the area. Most of the literature on efficient algorithms for robust mean estimation has focussed on the setting when at least some of the moments are bounded. In what follows we focus mainly on how the error guarantees depend on $\varepsilon$ and put other parameters, such as the dependence on the failure probability aside. [DKK+19, DKK+17, KMZ22] show how to efficiently estimate the mean of a Gaussian distribution, or sub-Gaussian with identity-covariance, up to error $O(\varepsilon\sqrt{\log(1/\varepsilon)})$. Assuming that the distribution has covariance bounded by identity, [CSV17, DKK+17, DKP20] show how to achieve

error $O(\sqrt{\varepsilon})$. Similarly, for $\alpha \in [0, 1]$, [CTBJ22] shows how to achieve error $O(\varepsilon^{\alpha/(1+\alpha)})$ when in every direction the $1 + \alpha$ moments are bounded. Note that in contrast to this, our algorithm can handle distributions without a first moment and achieves error $O(\varepsilon\sqrt{\log(1/\varepsilon)})$. An interesting question is whether assuming bounded higher-order moments can lead to improved error guarantees. Interestingly, [HL19] shows that this alone is likely to be not enough in order to go beyond error $\sqrt{\varepsilon}$. So far, there have been two ways of adding additional assumptions: First, assuming that $\mathcal{D}$ has $k$-th moments bounded by $\sigma_k$ in every direction, and assuming that it is certifiable in sum-of-squares, it is possible to achieve error $O(\sigma_k \varepsilon^{1-1/(2k)})$ using $d^{O(k)}$ samples and time [HL18, KSS18, ST21]. Second, if the $k$-th moments are bounded by $\sigma_k$, not necessarily certifiably in sum-of-squares, and the covariance matrix is a multiple of the identity (or known), [DKP20] achieves error $O(\sigma_k \varepsilon^{1-1/(2k)})$.

**Other Results On Symmetric Noise Distributions and the Filtering Techniques**  Huber-loss minimization has been used to design computationally efficient estimators for a variety of problems: Linear regression in the heavy-tailed/symmetric setting [TJSO14, PJL20, SZF20, dNS21, dLN$^+$21], as well as the robust setting [PJL20]. Other works developed algorithms for PCA under symmetric noise [CLMW11, dLN$^+$21, dNNS23].

The filtering technique [DKK$^+$19, DKK$^+$17] (or versions thereof) have been successfully applied to robust and heavy-tailed (when the covariance exists) mean estimation [DHL19, DKP20, HLZ20, DK22, DKLP22] as well as a pre-processing step in robust regression [PJL20].

## 2   Techniques

We will first describe the general theme of our techniques. These ideas will yield algorithms achieving (nearly) optimal guarantees using quasi-polynomially many samples. Note however, that using these, we can already achieve error $o(\sqrt{\varepsilon})$ using only polynomially many samples. At the end of this section we show how to improve this to nearly linearly many (in the dimension) samples for semi-product distributions.

**One-Dimensional Robust Estimation via Huber-Loss Minimization**  Current algorithms for robust mean estimation are only known to work for distributions with bounded moments. In many cases this is assumption is reasonable, and in many cases it is also necessary [KSS18, CTBJ22]. However, there are many other distributions that do not necessarily have any moments at all, but which one would expect to behave nicely. Current approaches are inapplicable in these settings. Of particular interest is the class of symmetric distributions, which might not even have a first moment - in this case we want to estimate the location parameter of the distribution, which always exists and coincides with the mean in case it exists. In non-robust statistics (when there are no corruptions), symmetric distributions in some sense behave as nicely as the Gaussian distribution: The minimizer of the entry-wise *Huber-loss* function achieves the same guarantees for such distributions as the sample mean in the Gaussian case. The entry-wise Huber-loss is the function $\mathcal{L}_H : \mathbb{R}^d \to \mathbb{R}$ defined as $\mathcal{L}_H(v) := \sum_{j=1}^d \Phi(v_j)$, where $\Phi \colon \mathbb{R} \to \mathbb{R}$ is a *Huber penalty*: $\Phi(x) = x^2/2$, if $|x| \leqslant 1$, and $\Phi(x) = |x| - 1/2$ otherwise. $\Phi$ is convex, and has many appealing properties, in particular with respect to symmetric distributions, since its derivative is a bounded odd Lipschitz function: $\phi(x) := \Phi'(x) = x$, if $|x| \leqslant 1$, and $\phi(x) = \mathrm{sign}(x)$ otherwise.

Since the Gaussian distribution is symmetric itself, the Huber-loss estimator has the same guarantees as the sample mean in this setting. Beyond that, the Huber-loss estimator enjoys some robustness guarantees that the sample mean does not: Let us consider the one-dimensional case, where the Huber-loss behaves similarly to the median but has additional properties that will be useful to us, e.g., it is differentiable everywhere. In this setting, the sample mean has unbounded error even if there is only a single corrupted sample, while the Huber-loss achieves the information theoretically optimal error $O(\varepsilon)$ when an arbitrary $\varepsilon$-fraction of the samples is corrupted. Moreover, these guarantees extend to arbitrary one-dimensional symmetric distributions that satisfy mild scaling assumptions. Unfortunately, this does not directly yield a good robust estimator for the high-dimensional setting. If we naively apply the one-dimensional estimator entry-wise, we can only guarantee an error bound of $O(\varepsilon\sqrt{d})$, which is far away from the $O(\varepsilon)$ error that is statistically possible for the Gaussian distribution and symmetric distributions. Moreover, it is also inferior to error rates obtained by efficient estimators that use assumptions about bounded moments: Such estimators achieve error

that does not depend on $d$. In this work we show that there exist more sophisticated estimators based on the Huber-loss[3] that achieve dimension-independent error, often matching what is possible for the Gaussian distribution, for estimating the location of symmetric distributions even in the high-dimensional robust setting.

**Proofs of identifiability and the filtering technique** Before describing how we use the Huber-loss in the high-dimensional setting, it will be instructive to recall how the classical approach for distributions with bounded moments works – for simplicity, we focus on bounded second moments and sketch how this extends to higher-order moments. Later, we will see how to modify these ideas, using the Huber-loss, to work also in the symmetric setting without moment assumptions. In particular the version for Gaussian-like error for semi-product distribution requires several new technical ideas compared to the Gaussian setting.

At the heart lie *identifiability proofs* which can be made algorithmic using either sum-of-squares [KSS18, HL18] or the filtering technique [DKK$^+$19, DKK$^+$17, DHL19]. For bounded covariance distributions these take the following form: If two distributions $D_1$ and $D_2$ with means $\mu_1$, respectively $\mu_2$, and covariance matrices $\Sigma_1 \preceq \mathrm{Id}_d$, respectively $\Sigma_2 \preceq \mathrm{Id}_d$, are $\varepsilon$-close in statistical distance, then $\|\mu_1 - \mu_2\| \leqslant O\left(\sqrt{\varepsilon}\right)$. In what follows, for simplicity of the discussion, we will sometimes switch between empirical and true distributions. In robust mean estimation, we assume that $D_1$ is the empirical distribution over the corrupted samples we observe, while $D_2$ is the empirical distribution of the uncorrupted samples. Hence, by assumption, $\Sigma_2 \preceq \mathrm{Id}_d$ and we wish to estimate $\mu_2$. The above statement of identifiability asserts that as long as $\Sigma_1 \preceq \mathrm{Id}_d$, the empirical mean of the corrupted samples, i.e., $\mu_1$, is $O(\sqrt{\varepsilon})$-close to $\mu_2$. Of course, $\Sigma_1 \preceq \mathrm{Id}_d$ might not hold, since the outliers could potentially introduce a large eigenvalue in the empirical covariance matrix. However, since we know that these large eigenvalues must have been introduced by outliers, this gives rise to a win-win analysis: We check if $\Sigma_1 \preceq \mathrm{Id}_d$ holds. If it does, $\mu_1$ is $O(\sqrt{\varepsilon})$-close to $\mu_2$ already, if not, we compute its top eigenvector and remove samples that have large correlation with this eigenvector. It turns out that this procedure will always remove more corrupted samples than uncorrupted ones. Thus, we can iterate this procedure and are guaranteed that it terminates after $O(\varepsilon n)$ iterations. After its termination, we are left with a distribution $D_1'$, with mean $\mu_1'$, that is $O(\varepsilon)$-close in total variation distance to $D_1$ and $D_2$, and has small covariance $\Sigma_1' \preceq \mathrm{Id}_d$, hence $\|\mu_1' - \mu_2\| \leqslant O\left(\sqrt{\varepsilon}\right)$.

Of course, for arbitrary symmetric distributions the covariance might not exist. So neither the above proof of identifiability nor the filtering technique apply to this setting. However, one of our main observations is that we can obtain a similar statement of identifiability also in this case. In addition, we will later show how to adjust the filtering technique to work with our identifiability proof - this requires non-trivial modifications of the original approach. First, observe that the identifiability statement in the bounded moment setting, can be phrased as follows: For $j \in \{1, 2\}$, the mean $\mu_j$ of a distribution $D_j$ corresponds to the minimizer of the quadratic loss:

$$\mu_j = \underset{a \in \mathbb{R}^d}{\arg \min} \; \underset{\boldsymbol{y} \sim D_j}{\mathbb{E}} \|\boldsymbol{y} - a\|^2 \;.$$

Hence, if $D_1$, with covariance $\Sigma_1 \preceq \mathrm{Id}_d$, and $D_2$ with covariance $\Sigma_2 \preceq \mathrm{Id}_d$, are $\varepsilon$-close in statistical distance, then the minimizers, $\mu_1$ and $\mu_2$, of the quadratic loss are $O(\sqrt{\varepsilon})$-close to each other. A crucial observation is that, if a distribution $D$ is symmetric, then its location $\mu$ is the entry-wise Huber-loss minimizer[4]

$$\mu = \underset{a \in \mathbb{R}^d}{\arg \min} \; \underset{\boldsymbol{y} \sim D}{\mathbb{E}} \mathcal{L}_H \left(\boldsymbol{y} - a\right) \;.$$

Hence, we would like to obtain a similar statement about minimizers of the Huber-loss for two $\varepsilon$-close distributions $D_1$ and $D_2$. This will be useful since we will need to learn the location $\mu_2$ of a symmetric distribution $D_2$ from $D_1$.

**Abstract proof of identifiability** In order to formalize the above, consider the following abstract setting: Let $\mathcal{L}$ be some loss function that is differentiable, $\Omega(1)$-strongly convex and smooth (i.e. its gradient $\nabla \mathcal{L}$ is 1-Lipschitz). Let $D_1$ and $D_2$ be two distributions that are $\varepsilon$-close in the statistical distance, Then, if for $j \in \{1, 2\}$ the minimizers $\hat{\mu}_j := \arg \min_{a \in \mathbb{R}^d} \mathbb{E}_{\boldsymbol{y} \sim D_j} \mathcal{L}\left(\boldsymbol{y} - a\right)$ satisfy

---

[3]We use entry-wise Huber-loss for semi-product distributions. For elliptical distributions we use another loss function, that is also related to the Huber penalty.

[4]Assuming that each entry of $\boldsymbol{y} - \mu$ has nonzero probability mass in the interval $[-1, 1]$.

$\mathbb{E}_{\boldsymbol{y} \sim D_j} \left\langle \nabla \mathcal{L} \left(\boldsymbol{y} - \hat{\mu}_j\right), u \right\rangle^2 \leqslant 1$ for all unit vectors $u$, then $\|\hat{\mu}_1 - \hat{\mu}_2\| \leqslant O(\sqrt{\varepsilon})$. Indeed, by strong convexity and the definition of $\hat{\mu}_1$,

$$\Omega\left(\|\hat{\mu}_1 - \hat{\mu}_2\|^2\right) \leqslant \mathbb{E}_{\boldsymbol{y} \sim D_1} \mathcal{L}\left(\boldsymbol{y} - \hat{\mu}_1\right) - \mathbb{E}_{\boldsymbol{y} \sim D_1} \mathcal{L}\left(\boldsymbol{y} - \hat{\mu}_2\right) - \mathbb{E}_{\boldsymbol{y} \sim D_1} \left\langle \nabla \mathcal{L}\left(\boldsymbol{y} - \hat{\mu}_2\right), \hat{\mu}_1 - \hat{\mu}_2 \right\rangle$$
$$\leqslant \mathbb{E}_{\boldsymbol{y} \sim D_1} \left\langle \nabla \mathcal{L}\left(\boldsymbol{y} - \hat{\mu}_2\right), \hat{\mu}_1 - \hat{\mu}_2 \right\rangle .$$

Notice that there is a coupling $\omega$ of $D_1$ and $D_2$ such that $\mathbb{P}_{(\boldsymbol{y}_1, \boldsymbol{y}_2) \sim \omega}(\boldsymbol{y}_1 \neq \boldsymbol{y}_2) \leqslant \varepsilon$. Since[5] $\mathbb{E}_{\boldsymbol{y} \sim D_2} \nabla \mathcal{L}\left(\boldsymbol{y} - \hat{\mu}_2\right) = 0$, we obtain using the Cauchy-Schwarz Inequality

$$\Omega\left(\|\hat{\mu}_1 - \hat{\mu}_2\|^2\right) \leqslant \mathbb{E}_{(\boldsymbol{y}_1, \boldsymbol{y}_2) \sim \omega} \left\langle \nabla \mathcal{L}\left(\boldsymbol{y}_1 - \hat{\mu}_2\right) - \nabla \mathcal{L}\left(\boldsymbol{y}_2 - \hat{\mu}_2\right), \hat{\mu}_1 - \hat{\mu}_2 \right\rangle$$
$$= \mathbb{E}_{(\boldsymbol{y}_1, \boldsymbol{y}_2) \sim \omega} \mathbf{1}_{[\boldsymbol{y}_1 \neq \boldsymbol{y}_2]} \left\langle \nabla \mathcal{L}\left(\boldsymbol{y}_1 - \hat{\mu}_2\right) - \nabla \mathcal{L}\left(\boldsymbol{y}_2 - \hat{\mu}_2\right), \hat{\mu}_1 - \hat{\mu}_2 \right\rangle$$
$$\leqslant \sqrt{\varepsilon \mathbb{E}_{(\boldsymbol{y}_1, \boldsymbol{y}_2) \sim \omega} \left\langle \nabla \mathcal{L}\left(\boldsymbol{y}_1 - \hat{\mu}_2\right) - \nabla \mathcal{L}\left(\boldsymbol{y}_2 - \hat{\mu}_2\right), \hat{\mu}_1 - \hat{\mu}_2 \right\rangle^2} .$$

Since $\nabla \mathcal{L}$ is 1-Lipschitz, using the bounds on $\max_{\|u\|=1} \mathbb{E}_{\boldsymbol{y} \sim D_j} \left\langle \nabla \mathcal{L}\left(\boldsymbol{y} - \hat{\mu}_j\right), u \right\rangle^2$ yields

$$\mathbb{E}_{(\boldsymbol{y}_1, \boldsymbol{y}_2) \sim \omega} \left\langle \nabla \mathcal{L}\left(\boldsymbol{y}_1 - \hat{\mu}_2\right) - \nabla \mathcal{L}\left(\boldsymbol{y}_2 - \hat{\mu}_2\right), \hat{\mu}_1 - \hat{\mu}_2 \right\rangle^2$$
$$= \mathbb{E}_{\boldsymbol{y}_j \sim D_j} \left\langle \nabla \mathcal{L}\left(\boldsymbol{y}_1 - \hat{\mu}_2\right) - \nabla \mathcal{L}\left(\boldsymbol{y}_1 - \hat{\mu}_1\right) + \nabla \mathcal{L}\left(\boldsymbol{y}_1 - \hat{\mu}_1\right) - \nabla \mathcal{L}\left(\boldsymbol{y}_2 - \hat{\mu}_2\right), \hat{\mu}_1 - \hat{\mu}_2 \right\rangle^2$$
$$\leqslant O\left(\mathbb{E}_{\boldsymbol{y}_j \sim D_j} \left\langle \nabla \mathcal{L}\left(\boldsymbol{y}_1 - \hat{\mu}_2\right) - \nabla \mathcal{L}\left(\boldsymbol{y}_1 - \hat{\mu}_1\right), \hat{\mu}_1 - \hat{\mu}_2 \right\rangle^2 + \|\hat{\mu}_1 - \hat{\mu}_2\|^2\right)$$
$$\leqslant O\left(\|\hat{\mu}_1 - \hat{\mu}_2\|^4 + \|\hat{\mu}_1 - \hat{\mu}_2\|^2\right) .$$

Hence, $\|\hat{\mu}_1 - \hat{\mu}_2\|^2 \leqslant O(\sqrt{\varepsilon} \cdot \|\hat{\mu}_1 - \hat{\mu}_2\|^2 + \sqrt{\varepsilon} \cdot \|\hat{\mu}_1 - \hat{\mu}_2\|)$. If $\varepsilon$ is small enough (smaller than some constant that depends on the strong convexity parameter), we obtain that $\|\hat{\mu}_1 - \hat{\mu}_2\| \leqslant O(\sqrt{\varepsilon})$.

Similar to the bounded moment setting, we can generalize this observation to the case when for some integer $k > 1$ and some $m_{2k} > 0$, $\max_{\|u\|=1} \mathbb{E}_{\boldsymbol{y}_j \sim D_j} \left\langle \nabla \mathcal{L}\left(\boldsymbol{y} - \hat{\mu}_j\right), u \right\rangle^{2k} \leqslant m_{2k}^{2k}$. In this case we can use Hölder's inequality to obtain the bound $\|\hat{\mu}_1 - \hat{\mu}_2\| \leqslant O(m_{2k} \cdot \varepsilon^{1 - 1/(2k)})$. Note that in order to combine higher-order moment bounds with the filtering technique, we need an efficient procedure to certify these bounds. Thus, previous works only applied to distributions for which such moment certificates, in sum-of-squares, exist. We remark that in the symmetric setting we are able to obtain the necessary certificates for filtering *without* making any sum-of-squares related assumption on the distribution.

**Filtering in the symmetric setting**   We will show how to adopt the filtering technique to make the above proof of identifiability algorithmic for symmetric distributions. Indeed, let $D_2$ be a symmetric distribution with location $\mu^*$. Assume we can choose a loss function $\mathcal{L}$ such that $\hat{\mu}_2 = \mu^*$ and $\mathbb{E}_{\boldsymbol{y} \sim D_2} \nabla \mathcal{L}\left(\boldsymbol{y} - \hat{\mu}_2\right) \nabla \mathcal{L}\left(\boldsymbol{y} - \hat{\mu}_2\right)^\top \preceq \mathrm{Id}_d$. Then, if $D_1$ is an $\varepsilon$-corruption of $D_2$, we can iteratively compute $\mathbb{E}_{\boldsymbol{y} \sim D_1} \nabla \mathcal{L}\left(\boldsymbol{y} - \hat{\mu}_1\right) \nabla \mathcal{L}\left(\boldsymbol{y} - \hat{\mu}_1\right)^\top$ and check its top eigenvalue. If it is smaller than 1, we know that $\hat{\mu}_1$ must be $O(\sqrt{\varepsilon})$-close to $\hat{\mu}_2 = \mu^*$, if not, we can remove points which are aligned with the top eigenvector. Similar to the classical filtering setting, we can argue that this removes more corrupted points than uncorrupted points. Thus, after at most $O(\varepsilon n)$ iterations, we must have that $\hat{\mu}_1$ is indeed $O(\sqrt{\varepsilon})$-close to $\mu^*$. A similar argument works for higher-order moments.

It remains to verify the assumptions we used on the loss function $\mathcal{L}$ and its interaction with our distribution. First, we required that $\mathcal{L}$ is smooth and strongly convex, which is very restrictive. Fortunately, we can relax the second assumption: For our argument it is enough if $\mathcal{L}$ is *locally* strongly convex around the minimizer (and globally smooth). Second, we assumed that $\mathbb{E}_{\boldsymbol{y} \sim D_2} \nabla \mathcal{L}\left(\boldsymbol{y} - \hat{\mu}_2\right) \nabla \mathcal{L}\left(\boldsymbol{y} - \hat{\mu}_j\right)^\top \preceq \mathrm{Id}_d$. $\nabla \mathcal{L}\left(\boldsymbol{y} - \hat{\mu}_2\right)$ can be seen as some transformation of the distribution, and our condition requires that the covariance of the transformed distribution is bounded. In general, this might not be satisfied even if the transformations are well-behaved (e.g., bounded

---

[5] In this discussion we assume that the operators $\mathbb{E}$ and $\nabla$ commute. In the analyses of the algorithms we use empirical means, and in this case they commute by linearity of the gradient.

and Lipschitz). However, we show that for appropriate $\mathcal{L}$, it is indeed satisfied for elliptical and semi-product distributions. For the sake of exposition, we focus here only on elliptical distributions. For such distributions, we use the loss function $\mathcal{L}_E(v) := r^2 \cdot \Phi(\|v\|/r)$ for some $r > 0$. It is not hard to see that it is globally smooth and locally strongly convex (in some neighborhood of the minimizer). Further, $\nabla \mathcal{L}_E(v)$ is a projection of $v$ onto the ball of radius $r$ (with center at zero). The covariance of such a projection is bounded by the covariance of the projection onto the sphere of radius $r$. Fortunately, spherical projections of elliptical distributions are well-behaved. In particular, they only depend on the scatter matrix $\Sigma$, and hence we can without loss of generality assume that the initial elliptical distribution was Gaussian. For which we can obtain a bound on the covariance of $\nabla \mathcal{L}_E(\boldsymbol{y} - \hat{\mu}_2)$.

Similarly, we show that under mild assumptions on the scatter matrix (that in particular are satisfied for scatter matrices with condition number $O(1)$), we can certify tight bounds on the moments of $\nabla \mathcal{L}_E(\boldsymbol{y} - \hat{\mu}_2)$ in sum-of-squares, which leads to polynomial-time algorithms that obtain error $o(\sqrt{\varepsilon})$. We remark that prior to this work, in the unknown covariance case, robust mean estimation with error $o(\sqrt{\varepsilon})$ was only possible under the assumption that higher-order moments of $D_1$ are certifiably bounded in sum-of-squares. We show that error $o(\sqrt{\varepsilon})$ is possible for arbitrary elliptical distributions that might not even have a first moment. Moreover, we show that by exploiting $O(\log(1/\varepsilon))$ many bounded moments of the transformed distributions, we achieve the near optimal error of $O(\varepsilon\sqrt{\log(1/\varepsilon)})$ using quasi-polynomially many samples and time polynomial in the input.

The above approach also works for semi-product distributions by choosing $\mathcal{L}$ to be the entry-wise Huber-loss. This works both for the setting when $\nabla \mathcal{L}(\boldsymbol{y} - \hat{\mu}_2)$ has bounded covariance and certifiably bounded higher-order moments. We remark that in this case we can achieve the information theoretically optimal error $O(\varepsilon)$ with quasi-polynomial number of samples in polynomial time (in the number of samples) – this again follows by exploiting $O(\log(1/\varepsilon))$ many moments.

### 2.1 Nearly optimal error for semi-product distributions using polynomially many samples

Note that even in the standard Gaussian case, using the standard filtering approach yields (nearly) optimal algorithms only when using quasi-polynomially many samples. Note that this seems somewhat inherent to the approach since it only uses low-degree moment information. To reduce this to polynomially many samples, we can use a stronger identifiability statement that exists for the standard Gaussian distribution. In particular, let $D_2 = N(\mu_2, \mathrm{Id}_d)$ and $D_1$ be an $\varepsilon$-corruption of $D_1$ with mean $\mu_1$ and covariance $\Sigma_1$. Then, it holds that [DKK+19, DKK+17] (see also [Li19])

$$\|\mu_1 - \mu_2\| \leqslant O\left(\varepsilon\sqrt{\log(1/\varepsilon)} + \sqrt{\varepsilon\left(\|\Sigma_1 - \mathrm{Id}_d\| + \varepsilon\log(1/\varepsilon)\right)}\right).$$

Thus, we can hope to achieve error $O(\varepsilon\sqrt{\log(1/\varepsilon)})$ by checking if $\|\Sigma_1 - \mathrm{Id}_d\| \leqslant O(\varepsilon\sqrt{\log(1/\varepsilon)})$ and iteratively removing points aligned with the top eigenvector of $\Sigma_1 - \mathrm{Id}_d$. Indeed, a procedure very similar to this works (see e.g. [Li19]). However, it is crucial that we only remove points among the top $\varepsilon$-fraction of points correlated with the top eigenvector, since otherwise we cannot ensure that we remove more corrupted than uncorrupted points. The proof of the above uses *stability conditions* of the Gaussian distribution (cf. [DK22]). That is, the proof uses that for i.i.d. samples $\boldsymbol{\eta}_1^*, \ldots, \boldsymbol{\eta}_n^* \sim N(\mu^*, \mathrm{Id}_d)$ with $n$ sufficiently large, it holds that for all subsets $T$ of size $(1 - 10\varepsilon)n$ we have

$$\left\| \frac{1}{|T|} \sum_{i \in T} \boldsymbol{\eta}^* \right\| \leqslant O\left(\varepsilon\sqrt{\log(1/\varepsilon)}\right) \qquad \text{and} \qquad \left\| \frac{1}{|T|} \sum_{i \in T} \boldsymbol{\eta}^* \boldsymbol{\eta}^{*\top} - \mathrm{Id}_d \right\| \leqslant O(\varepsilon\log(1/\varepsilon)).$$

We show that the above approach can be adapted to the semi-product setting. Indeed, let $D_2$ be a semi-product distribution with location $\mu^*$ and $\Sigma_2^{\mathcal{L}} = \mathbb{E}_{\boldsymbol{y} \sim D_2} \nabla \mathcal{L}(\boldsymbol{y} - \mu^*)(\nabla \mathcal{L}(\boldsymbol{y} - \mu^*))^\top$, where $\mathcal{L}$ is the entry-wise Huber-loss. For simplicity assume that $\Sigma_2^{\mathcal{L}} = \gamma \cdot \mathrm{Id}_d$ for some known $\gamma$ (our approach extends to more general diagonal matrices and unknown $\gamma$ as well – we show how to estimate all relevant parameters from the corrupted sample). Then, if $D_1$ is an $\varepsilon$-corruption of $D_2$, we show that for $\hat{\mu}_1 := \arg\min_{a \in \mathbb{R}^d} \mathbb{E}_{\boldsymbol{y} \sim D_1} \mathcal{L}(\boldsymbol{y} - a)$ it holds that

$$\|\hat{\mu}_1 - \mu^*\| \leqslant O\left(\varepsilon\sqrt{\log(1/\varepsilon)} + \sqrt{\varepsilon\left(\|\Sigma_1^{\mathcal{L}} - \gamma \cdot \mathrm{Id}_d\| + \varepsilon\log(1/\varepsilon)\right)}\right),$$

where $\Sigma_1^{\mathcal{L}}$ is defined analogously to $\Sigma_2^{\mathcal{L}}$. Second, we show that for the filtering approach to work, it is enough that the *transformed distribution*, i.e., $\nabla\mathcal{L}(\boldsymbol{y} - \mu^*)$ for $\boldsymbol{y} \sim D_2$, satisfies the stability condition. This indeed follows since for our choice of $\mathcal{L}$ (the entry-wise Huber-loss), $\nabla\mathcal{L}(\boldsymbol{y} - \mu^*)$ is sub-Gaussian.

However, since we do not work with the quadratic loss anymore, there are several technical obstacles. For the quadratic loss, the gradient is the identity function, and this fact is extensively used in the analysis of the Gaussian setting. For the Huber-loss gradient, different arguments are needed. To exemplify this, consider the following example – note that we do not expect the reader to see at which point in the analysis this step is necessary, it should merely illustrate the types of problems that arise. Let $S_g$ denote the set of uncorrupted samples, $T \subseteq [n]$ be a set of size at least $(1 - \varepsilon)n$, and $\hat{\mu}(T)$ be the minimizer of $\mathcal{L}$ over samples in $T$. In the analysis, terms of the following nature arise

$$\frac{1}{|T \cap S_g|} \sum_{i \in T \cap S_g} \nabla\mathcal{L}\left(\hat{\mu}(T) - \boldsymbol{y}_i^*\right) \nabla\mathcal{L}\left(\hat{\mu}(T) - \boldsymbol{y}_i^*\right)^\top$$

and we need to show that this term is bounded from below, in Loewner order, by $(\gamma - O(\varepsilon \log(1/\varepsilon)))\mathrm{Id}_d$ (uniformly for all $T$ of size at least $(1 - \varepsilon)\,n$). In case of the quadratic loss, this follows easily from the stability conditions by adding and subtracting $\hat{\mu}(T \cap S_g)$.

When $\mathcal{L}$ is the entry-wise Huber-loss, a more sophisticated argument is required. To describe our argument, we assume for simplicity that the entries of $\boldsymbol{\eta}$ are mutually independent. We first show that $\hat{\Delta} := \hat{\mu}(T) - \mu^*$ has entries of magnitude $O(\varepsilon)$. Then we show that for arbitrary but fixed (that is, non-random) $\Delta$ with entries of small magnitude, the distribution of $\nabla\mathcal{L}(\Delta - \boldsymbol{\eta}^*)$ is sub-Gaussian. We use this fact to show that with overwhelming probability [6]

$$\frac{1}{|T \cap S_g|} \sum_{i \in T \cap S_g} \nabla\mathcal{L}\left(\Delta - \boldsymbol{\eta}_i^*\right) \nabla\mathcal{L}\left(\Delta - \boldsymbol{\eta}_i^*\right)^\top \succeq \left(1 - \tilde{O}(\varepsilon)\right) \mathrm{Cov}\left(\nabla\mathcal{L}\left(\Delta - \boldsymbol{\eta}^*\right)\right).$$

Then we use the fact that $\Delta$ has small entries to show that $\mathrm{Cov}\left(\nabla\mathcal{L}\left(\Delta - \boldsymbol{\eta}^*\right)\right) \succeq (\gamma - O(\varepsilon))\,\mathrm{Id}_d$. Finally, we use an $\varepsilon$-net over all possible $\Delta$ to get the desired lower bound uniformly for all $\Delta$, including $\hat{\Delta}$. This last $\varepsilon$-net argument is where we incur the (possibly) sub-optimal $O(d \log d)$ term (instead of the optimal $O(d)$) in our sample complexity. Finally, if the entries of $\boldsymbol{\eta}$ are not independent, we need to use this argument conditioned on the absolute values of the entries of $\boldsymbol{\eta}$.

Summarizing, we have shown how to adjust the well-known filtering technique and incorporate the Huber-loss, to design robust algorithms for symmetric distributions, often matching what is known for the Gaussian distribution in (quasi-)polynomial time.

## Acknowledgments and Disclosure of Funding

This project has received funding from the European Research Council (ERC) under the European Union's Horizon 2020 research and innovation programme (grant agreement No 815464).

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

# A   Overview of Appendices

We will provide a general framework underlying the proofs of Theorem 1.5 and 1.6 in Appendices C to E. We will instantiate it to prove (a slight generalization of) Theorem 1.5 in Appendix F and to prove Theorem 1.6 in Appendix G. We will prove Theorem 1.4 in Appendices H and I. Appendix H contains the proof of identifiability and Appendix I contains the adaptation of the filtering algorithm.

As alluded to in the introduction, we will consider the following (slight) generalization of $(\rho)$-semi-product distributions (Definition 1.2) which allows for even less probability mass around the location. Setting $\alpha = \frac{1}{100}$ recovers Definition 1.2.

**Definition A.1** (Semi-Product Distributions (generalized form))**.** Let $\rho, \alpha > 0$. We say a distribution $\mathcal{D}$ over $\mathbb{R}^d$ is an $(\alpha, \rho)$-*semi-product* distribution, if for $\boldsymbol{\eta} \sim \mathcal{D}$ it holds that

1. For all $j \in [d]$, the distribution of $\boldsymbol{\eta}_j$ is symmetric about 0,

2. For all $j \in [d]$, $\mathbb{P}\left(|\boldsymbol{\eta}_j| \leqslant \rho\right) \geqslant \alpha$,

3. The random vectors $(\mathrm{sign}(\boldsymbol{\eta}_j))_{j=1}^d$ and $(|\boldsymbol{\eta}_j|)_{j=1}^d$ are independent, and the random variables $\mathrm{sign}\left(\boldsymbol{\eta}_1\right), \dots, \mathrm{sign}\left(\boldsymbol{\eta}_d\right)$ are mutually independent.

We will prove the following two (slight) generaliztations of Theorems 1.4 and 1.5. Setting $\alpha = \frac{1}{100}$ recovers Theorems 1.4 and 1.5, respectively.

**Theorem A.2.** *Let $\mu^* \in \mathbb{R}^d, \varepsilon, \rho, \alpha > 0$ and $\mathcal{D}$ be a $(\alpha, \rho)$-semi-product distribution with location $\mu^*$. Let $C > 0$ be a large enough absolute constant and assume that $\sqrt[3]{\varepsilon} \leqslant \alpha/C$ and $n \geqslant C \cdot \frac{d \log(d) + \log(1/\delta)}{\alpha^4 \varepsilon^2 \log(1/\varepsilon)}$. Then, there exists an algorithm that, given an $\varepsilon$-corrupted sample from $\mathcal{D}$ and $\alpha$, runs in time $n^{O(1)}$ and outputs $\hat{\mu} \in \mathbb{R}^d$ such that with probability at least $1 - \delta$ it holds that*

$$\|\hat{\mu} - \mu^*\| \leqslant O\left(\rho \cdot \left[\sqrt{\frac{d \log(d) + \log(1/\delta)}{\alpha^4 n}} + \frac{\varepsilon}{\alpha^3}\sqrt{\log(1/\varepsilon)}\right]\right) .$$

**Theorem A.3.** *Let $\mu^* \in \mathbb{R}^d, \varepsilon, \alpha, \rho > 0$ and $\mathcal{D}$ be a $(\alpha, \rho)$-semi-product distribution with location $\mu^*$. Let $C > 0$ be a large enough absolute constant and assume that $\varepsilon \leqslant \alpha/C$ and $n \geqslant \frac{C}{\alpha^4} \cdot d^{C \log(1/\varepsilon)}$. Then, there exists an algorithm that, given an $\varepsilon$-corrupted sample from $\mathcal{D}$ and $\alpha$, runs in time $n^{O(1)}$ and outputs $\hat{\mu} \in \mathbb{R}^d$ such that with probability at least $1 - \delta$ it holds that*

$$\|\hat{\mu} - \mu^*\| \leqslant O\left(\rho \cdot \left[\sqrt{\frac{d + \log(1/\delta)}{\alpha^4 n}} + \frac{\varepsilon}{\alpha^2}\right]\right) .$$

We remark that if, the algorithm receives $\rho$ as input, the error guarantee improves to

$$\|\hat{\mu} - \mu^*\| \leqslant O\left(\rho \cdot \left[\sqrt{\frac{d + \log(1/\delta)}{\alpha^2 n}} + \frac{\varepsilon}{\alpha}\right]\right) .$$

Note that for the setting of $\alpha = 1/100$ discussed in the introduction, both guarantees are the same (up to constant factors). Further, if $\rho$ is given as input, it is not necessary to receive $\alpha$ as input. We emphasize, that $\rho$ need not be known, and can be estimated from the corrupted samples (given $\alpha$).

We also remark, that our upper bound on the fraction of corruptions is qualitatively tight, since $\varepsilon$ can be at most roughly $\alpha$: Consider a mixture distribution that with probability $1 - \alpha$ outputs a sample from $N(\mu^*, \sigma^2 \mathrm{Id}_d)$ for $\sigma^2$ arbitrarily large and with probability $\alpha$ outputs a sample from $N(\mu^*, \mathrm{Id}_d)$. This is $(\alpha, \Theta(1))$-semi-product. Yet, if an adversary changes the $N(\mu^*, \mathrm{Id}_d)$ component to $N(\mu^*, \sigma^2 \mathrm{Id}_d)$, we cannot hope to achieve error better than $\sigma^2 \sqrt{\frac{d + \log(1/\delta)}{n}}$, which can be arbitrarily large. We did not attempt to optimize the range of $\varepsilon$ allowed for our algorithms.

# B   Preliminaries

**Notation**   We use bold-font for random variables. We use regular (non-bold-font) for random variables that potentially have been adversarially corrupted. Further, we use $\gtrsim$ and $\lesssim$ to suppress absolute constants which do not depend on any other problem parameters. For a (pseudo)-distribution $\mu$, we denote by $\mathrm{supp}(\mu)$ its support.

**Sum-of-Squares Proofs and Pseudo-Expectations**   In this section, we will introduce sum-of-squares proofs and their convex duals, so-called pseudo-distributions.

Let $p\colon \mathbb{R}^n \to \mathbb{R}$ be a polynomial in formal variables $X = (X_1, \ldots, X_n)$. We say the inequality $p(X) \geqslant 0$ has a sum-of-squares proof (short SoS proof) if we can write $p(X)$ as a sum of squares in $X_1, \ldots, X_n$. Further, if every polynomial in this decomposition has degree at most $t$, we say that the sum-of-squares proof has *degree-$t$* and write $\left|\frac{X}{t}\right. p \geqslant 0$. We write $\mathcal{A} \left|\frac{X}{t}\right. p \geqslant p'$ if $\mathcal{A} \left|\frac{X}{t}\right. p - p' \geqslant 0$ and $\mathcal{A} \left|\frac{X}{t}\right. p = p'$ if $\mathcal{A} \left|\frac{X}{t}\right. p \leqslant p'$ and $\mathcal{A} \left|\frac{X}{t}\right. p \geqslant p'$. It is not hard to verify that the following composition rule holds: If $\left|\frac{X}{t}\right. p \geqslant p'$ and $\left|\frac{X}{t'}\right. p' \geqslant p''$, then it also holds that $\left|\frac{X}{t''}\right. p \geqslant p''$, where $t'' = \max\{t, t'\}$.

Next, we introduce so-called pseudo-distributions, the convex duals of sum-of-squares proofs: For $d \in \mathbb{N}_{\geqslant 1}$, a *degree-$d$ pseudo-distribution* is a finitely-supported function $\mu\colon \mathbb{R}^n \to \mathbb{R}$ such that $\sum_{x \in \mathrm{supp}(\mu)} \mu(x) = 1$ and $\sum_{x \in \mathrm{supp}(\mu)} \mu(x) f^2(x) \geqslant 0$ for all polynomials $f$ of degree at most $d/2$. The *pseudo-expectation* corresponding to a pseudo-distribution $\mu$ is defined as the linear operator mapping a degree-$d$ polynomial $f$ to $\tilde{\mathbb{E}}_\mu f := \sum_{x \in \mathrm{supp}(\mu)} \mu(x) f(x)$. We say that a pseudo-distribution $\tilde{\mathbb{E}}_\mu$ satisfies a set of inequalities $\{q_1 \geqslant 0, \ldots, q_m \geqslant 0\}$ at degree $r$, if for all $S \subseteq [m]$ and $h$ a sum-of-squares polynomial such that $\deg(h \cdot \Pi_{j \in S}) \leqslant r$ it holds that

$$\tilde{\mathbb{E}}_\mu h \cdot \Pi_{j \in S} \geqslant 0 \,.$$

We relate pseudo-distributions and sum-of-sqaures proofs using the following facts.

**Fact B.1.** *Let $\mu$ be a degree-$d$ pseudo-distribution that satisfies $\mathcal{A}$ and suppose that $\mathcal{A} \left|\frac{}{t}\right. p \geqslant 0$. Let $h$ be an arbitrary sum-of-squares polynomial, if $\deg h + t \leqslant d$ we have $\tilde{\mathbb{E}}_\mu h \cdot p \geqslant 0$. In particular, we have $\tilde{\mathbb{E}}_\mu p \geqslant 0$ as long as $t \leqslant d$. If $\mu$ only approximately satisfies $\mathcal{A}$ it holds that $\tilde{\mathbb{E}}_\mu p \geqslant -\eta \|h\|_2$ for $\eta = 2^{-n^{\Omega(d)}}$. Further, if there is no degree-$d$ sum-of-squares proof that $\mathcal{A} \left|\frac{}{t}\right. p \geqslant 0$, then for there exists a degree-$d$ pseudo-distribution $\tilde{\mathbb{E}}_\mu$ satisfying $\mathcal{A}$ and $\varepsilon > 0$ such that $\tilde{\mathbb{E}}_\mu p < 0$.*

An essential fact is that pseudo-distributions approximately satisfying a system of polynomial inequalities can be found efficiently as long as the constraints have bit-complexity at most $(m+n)^{\mathcal{O}(1)}$.

**Fact B.2.** *Given any feasible system of polynomial constraints $\mathcal{A} = \{q_1 \geqslant 0, \ldots, q_m \geqslant 0\}$ [7] over $\mathbb{R}^n$ whose bit-complexity satisfies the constraints mentioned above and $\eta = 2^{-n^{\Theta(d)}}$ we can in time $(n + m)^{\mathcal{O}(d)}$ find a degree-$d$ pseudo-distribution that satisfies $\mathcal{A}$ up to error $\eta$. [8]*

## C   Proof of Identifiability based on Transformed Moments

In this section, we give a formal proof of the statement of identifiability based on transformed moments that we sketched in the Techniques section. We will give the proof assuming an abstract set of assumptions and verify that they indeed hold for semi-product and elliptical distributions in Appendices F and G.

We use the same notation as in Appendix H, with the only difference being that we now use an abstract loss function: Let $F\colon \mathbb{R}^d \to \mathbb{R}$ and denote by $f = \nabla F$ its gradient. Further assume that $F$ and $f$ are such that $f$ is 1-Lipschitz. For $w \in \mathbb{R}^n$ with non-negative entries, we define $\mathcal{L}^w\colon \mathbb{R}^d \to \mathbb{R}$ as

$$\mathcal{L}^w(\mu) := \sum_{i=1}^n w_i F(\mu - y_i) \,.$$

It follows that $\nabla \mathcal{L}^w(\mu) = \sum_{i=1}^n w_i f(\mu - y_i)$. Let $2k \in \mathbb{N}_{\geqslant 2}$ and define $\hat{\mu}(w) := \min_{\mu \in \mathbb{R}^d} \mathcal{L}^w(\mu)$ as well as

$$\bar{\mathbf{m}}_{2k} := \max_{v \in \mathbb{S}^{d-1}} \left[ \frac{1}{n} \sum_{i=1}^n \langle f(\boldsymbol{\eta}_i^*), v \rangle^{2k} \right]^{\frac{1}{2k}} \quad \text{and} \quad \hat{\mathbf{m}}_{2k}^w := \max_{v \in \mathbb{S}^{d-1}} \left[ \sum_{i=1}^n w_i \langle f(\hat{\mu} - y_i), v \rangle^{2k} \right]^{\frac{1}{2k}} \,.$$

---

[7] For technical reason we also assume that $\mathcal{A}$ is *explicitly bounded* meaning it contains a constraint of the form $\|x\|_2^2 \leqslant B$ for some large number $B$. Usually we can even take this to be at least polynomial in our variables. We remark that for all the problems we are considering this can be added without changing our proofs.

[8] The positivity and normalization constraint can be satisfied exactly however.

We will prove the following theorem.

**Theorem C.1.** *Let $\alpha \in (0, 1)$ and let $k$ be a positive integer and $w \in \mathcal{W}_{2\varepsilon}$. Suppose that $\varepsilon^{1-\frac{1}{2k}} \leqslant 0.01 \cdot \alpha$ and*

$$\alpha \|\hat{\mu}^w - \mu^*\|^2 \leqslant 10 \langle \nabla \mathcal{L}^w(\mu^*), \mu^* - \hat{\mu}^w \rangle.$$

*Then*

$$\|\hat{\mu}^w - \mu^*\| \leqslant \frac{100}{\alpha} \cdot \left( \left\| \frac{1}{n} \sum_{i=1}^n f(\boldsymbol{\eta}_i^*) \right\| + \varepsilon^{1-\frac{1}{2k}} \cdot (\hat{\mathbf{m}}_{2k}^w + \bar{\mathbf{m}}_{2k}) \right).$$

This theorem easily follows from the following lemma.

**Lemma C.2.** *Let $k$ be a positive integer and $w \in \mathcal{W}_{2\varepsilon}$. Then*

$$\langle \nabla \mathcal{L}^w(\mu^*), u \rangle \leqslant 5 \cdot \|u\| \cdot \left( \left\| \frac{1}{n} \sum_{i=1}^n f(\boldsymbol{\eta}_i^*) \right\| + \varepsilon^{1-\frac{1}{2k}} \cdot (\hat{\mathbf{m}}_{2k}^w + \bar{\mathbf{m}}_{2k} + \|u\|) \right),$$

*where $u = \hat{\mu}(w) - \mu^*$.*

We first give the proof of Theorem C.1.

*Proof of Theorem C.1.* Let $u = \hat{\mu}(w) - \mu^*$. Using Lemma C.2 and $\varepsilon^{1-\frac{1}{2k}} \leqslant 0.01 \cdot \alpha$, cancelling the common factor of $\|u\|$ and rearranging now yields

$$\|u\| \leqslant \frac{100}{\alpha} \cdot \left( \left\| \frac{1}{n} \sum_{i=1}^n f(\boldsymbol{\eta}_i) \right\| + \varepsilon^{1-\frac{1}{2k}} \cdot (\hat{\mathbf{m}}_{2k}^w + \bar{\mathbf{m}}_{2k}) \right).$$

$\square$

Next, we prove Lemma C.2.

*Proof of Lemma C.2.* Again, let $u = \hat{\mu}(w) - \mu^*$. We can decompose $\langle \nabla \mathcal{L}^w(\mu^*), u \rangle$ as

$$\langle \nabla \mathcal{L}^w(\mu^*), u \rangle = \sum_{i=1}^n w_i \langle f(\mu^* - y_i), u \rangle$$

$$= \sum_{i=1}^n w_i \langle f(\mu^* - \boldsymbol{y}_i^*), u \rangle + \sum_{i=1}^n \mathbf{1}\{i \in S_b\} \cdot w_i \langle f(\mu^* - y_i) - f(\mu^* - \boldsymbol{y}_i^*), u \rangle.$$

For the first sum,

$$\sum_{i=1}^n w_i \langle f(\mu^* - \boldsymbol{y}_i^*), u \rangle = \sum_{i=1}^n w_i \langle f(\boldsymbol{\eta}_i^*), u \rangle \leqslant \left\| \frac{1}{n} \sum_{i=1}^n f(\boldsymbol{\eta}_i^*) \right\| \cdot \|u\| + \sum_{i=1}^n \left( w_i - \tfrac{1}{n} \right) \langle f(\boldsymbol{\eta}_i^*), u \rangle.$$

By Hölder's Inequality,

$$\sum_{i=1}^n \left( w_i - \tfrac{1}{n} \right) \langle f(\boldsymbol{\eta}_i^*), u \rangle \leqslant \left( \sum_{i=1}^n \left| w_i - \tfrac{1}{n} \right| \right)^{1-\frac{1}{2k}} \left( \sum_{i=1}^n \left| w_i - \tfrac{1}{n} \right| \cdot \langle f(\boldsymbol{\eta}_i^*), u \rangle^{2k} \right)^{\frac{1}{2k}}$$

$$\leqslant 4 \cdot \varepsilon^{1-\frac{1}{2k}} \cdot \bar{\mathbf{m}}_{2k} \cdot \|u\|.$$

For the second sum, we bound using Hölder's Inequality

$$\sum_{i=1}^n \mathbf{1}\{i \in S_b\} \cdot w_i \cdot \langle f(\mu^* - y_i) - f(\mu^* - \boldsymbol{y}_i^*), u \rangle \leqslant \varepsilon^{1-\frac{1}{2k}} \left( \sum_{i=1}^n w_i \cdot \langle f(\mu^* - y_i) - f(\mu^* - \boldsymbol{y}_i^*), u \rangle^{2k} \right)^{\frac{1}{2k}}$$

$$= \varepsilon^{1-\frac{1}{2k}} \left( \sum_{i=1}^n w_i \cdot \langle f(\mu^* - y_i) - f(\mu^* - \boldsymbol{y}_i^*) - f(\hat{\mu} - y_i) + f(\hat{\mu} - y_i), u \rangle^{2k} \right)^{\frac{1}{2k}}$$

$$\leqslant \varepsilon^{1-\frac{1}{2k}}\left(\sum_{i=1}^{n} w_i \left\langle f\left(\boldsymbol{\eta}_i^*\right), u\right\rangle^{2k}\right)^{\frac{1}{2k}} + \varepsilon^{1-\frac{1}{2k}}\left(\sum_{i=1}^{n} w_i \left\langle f\left(\hat{\mu} - y_i\right), u\right\rangle^{2k}\right)^{\frac{1}{2k}}$$

$$+ \varepsilon^{1-\frac{1}{2k}}\left(\sum_{i=1}^{n} w_i \left\langle f\left(\mu^* - y_i\right) - f\left(\hat{\mu} - y_i\right), u\right\rangle^{2k}\right)^{\frac{1}{2k}}.$$

We bound the three terms in the square bracket one by one.

The first term can be bounded as follows

$$\sum_{i=1}^{n} w_i \left\langle f\left(\boldsymbol{\eta}_i^*\right), u\right\rangle^{2k} \leqslant \sum_{i=1}^{n} \tfrac{1}{n} \left\langle f\left(\boldsymbol{\eta}_i^*\right), u\right\rangle^{2k} \leqslant \bar{\mathbf{m}}_{2k}^{2k} \|u\|^{2k}.$$

The second one is at most $\bar{\mathbf{m}}_{2k}\|u\|$ by definition of $\hat{\mathbf{m}}_{2k}^w$. For the third, we use that $f$ is 1-Lipschitz to bound

$$\left(\sum_{i=1}^{n} w_i \cdot \left\langle f\left(\mu^* - y_i\right) - f\left(\hat{\mu} - y_i\right), u\right\rangle^{2k}\right)^{\frac{1}{2k}} \leqslant \left(\sum_{i=1}^{n} w_i \cdot \left\| f\left(\mu^* - y_i\right) - f\left(\hat{\mu} - y_i\right)\right\|^{2k} \|u\|^{2k}\right)^{\frac{1}{2k}} \leqslant \|u\|^2.$$

It follows that

$$\sum_{i=1}^{n} \mathbf{1}\left\{i \in S_b\right\} \cdot w_i \cdot \left\langle f\left(\mu^* - y_i\right) - f\left(\mu^* - \boldsymbol{y}_i^*\right), u\right\rangle \leqslant \varepsilon^{1-\frac{1}{2k}} \|u\| \cdot \left(\hat{\mathbf{m}}_{2k}^w + \bar{\mathbf{m}}_{2k} + \|u\|\right).$$

Putting all the above together, we obtain the desired bound. $\qquad\square$

## D   Filtering Algorithm for Bounded Clipped Covariance

In this section, we show that the filtering algorithm for robust mean estimation under moment constraints (cf. for example [Li19, DK22]) can be adapted to also yield an efficient algorithm for robustly estimating the location parameter of symmetric distributions. For now, we focus on the case that $k = 2$, we discuss the extension to higher moments in Appendix E. Again, we will focus on the abstract setting and show that we can apply these results to semi-product and elliptical distributions. We will closely follow the exposition in [Li19, DK22].

Recall that we denote $\mathcal{L}^w(\mu) = \sum_{i=1}^{n} w_i F\left(\mu - y_i\right)$.

**Assumption D.1** (Goodness Condition). *Let $F \colon \mathbb{R}^d \to \mathbb{R}$, $f = \nabla F$ be such that $f$ is 1-Lipschitz. Let $\mu^* \in \mathbb{R}^d, 0 < \sigma, 0 < \alpha < 1$ and $y_1, \ldots, y_n \in \mathbb{R}^d$. Let $D$ be a distribution over $\mathbb{R}^d$ and $\boldsymbol{\eta}_1^*, \ldots, \boldsymbol{\eta}_n^*$ be $n$ i.i.d. samples from $D$. We say that the goodness condition holds, if*

1. $\alpha\|\hat{\mu}^w - \mu^*\|^2 \leqslant 10 \left\langle \nabla \mathcal{L}^w\left(\mu^*\right), \mu^* - \hat{\mu}^w\right\rangle,$

2. $\left\| \frac{1}{n}\sum_{i=1}^{n} f\left(\boldsymbol{\eta}_i^*\right)\right\| \leqslant \frac{\alpha}{100}\sigma,$

3. $\bar{\mathbf{m}}_2 = \max_{v \in \mathbb{S}^{d-1}} \sqrt{\frac{1}{n}\sum_{i=1}^{n} \left\langle f\left(\boldsymbol{\eta}_i^*\right), v\right\rangle^2} \leqslant \sigma.$

Under this condition we can show the following theorem.

**Theorem D.2.** *Let $\mu^* \in \mathbb{R}^d, 0 < \sigma, 0 < \alpha < 1, \varepsilon > 0$, and $\boldsymbol{\eta}_1^*, \ldots, \boldsymbol{\eta}_n^*$ be $n$ i.i.d. samples from some distribution $D$ over $\mathbb{R}^d$. Assume that $\sqrt{\varepsilon} \leqslant \frac{\alpha}{1000}$. Let $y_1, \ldots, y_n$ be an $\varepsilon$-corruption of $\boldsymbol{y}_1^* = \mu^* + \boldsymbol{\eta}_1^*, \ldots, \boldsymbol{y}_n^* = \mu^* + \boldsymbol{\eta}_n^*$. Assume Assumption D.1 (the goodness condition) holds, then Algorithm D.3, given $y_1, \ldots, y_n$ and $\sigma$, terminates after at most $2\varepsilon n + 1$ iterations and computes $\hat{\mu}$ such that*

$$\|\hat{\mu}^w - \mu^*\| \leqslant O\left(\frac{\left\|\frac{1}{n}\sum_{i=1}^{n} f\left(\boldsymbol{\eta}_i^*\right)\right\|}{\alpha} + \frac{\sqrt{\varepsilon}}{\alpha}\cdot\sigma\right).$$

*Moreover each iteration can be implemented in time $n^{O(1)} d^{O(1)}$.*

To describe the algorithm, let $C > 0$ be some universal constant and for $\mu \in \mathbb{R}^d, w \in \mathcal{W}_\varepsilon$ let

$$\hat{\mu}(w) = \min_{\mu \in \mathbb{R}^d} \mathcal{L}^w(\mu) \ ,$$

$$\Sigma_f(w) = \sum_{i=1}^n w_i f\left(\hat{\mu}^w - y_i\right) f\left(\hat{\mu}^w - y_i\right)^\top \ .$$

We will use the following Filtering Algorithm:

---

**Algorithm D.3** (Filtering Algorithm).
**Input:** $\varepsilon$-corrupted sample $y_1, \ldots, y_n$ and $\sigma > 0$.
**Output:** Location estimate $\hat{\mu}$.

- Let $w^{(0)} = \frac{1}{n}\not\Vdash_n$.

- Compute $\hat{\mu}^{(0)} = \min_{\mu \in \mathbb{R}^d} \mathcal{L}^{w^{(0)}}(\mu)$ and $\Sigma_f^{(0)} = \Sigma_f\left(w^{(0)}\right)$.

- Let $t = 0$.

- **while** $\left\|\Sigma_f^{(t)}\right\| > 100 \cdot \sigma$ **do**

    - Compute $v^{(t)}$ the top eigenvector of $\Sigma_f^{(t)}$.
    - For $i \in [n]$, compute $\tau_i^{(t)} = \left\langle v^{(t)}, f\left(\hat{\mu}^{(t)} - y_i\right)\right\rangle^2$.
    - For $i \in [n]$, set $w_i^{(t+1)} = \left(1 - \frac{\tau_i}{\tau_{\max}}\right) w_i^{(t)}$, where $\tau_{\max} = \max_{i \in [n]} \tau_i$.
    - Compute $\hat{\mu}^{(t+1)} = \min_{\mu \in \mathbb{R}^d} f^{w^{(t+1)}}(\mu)$ and $\Sigma_f^{(t+1)} = \Sigma_f\left(w^{(t+1)}\right)$.
    - $t \leftarrow t + 1$.

- Output $\hat{\mu}^{(t)}$.

---

The proof of Theorem D.2 is very similar to the one in [Li19]. We will use the following lemma whose proof we will give at the end of this section

**Lemma D.4.** *Assume that Theorem D.2* $\|\Sigma_f^{(t)}\| > 100 \cdot \sigma$ *and*

$$\sum_{i \in S_g} \left(\frac{1}{n} - w_i^{(t)}\right) < \sum_{i \in S_b} \left(\frac{1}{n} - w_i^{(t)}\right) \ .$$

*Then*

$$\sum_{i \in S_g} \left(\frac{1}{n} - w_i^{(t+1)}\right) < \sum_{i \in S_b} \left(\frac{1}{n} - w_i^{(t+1)}\right) \ .$$

With this in hand, we will prove Theorem D.2.

*Proof of Theorem D.2.* First note, that every iteration can clearly be implented to run in time $n^{O(1)} d^{O(1)}$. First, we will show that the algorithm terminates after at most $\lceil 2\varepsilon n \rceil$ iterations. We will prove it by contradiction. Suppose that the algorithm does not terminate after $T = \lceil 2\varepsilon n \rceil$ iterations. Note that the number of zero entries of $w^{(t)}$ increases by at least 1 in every iteration. Hence, after $T$ iterations we have set at least $\varepsilon n$ entries of $w$ to zero whose index lies in $S_g$. By assumption that the algorithm didn't terminate and Lemma D.4 it holds that

$$\varepsilon \leqslant \sum_{i \in S_g} \left(\frac{1}{n} - w_i^{(T)}\right) < \sum_{i \in S_b} \left(\frac{1}{n} - w_i^{(T)}\right) \leqslant \frac{|S_b|}{n} \leqslant \varepsilon \ .$$

which is a contradiction.

Next, we prove the correctness of the algorithm. Let $T$ be the index of the last iteration of the algorithm before termination. Note that by our invariant

$$\left\|\frac{1}{n} - w^{(T)}\right\|_1 = \sum_{i \in S_g} \frac{1}{n} - w_i^{(T)} + \sum_{i \in S_b} \frac{1}{n} - w_i^{(T)} < 2\sum_{i \in S_b} \frac{1}{n} - w_i^{(T)} \leqslant 2\varepsilon \ .$$

Since also $0 \leqslant w^{(T)} \leqslant \frac{1}{n}$, it follows that $w^{(T)} \in \mathcal{W}_{2\varepsilon}$. By Theorem C.1 and since $\left\| \Sigma_f^{(T)} \right\| \leqslant 100 \cdot \sigma$ it follows that

$$\|\hat{\mu}^w - \mu^*\| \leqslant O\left( \frac{\left\| \frac{1}{n} \sum_{i=1}^n f\left(\boldsymbol{\eta}_i^*\right)\right\|}{\alpha} + \frac{\sqrt{\varepsilon}}{\alpha} \cdot \sigma \right).$$

$\square$

Lastly, we will prove Lemma D.4.

*Proof of Lemma D.4.* For simplicity, let $w = w^{(t)}$ and $w' = w^{(t+1)}$. Note, that it is enough to show that

$$\sum_{i \in S_g} w_i - w_i' < \sum_{i \in S_b} w_i - w_i'.$$

Further, recall that $w_i' = \left(1 - \frac{\tau_i}{\tau_{\max}}\right) w_i$, so for all $i \in [n]$, $w_i - w_i' = \frac{1}{\tau_{\max}} \tau_i w_i$. Hence is enough to show that

$$\sum_{i \in S_g} \tau_i w_i < \sum_{i \in S_b} \tau_i w_i.$$

Since $S_g$ and $S_b$ partition $[n]$ and

$$\sum_{i=1}^n w_i \tau_i = \sum_{i=1}^n w_i \left\langle v^{(t)}, f\left(\hat{\mu}^{(t)} - y_i\right)\right\rangle^2 = \left(v^{(t)}\right)^\top \Sigma_f^{(t)} \left(v^{(t)}\right) = \left\| \Sigma_f^{(t)} \right\|,$$

we can prove $\sum_{i \in S_g} \tau_i w_i < \sum_{i \in S_b} \tau_i w_i$ by showing that

$$\sum_{i \in S_g} \tau_i w_i < \frac{\|\Sigma_f^{(t)}\|}{2}.$$

To this end, note that

$$\sum_{i \in S_g} \tau_i w_i = \sum_{i \in S_g} w_i \left\langle v^{(t)}, f\left(\hat{\mu}^{(t)} - y_i\right)\right\rangle^2$$

$$\leqslant \frac{2}{n} \sum_{i \in [n]} \left\langle v^{(t)}, f\left(\mu^* - y_i\right)\right\rangle^2 + \frac{2}{|S_g|} \sum_{i \in S_g} \left\langle v^{(t)}, f\left(\hat{\mu}^{(t)} - y_i\right) - f\left(\mu^* - y_i\right)\right\rangle^2.$$

The first term is at most $2\sigma^2$. For the second term we observe that since $f$ is 1-Lipschitz, for all $i \in [n]$ it holds that

$$\left\langle v^{(t)}, f\left(\hat{\mu}^{(t)} - y_i\right) - f\left(\mu^* - y_i\right)\right\rangle^2 \leqslant \left\| f\left(\hat{\mu}^{(t)} - y_i\right) - f\left(\mu^* - y_i\right)\right\|^2 \leqslant \left\| \hat{\mu}^{(t)} - \mu^*\right\|^2.$$

Hence, by Theorem C.1,

$$\sum_{i \in S_g} \tau_i w_i \leqslant 2\sigma^2 + \left\| \hat{\mu}^{(t)} - \beta^*\right\|^2 \leqslant 2\sigma^2 + \frac{100^2}{\alpha^2} \left( \left\| \frac{1}{n} \sum_{i=1}^n f\left(\boldsymbol{\eta}_i^*\right)\right\| + \sqrt{\varepsilon} \cdot \Sigma_f^{(t)}\right)^2 < \frac{\Sigma_f^{(t)}}{2},$$

where we used $\varepsilon \lesssim \alpha^2$ and $\left\| \frac{1}{n} \sum_{i=1}^n f\left(\boldsymbol{\eta}_i^*\right)\right\| \lesssim \alpha\sigma$. $\square$

# E   Filtering Algorithm for Bounded Higher-Order Clipped Moments

In this section, we show how to adapt the filtering algorithm to incorporate higher-order moments of the transformed noise. We follow closely the exposition in [DK22, Chapter 6.5]. In particular, consider the following definition similar to the by now standard notion of ceritfiably bounded moments or certifiable subgaussianity [KSS18, HL18]. Let $f \colon \mathbb{R}^d \to \mathbb{R}^d$ be some function.

**Definition E.1.** Let $k, \ell \in \mathbb{N}$ and $\sigma > 0$. We say a distribution $D$ over $\mathbb{R}^d$ has $(2k, \ell)$-*certifiably $\sigma$-bounded $f$-moments* if there is a degree-$\ell$ SoS proof (cf. Appendix B), in formal variables $v$, that

$$\mathop{\mathbb{E}}_{\boldsymbol{\eta} \sim D} \langle f\left(\boldsymbol{\eta}\right), v\rangle^{2k} \leqslant (\sigma \cdot \|v\|)^{2k}.$$

We again define a goodness condition analogous to Assumption D.1. Specifically, we assume

**Assumption E.2** (Higher-Order Goodness Condition). Let $F\colon \mathbb{R}^d \to \mathbb{R}$, $f = \nabla F$ be such that $f$ is 1-Lipschitz. Let $\mu^* \in \mathbb{R}^d, 0 < \sigma, 0 < \alpha < 1, k \in \mathbb{N}$ and $y_1, \ldots, y_n \in \mathbb{R}^d$. Let $D$ be a distribution over $\mathbb{R}^d$ and $\boldsymbol{\eta}_1^*, \ldots, \boldsymbol{\eta}_n^*$ be $n$ i.i.d. samples from $D$. We say that the *higher-order goodness condition* holds, if

1. $\alpha \|\hat{\mu}^w - \mu^*\|^2 \leqslant 10 \langle \nabla \mathcal{L}^w (\mu^*), \mu^* - \hat{\mu}^w \rangle$,

2. $\left\| \frac{1}{n} \sum_{i=1}^n f(\boldsymbol{\eta}_i^*) \right\| \leqslant \frac{\alpha}{100} \sigma$,

3. the uniform distribution over the set $\{\boldsymbol{\eta}_1^*, \ldots, \boldsymbol{\eta}_n^*\}$ has $(2k, \ell)$-certifiably $\sigma$-bounded $f$-moments. In other words,

$$\left| \frac{v}{\ell} \frac{1}{n} \sum_{i=1}^n \langle f(\boldsymbol{\eta}_i^*), v \rangle^{2k} \leqslant (\sigma \cdot \|v\|)^{2k} \right. .$$

Specifically, we will show

**Theorem E.3.** *Let $\mu^* \in \mathbb{R}^d, 0 < \sigma, 0 < \alpha < 1, k \in \mathbb{N}$, and $\boldsymbol{\eta}_1^*, \ldots, \boldsymbol{\eta}_1^*$ be $n$ i.i.d. samples from some distribution $D$ over $\mathbb{R}^d$. Assume that $\varepsilon^{1-\frac{1}{2k}} \leqslant \frac{\alpha}{1000}$. Let $y_1, \ldots, y_n$ be an $\varepsilon$-corruption of $y_1^* = \mu^* + \boldsymbol{\eta}_1^*, \ldots, y_n^* = \mu^* + \boldsymbol{\eta}_n^*$. Assume Assumption E.2 (the higher-order goodness condition) holds, then Algorithm E.4, given $y_1, \ldots, y_n$ and $\sigma$, terminates after at most $2\varepsilon n + 1$ iterations and computes $\hat{\mu}$ such that*

$$\|\hat{\mu} - \mu^*\| \leqslant O\left( \frac{\left\| \frac{1}{n} \sum_{i=1}^n f(\boldsymbol{\eta}_i^*) \right\|}{\alpha} + \frac{\varepsilon^{1-\frac{1}{2k}}}{\alpha} \cdot \sigma \right) .$$

*Moreover each iteration can be implemented in time $n^{O(1)} \cdot d^{O(\ell)}$.*

Additionally to the notation introduced in Appendix D, consider the following polynomial in formal variables $v$

$$p^w(v) = \sum_{i=1}^n w_i \langle v, f(y_i - \hat{\mu}) \rangle^{2k} .$$

In what follows we will write $p^{w^{(t)}}$ as just $p^{(t)}$.

---

**Algorithm E.4** (Filtering Algorithm).
**Input:** $\varepsilon$-corrupted sample $y_1, \ldots, y_n$ and $\sigma > 0$.
**Output:** Location estimate $\hat{\mu}$.

- Let $w^{(0)} = \frac{1}{n}$.

- Compute $\hat{\mu}^{(0)} = \min_{\mu \in \mathbb{R}^d} \mathcal{L}^{w^{(0)}}(\mu)$.

- Let $t = 0$.

- **while** there is no degree-$\ell$ SoS proof (in variables $v$) that $p^{(t)}(v) \leqslant (100 \cdot \sigma \cdot \|v\|)^{2k}$ **do**

  - Find a degree-$\ell$ pseudo-expectation that satisfies the constraint $\|v\|^2 = 1$ such that $\tilde{\mathbb{E}} p^t(v) > (99 \cdot \sigma)^{2k}$.

  - For $i \in [n]$, compute $\tau_i^{(t)} = \tilde{\mathbb{E}} \langle v, f(\hat{\mu}^{(t)} - y_i) \rangle^{2k}$.

  - For $i \in [n]$, set $w_i^{(t+1)} = \left(1 - \frac{\tau_i}{\tau_{\max}}\right) w_i^{(t)}$, where $\tau_{\max} = \max_{i \in [n]} \tau_i$.

  - Compute $\hat{\mu}^{(t+1)} = \min_{\mu \in \mathbb{R}^d} f^{w^{(t+1)}}(\mu)$.

  - $t \leftarrow t + 1$.

- Output $\hat{\mu}^{(t)}$.

---

The proof of Theorem E.3 is very similar to the one of Theorem D.2. Again, we will use the following lemma whose proof we will give at the end of this section

**Lemma E.5.** *Assume that*

$$\sum_{i \in S_g} \frac{1}{n} - w_i^{(t)} < \sum_{i \in S_b} \frac{1}{n} - w_i^{(t)}$$

*and $p^t(v) \leqslant \left(100 \cdot \sigma \cdot \|v\|^2\right)^k$ does not have a degree-$\ell$ SoS proof. Then also*

$$\sum_{i \in S_g} \frac{1}{n} - w_i^{(t+1)} < \sum_{i \in S_b} \frac{1}{n} - w_i^{(t+1)} .$$

With this in hand, we will prove Theorem E.3.

*Proof of Theorem E.3.* First, note that by the facts in Appendix B we can compute the pseudo-expectation in each iteration in time $n^{O(1)} d^{O(l)}$ and hence, every iteration can be implemented to run in this time. Exactly as in the proof of Theorem D.2 we can conclude that the algorithm terminates after $T \leqslant 2\varepsilon n + 1$ iterations and that $w^T \in \mathcal{W}_{2\varepsilon}$.

Recall that

$$\hat{\mathfrak{m}}_{2k}^w = \max_{v \in \mathbb{S}^{d-1}} \left[ \sum_{i=1}^n w_i \left\langle f\left(\hat{\mu} - y_i\right), v \right\rangle^{2k} \right]^{\frac{1}{2k}} = \max_{v \in \mathbb{S}^{d-1}} \left[ p^w(v) \right]^{\frac{1}{2k}} .$$

By Theorem C.1 it follows that

$$\left\| \hat{\mu}^{(T)} - \mu^* \right\| \leqslant O \left( \frac{\left\| \frac{1}{n} \sum_{i=1}^n f\left(\boldsymbol{\eta}_i^*\right) \right\|}{\alpha} + \frac{\varepsilon^{1-1/(2k)} \cdot \left( \hat{\mathfrak{m}}_{2k}^{w^T} + \bar{\mathfrak{m}}_{2k} \right)}{\alpha} \right) .$$

Note that since the uniform distribution over $\{\boldsymbol{\eta}_1^*, \ldots, \boldsymbol{\eta}_n^*\}$ has $(2k, \ell)$-certifiably $\sigma$-bounded $f$-moments, it holds that $\bar{\mathfrak{m}}_{2k} \leqslant \sigma$. Since the algorithm terminates after $T$ iterations it holds that $\hat{\mathfrak{m}}_{2k}^{w^T} \leqslant 100\sigma$ which completes the proof. $\qquad \square$

Next, we will prove Lemma E.5.

*Proof of Lemma E.5.* Again, for simplicity, let $w = w^{(t)}$ and $w' = w^{(t+1)}$. As in the proof of Lemma D.4, it is enough to show that

$$\sum_{i \in S_g} \tau_i w_i < \sum_{i \in S_b} \tau_i w_i .$$

Further,

$$\sum_{i=1}^n w_i \tau_i = \sum_{i=1}^n w_i \tilde{\mathbb{E}} \left\langle v, f\left(\hat{\mu}^{(t)} - y_i\right) \right\rangle^k = \tilde{\mathbb{E}} p^t(v) \geqslant (99 \cdot \sigma)^{2k} .$$

Using Fact K.1, we continue to bound

$$\sum_{i \in S_g} \tau_i w_i = \sum_{i \in S_g} w_i \tilde{\mathbb{E}} \left\langle v, f\left(\hat{\mu}^{(t)} - y_i\right) \right\rangle^{2k}$$

$$\leqslant 4^k \cdot \left[ \frac{1}{n} \sum_{i \in [n]} \tilde{\mathbb{E}} \left\langle v, f\left(\boldsymbol{\eta}_i^*\right) \right\rangle^{2k} + \frac{1}{|S_g|} \sum_{i \in S_g} \tilde{\mathbb{E}} \left\langle v, f\left(\hat{\mu}^{(t)} - y_i\right) - f\left(\mu^* - y_i\right) \right\rangle^{2k} \right] .$$

The first term is at most $\sigma^{2k}$. For the second term, notice that by Fact K.2

$$\left|\frac{v}{2k} \left\langle v, f\left(\hat{\mu}^{(t)} - y_i\right) - f\left(\mu^* - y_i\right) \right\rangle^{2k} \leqslant \|v\|^{2k} \left\| f\left(\hat{\mu}^{(t)} - y_i\right) - f\left(\mu^* - y_i\right) \right\|^{2k} \leqslant \|v\|^{2k} \left\| \hat{\mu}^{(t)} - \mu^* \right\|^{2k} .$$

Hence, the second term is at most $\left\| \hat{\mu}^{(t)} - \mu^* \right\|^{2k}$. By Theorem C.1 it follows that

$$\left\| \hat{\mu}^{(T)} - \mu^* \right\| \leqslant 100 \cdot \left( \frac{\left\| \frac{1}{n} \sum_{i=1}^n f\left(\boldsymbol{\eta}_i^*\right) \right\|}{\alpha} + \frac{\varepsilon^{1-1/(2k)} \cdot \left( \hat{\mathfrak{m}}_{2k}^{w^t} + \bar{\mathfrak{m}}_{2k} \right)}{\alpha} \right) .$$

Recall that $\bar{\mathbf{m}}_{2k} \leqslant \sigma$. Since

$$\hat{\mathbf{m}}_{2k}^w = \max_{v \in \mathbb{S}^{d-1}} \left[ \sum_{i=1}^n w_i \langle f\left(\hat\mu - y_i\right), v \rangle^{2k} \right]^{\frac{1}{2k}} = \max_{v \in \mathbb{S}^{d-1}} \left[ p^w(v) \right]^{\frac{1}{2k}},$$

$\varepsilon^{1-\frac{1}{2k}} \leqslant \frac{\alpha}{1000}$ and $\left\| \frac{1}{n} \sum_{i=1}^n f\left(\boldsymbol{\eta}_i^*\right) \right\| \leqslant \frac{\alpha}{100}\sigma$, we get

$$\sum_{i \in S_g} \tau_i w_i < 0.2 \cdot \tilde{\mathbb{E}} p^t(v) + 0.1 \cdot (99 \cdot \sigma)^{2k} < \tilde{\mathbb{E}} p^t(v)/2.$$

$\square$

## F  Product Distributions

In this section we prove that $(\alpha, \rho)$-semi-product distributions satisfy Assumption D.1 with $\sigma = O(\rho)$ and with $F$ equals to the entry-wise Huber loss function with parameter $2\rho$:

$$F(x) = \sum_{j=1}^d \Phi_{2\rho}\left(x_j\right)$$

with probability $1 - \delta$ as long as $n \gtrsim \frac{d + \log(1/\delta)}{\alpha^4}$. Moreover, if in addition $n \gtrsim \frac{1}{\alpha^4} \cdot d^k \cdot \log\left(d/\delta\right)$, then Assumption E.2 is also satisfied. We can use this to prove Theorem A.3 as follows: Let $\boldsymbol{\eta}_1^*, \dots \boldsymbol{\eta}_n^*$ follow an $(\alpha, \rho)$-semi-product distribution. We will see in Lemma F.3 that for any $n$, it holds with probability at least $1 - \delta$ that $\left\| \frac{1}{n} \sum_{i=1}^n f(\boldsymbol{\eta}_i^*) \right\| \leqslant \rho \cdot \sqrt{\frac{d + \log(1/\delta)}{n}}$. Hence, by Theorem E.3 (or Algorithm D.3 for $k = 1$) it follows that as long as $n \gtrsim \frac{1}{\alpha^4} \cdot d^k \log(d/\delta)$ (or $n \gtrsim \frac{d + \log(1/\delta)}{\alpha^4}$ for $k = 1$) our estimator achieves error

$$O\left( \frac{\left\| \frac{1}{n} \sum_{i=1}^n f\left(\boldsymbol{\eta}_i^*\right) \right\|}{\alpha} + \frac{\varepsilon^{1-\frac{1}{2k}}}{\alpha} \cdot \sigma \right) = O\left( \rho \cdot \left[ \sqrt{\frac{d + \log(1/\delta)}{\alpha^4 n}} + \frac{\varepsilon^{1-\frac{1}{2k}}}{\alpha} \right] \right).$$

Letting $k = O(\log(1/\varepsilon))$ proves Theorem A.3. Note that similarly to Theorem 1.6, for $1 \leqslant k \leqslant O(\log(1/\varepsilon))$ we get a sequence of algorithms interpolating between error $\sqrt{\varepsilon}$ and $\varepsilon$ (using roughly $d^k$ samples).

Note that $f = \nabla F$ is indeed 1-Lipschitz since the derivative of the Huber loss is Lipschitz. In the next to lemmas we prove the first assumption from Assumption D.1.

**Lemma F.1.** *Let $w \in \mathcal{W}_{2\varepsilon}$ for $\varepsilon \lesssim \alpha$. Suppose that $n \gtrsim \log(d)/\alpha$. Then with probability $1 - \exp\left(-\Omega\left(\alpha n\right)\right)$ for all $u \in \mathbb{R}^d$ such that $\|u\|_{\max} \leqslant \rho$,*

$$\mathcal{L}^w\left(\mu^* + u\right) - \mathcal{L}^w\left(\mu^*\right) - \langle \nabla \mathcal{L}^w\left(\mu^*\right), u \rangle \geqslant \frac{\alpha}{4} \cdot \|u\|^2.$$

*Proof.* Let $j \in [d]$ and $\zeta_i = \mu - y_i$.

$$\mathcal{L}_j^w\left(\mu_j^* + u_j\right) - \mathcal{L}_j^w\left(\mu_j^*\right) - \left(\mathcal{L}_j^w\right)'\left(\mu_j^*\right) \cdot u_j = \sum_{i=1}^n w_i \left[ \Phi\left(\zeta_{ij} + u_j\right) - \Phi\left(\zeta_{ij}\right) - u_j \cdot \phi\left(\zeta_{ij}\right) \right].$$

Let $M$ be the set of uncorrupted samples such that $|\zeta_{ij}| = |\boldsymbol{\eta}_{ij}| \leqslant h/2$. Note that for $i \in M$, $\Phi\left(\zeta_{ij} + u_j\right) = \frac{1}{2}|\zeta_{ij} + u_j|^2$ and $\Phi\left(\zeta_{ij} + u_j\right) = \frac{1}{2}|\zeta_{ij}|^2$. Hence

$$\sum_{i \in M} w_i \left[ \Phi\left(\zeta_{ij} + u_j\right) - \Phi\left(\zeta_{ij}\right) - u_j \cdot \phi\left(\zeta_{ij}\right) \right] = \frac{1}{2} \sum_{i \in M} w_i u_j^2.$$

And by convexity,

$$\sum_{i \notin M} w_i \left[ \Phi\left(\|\zeta_i + u\|\right) - \Phi\left(\|\zeta_i\|\right) - \langle f\left(\zeta_i\right), u \rangle \right] \geqslant 0.$$

Since $w \in \mathcal{W}_{2\varepsilon}$,

$$\sum_{i \in M} \left| w_i - \frac{1}{n} \right| \leqslant 2\varepsilon \,.$$

Hence

$$\sum_{i \in M} w_i \geqslant \frac{|M|}{n} - 2\varepsilon \,.$$

By a Chernoff bound,

$$|M| \geqslant 0.9\alpha - \varepsilon \geqslant \alpha/2$$

with probability $1 - \exp(-\alpha n)$. The result follows from a union bound over all $j \in [d]$. $\qquad \square$

**Lemma F.2.** *Let $w \in \mathcal{W}_{2\varepsilon}$ for $\varepsilon \lesssim \alpha$, and suppose that $n \gtrsim \log(d)/\alpha^2$. Then with probability $1 - \exp\left(-\Omega\left(\alpha^2 n\right)\right)$,*

$$\langle \nabla \mathcal{L}^w (\mu^*), \mu^* - \hat{\mu}^w \rangle \geqslant \frac{\alpha}{4} \| \hat{\mu}^w - \mu^* \|^2 \,.$$

*Proof.* Fix $j \in d$ and let $u_j = \mu_j^* - \hat{\mu}_j(w)$. Let $u_j'$ and $t$ be such that

$$|u_j'| \leqslant \rho \,,$$
$$\hat{\mu}_j(w) = \mu_j^* + t \cdot u_j' \,.$$

It follows that $t = \max\{1, |u_j|\}$. Clearly, $\hat{\mu}_j(w)$ is the (unique) minimizer of $\mathcal{L}_j^w$. Let $\hat{\mu}_j' = \mu_j^* + u_j'$, by convexity of $\mathcal{L}_j^w$ it holds that $\mathcal{L}_j^w(\hat{\mu}_j') \leqslant \mathcal{L}_j^w(\mu_j^*)$. Since $|u_j'| \leqslant \rho$ it follows by Lemma F.1 that

$$\mathcal{L}_j^w(\mu_j^*) \geqslant \mathcal{L}_j^w(\hat{\mu}_j') \geqslant \mathcal{L}_j^w(\mu_j^*) + \left(\mathcal{L}_j^w\right)'(\mu_j^*) \cdot (-u_j') + \frac{\alpha}{4}(u_j')^2$$

with probability at least $1 - \exp(-\Omega(\alpha n))$. Rearranging yields

$$|u_j'| \leqslant \frac{4}{\alpha} \cdot \left(\mathcal{L}_j^w\right)'(\mu_j^*) \,.$$

We next examine

$$\begin{aligned}
\left(\mathcal{L}_j^w\right)'(\mu_j^*) &= \sum_{i=1}^n w_i \phi\left((y_i)_j - \mu_j^*\right) \\
&= \sum_{i=1}^n w_i \phi(\eta_{ij}) + \sum_{i \in S_b} w_i \left[\phi\left((y_i)_j - \mu_j^*\right) - \phi\left((\boldsymbol{y}_i^*)_j - \mu_j^*\right)\right] \,.
\end{aligned}$$

By Hoeffding's inequality,

$$\left| \sum_{i=1}^n w_i \phi(\eta_{ij}) \right| \leqslant \left| \frac{1}{n} \sum_{i=1}^n \phi(\eta_{ij}) \right| + 2\rho\varepsilon \leqslant 2\rho\tau/\sqrt{n} + 4\rho\varepsilon$$

with probability at least $1 - \exp(-\tau^2/2)$.

The second term can be bounded as follows:

$$\sum_{i \in S_b} w_i \cdot \left[\phi\left(\mu_j^* - (y_i)_j\right) - \phi\left(\mu_j^* - (\boldsymbol{y}_i^*)_j\right)\right] \leqslant 4\rho\varepsilon \,,$$

where we used that for all $x, y \in \mathbb{R}$ it holds that $|\phi(x) - \phi(y)| \leqslant 2\rho$ and $\sum_{i \in S_b} w_i \leqslant 2\varepsilon$. Putting everything together we obtain

$$|u_j'| \leqslant \rho \frac{2\tau}{\alpha\sqrt{n}} + \frac{4\rho\varepsilon}{\alpha}$$

with probability $1 - \exp(-\tau^2/2)$. For $\tau = \alpha\sqrt{n}/10$, $|u_j'| < \rho$. Hence, $t = 1$ since otherwise

$$|u_j| = |\hat{\mu}_j(w) - \mu_j^*| = t \cdot |u_j'| < |u_j| \,.$$

It follows that $u_j' = u_j$ and thus $|\hat{\mu}_j(w) - \mu_j^*| < \rho$.

By Lemma F.1

$$\mathcal{L}^w \left( \mu^* \right) \geqslant \mathcal{L}^w \left( \hat{\mu} \right) \geqslant \mathcal{L}^w \left( \mu^* \right) + \left\langle \nabla \mathcal{L}^w \left( \mu^* \right), -u \right\rangle + \frac{\alpha}{4} \left\| u \right\|^2 ,$$

Hence

$$\left\langle \nabla \mathcal{L}^w \left( \mu^* \right), u \right\rangle \geqslant \frac{\alpha}{4} \left\| u \right\|^2 .$$

$\square$

In the following lemma we show that Assumption 2 of Assumption D.1 is satisfied for $\sigma = O(\rho)$ with probability $1 - \delta$ as long as $n \gtrsim \frac{d + \log(1/\delta)}{\alpha^2}$.

**Lemma F.3.** *With probability at least* $1 - \delta$,

$$\left\| \frac{1}{n} \sum_{i=1}^n f \left( \boldsymbol{\eta}_i \right) \right\| \leqslant 10 \rho \sqrt{\frac{d + \log \left( 1/\delta \right)}{n}} .$$

*Proof.* Let $u \in \mathbb{R}^d$ be some fixed vector. Since $\mathrm{sign} \left( \boldsymbol{\eta} \right)$ is independent of $\mathrm{abs} \left( \boldsymbol{\eta} \right)$, random variables $\tilde{\boldsymbol{u}}_{ij} = u_j \cdot \left| \boldsymbol{\eta}_{ij} \right|$ are independent of $\mathrm{sign} \left( \boldsymbol{\eta} \right)$. By Hoeffding's inequality

$$\left| \left\langle \frac{1}{n} \sum_{i=1}^n f \left( \boldsymbol{\eta}_i \right), u \right\rangle \right| = \left| \frac{1}{n} \sum_{i=1}^n \sum_{j=1}^d \mathrm{sign} \left( \boldsymbol{\eta}_{ij} \right) \tilde{\boldsymbol{u}}_{ij} \right| \leqslant 2\rho \cdot \left\| u \right\| \cdot \tau / \sqrt{n}$$

with probability at least $1 - \exp(-\tau^2/2)$. Let $\mathcal{N}_{1/2}$ be a $1/2$-net of size $6^d$ in the Euclidean unit $d$-dimensional ball. Then

$$\sup_{\|u\| \leqslant 1} \left| \left\langle \frac{1}{n} \sum_{i=1}^n f \left( \boldsymbol{\eta}_i \right), u \right\rangle \right| \leqslant \sup_{\|u\| \leqslant 1/2} \left| \left\langle \frac{1}{n} \sum_{i=1}^n f \left( \boldsymbol{\eta}_i \right), u \right\rangle \right| + \sup_{u \in \mathcal{N}_{1/2} 1/2} \left| \left\langle \frac{1}{n} \sum_{i=1}^n f \left( \boldsymbol{\eta}_i \right), u \right\rangle \right| .$$

Hence by union bound,

$$\left\| \frac{1}{n} \sum_{i=1}^n f \left( \boldsymbol{\eta}_i \right) \right\| = \sup_{\|u\| \leqslant 1} \left| \left\langle \frac{1}{n} \sum_{i=1}^n f \left( \boldsymbol{\eta}_i \right), u \right\rangle \right| \leqslant 2 \sup_{u \in \mathcal{N}_{1/2}} \left| \left\langle \frac{1}{n} \sum_{i=1}^n f \left( \boldsymbol{\eta}_i \right), u \right\rangle \right| \leqslant 4\rho \sqrt{\frac{2 \ln(6) \cdot d + \log \left( 1/\delta \right)}{n}} .$$

$\square$

In the following lemma we show that Assumption 3 of Assumption D.1 is satisfied for $\sigma = O(\rho)$ with probability $1 - \delta$ as long as $n \gtrsim d + \log(1/\delta)$.

**Lemma F.4.** *Let* $\delta \in (0, 1)$ *and suppose that* $n \geqslant d + \log \left( 1/\delta \right)$. *Then*

$$\left\| \frac{1}{n} \sum_{i=1}^n f \left( \boldsymbol{\eta}_i \right) \left( f \left( \boldsymbol{\eta}_i \right) \right)^\top \right\| \leqslant O \left( \rho^2 \right)$$

*with probability at least* $1 - \delta$.

*Proof.* Let $u \in \mathbb{R}^d$ be a fixed vector. Since the random vector $\mathrm{sign} \left( \boldsymbol{\eta} \right)$ is independent of $\mathrm{abs} \left( \boldsymbol{\eta} \right)$, the random vector $\mathrm{abs} \left( f \left( \boldsymbol{\eta} \right) \right) \circ u$ with entries $u_j \cdot \left| \boldsymbol{\eta}(j) \right|$ is independent of $\mathrm{sign} \left( \boldsymbol{\eta} \right)$. Since the entries of $\mathrm{sign} \left( \boldsymbol{\eta} \right)$ are iid Rademacher, for every positive integer $k$,

$$\begin{aligned}
\sup_{\|u\| \leqslant 1} \mathbb{E} \left\langle f \left( \boldsymbol{\eta} \right), u \right\rangle^{2k} &\leqslant \sup_{\|u\| \leqslant 1} \mathbb{E} \left\langle \mathrm{sign} \left( \boldsymbol{\eta} \right), \mathrm{abs} \left( f \left( \boldsymbol{\eta} \right) \right) \circ u \right\rangle^{2k} \\
&\leqslant \sup_{\|u\| \leqslant 2\rho} \mathbb{E} \left\langle \mathrm{sign} \left( \boldsymbol{\eta} \right), u \right\rangle^{2k} \\
&\leqslant \left( 2\rho \right)^{2k} \sup_{\|u\| \leqslant 1} \mathbb{E} \left\langle \mathrm{sign} \left( \boldsymbol{\eta} \right), u \right\rangle^{2k} \\
&\leqslant \left( 2\rho \right)^{2k} .
\end{aligned}$$

Hence $f \left( \boldsymbol{\eta} \right)$ is $2\rho$-subgaussian and the result follows from concentration of empirical covariance for subgaussian distributions (see Theorem 6.5 in [Wai19]). $\square$

In the following lemma we show that Assumption 3 of Assumption E.2 is satisfied for $\sigma = O(\rho)$ with probability $1 - \delta$ as long as $n \gtrsim d^k \cdot \log(d/\delta)$.

**Lemma F.5.** *Let $k$ be a positive integer and suppose that $n \geqslant 10 \cdot d^k \cdot \log(d/\delta)$. Then with probability at least $1 - \delta$, uniform distribution over the set $\{\boldsymbol{\eta}_1, \ldots, \boldsymbol{\eta}_n\}$ has $(2k, 2k)$-certifiable $4\rho$-bounded $f$-moments.*

*Proof.* First we show that the random vector $\boldsymbol{\eta}$ has $(2k, 2k)$-certifiable $2\rho$-bounded $f$-moments. Let's call a multi-index $\beta$ *even* if every element in $\beta$ appears even number of times. Note that $\beta$ is even iff $\beta = 2\beta'$ for a multi-index $\beta'$ of size $k$. Denote by $\mathcal{E}$ the set of all even multi-indices of size $2k$. Let $v_1, \ldots, v_d$ be variables, and consider the polynomial $\mathbb{E}\langle f(\boldsymbol{\eta}), v\rangle^{2k}$. It follows that

$$
\left|\frac{v}{2k}\right| \mathbb{E}\langle f(\boldsymbol{\eta}), v\rangle^{2k} = \mathbb{E}\,\mathbb{E}\left[\langle \operatorname{sign}(\boldsymbol{\eta}), \operatorname{abs}(f(\boldsymbol{\eta})) \circ v\rangle^{2k} \mid \operatorname{abs}(f(\boldsymbol{\eta}))\right]
$$

$$
= \mathbb{E}\sum_{\beta \in \mathcal{E}} \mathbb{E}\left[\operatorname{sign}(\boldsymbol{\eta})^\beta \left(\operatorname{abs}(f(\boldsymbol{\eta})) \circ v\right)^\beta \mid \operatorname{abs}(f(\boldsymbol{\eta}))\right]
$$

$$
= \mathbb{E}\sum_{\beta \in \mathcal{E}} \left(\operatorname{abs}(f(\boldsymbol{\eta})) \circ v\right)^\beta
$$

$$
\leqslant (2\rho)^{2k} \sum_{\beta \in \mathcal{E}} v^\beta
$$

$$
= (2\rho)^{2k} \sum_{|\beta'|=k} v^{2\beta'}
$$

$$
= (2\rho)^{2k} \|v\|^{2k}.
$$

Hence $\boldsymbol{\eta}$ has $(2k, 2k)$-certifiable $2\rho$-bounded $f$-moments. Since $\forall x \in \mathbb{R}^d$, $\|f(x)\| \leqslant 2\rho\sqrt{d}$, the result follows from Lemma J.4. $\qquad\square$

## G   Elliptical Distributions

In this section we use a different normalization than the one described in the introduction. Concretely, we assume that $\operatorname{Tr}(\Sigma) = d$ and that for some $\rho > 0$ and $\alpha \in (0, 1)$, $\mathbb{P}\left[\boldsymbol{R} \leqslant \rho\sqrt{d}\right] \geqslant \alpha$. For $\alpha = 1/2$, this is the same model as in the Introduction, since we can multiply $\Sigma^{1/2}$ by $s = \sqrt{d/\operatorname{Tr}(\Sigma)}$ and divide $\boldsymbol{R}$ by $s$. Hence the new parameter $\rho$ written in terms of the parameters defined in the Introduction is equal to $\sqrt{2}/s = \sqrt{2\operatorname{Tr}(\Sigma)/d}$. Note that $\rho\|\Sigma\|$ in new notation is equal to $\sqrt{2}\|\Sigma\|$ in the notation used in the introduction. Note that the effective rank, $\operatorname{r}(\Sigma) = \frac{\operatorname{Tr}\Sigma}{\|\Sigma\|}$, is the same in both parametrizations.

We assume that $\rho$ is known. In the case of unknown $\rho$ and $\alpha \geqslant \Omega(1)$, we can estimate $\rho$ using the corrupted samples and obtain the same guarantees as for known $\rho$ (up to a constant factor). We describe how to achiev this in Appendix G.1.

We prove that elliptical distributions satisfy Assumption D.1 and with, $\sigma = O(\rho\sqrt{\|\Sigma\|})$ and $F : \mathbb{R}^d \to \mathbb{R}$ defined as

$$
F(x) = \Phi_{20\rho\sqrt{d}}(\|x\|),
$$

with probability $1 - \delta$ as long as $n \gtrsim \frac{\operatorname{r}(\Sigma)\log(d/\delta)}{\alpha^4}$. Moreover, if in addition $n \gtrsim \frac{1}{\alpha^4}\left(\frac{\operatorname{r}(\Sigma)}{k}\right)^k \cdot \log(d/\delta)$, then Assumption E.2 is also satisfied with $\sigma = O(\rho\sqrt{k\|\Sigma\|})$.

Going back to the parametrization of the introduction and letting $\alpha = \frac{1}{2}$, this means the following: For $n \gtrsim \frac{\operatorname{r}(\Sigma)\log(d/\delta)}{\alpha^4}$, elliptical distributions satisfy Assumption D.1 with $\sigma = O(\sqrt{\|\Sigma\|})$. Further, if $n \gtrsim \frac{1}{\alpha^4}\left(\frac{\operatorname{r}(\Sigma)}{k}\right)^k \cdot \log(d/\delta)$, they satisfy Assumption E.2 with $\sigma = O(\sqrt{k\|\Sigma\|})$. Further, for $\boldsymbol{\eta}_1^*, \ldots, \boldsymbol{\eta}_n^*$ following an elliptical distribution with the above parameters, we will in Lemma G.4 show that for $n \geqslant \log(1/\delta)$ it holds with probability at least $1 - \delta$ that $\|\frac{1}{n}\sum_{i=1}^n f(\boldsymbol{\eta}_i^*)\| \leqslant O\left(\sqrt{\frac{\operatorname{Tr}(\Sigma)\cdot\log(d/\delta)}{n}}\right)$.

Hence, by Theorem E.3 (or Theorem D.2 for $k = 1$) our estimator achieves error

$$O\left(\left\|\frac{1}{n}\sum_{i=1}^{n} f\left(\boldsymbol{\eta}_i^*\right)\right\| + \varepsilon^{1-\frac{1}{2k}} \cdot \sigma\right) = O\left(\sqrt{\|\Sigma\|} \cdot \left[\sqrt{\frac{\mathrm{r}(\Sigma)\log(d/\delta)}{n}} + \sqrt{k}\varepsilon^{1-\frac{1}{2k}}\right]\right).$$

This gives the proof of Theorem 1.6.

Note that $f = \nabla F$ sends $x$ to $\phi_{20\rho\sqrt{d}}(\|x\|) \cdot \frac{x}{\|x\|}$ and hence is indeed 1-Lipschitz since it is the projection of $x$ onto the $\ell_2$-ball of radius $20\rho\sqrt{d}$. In the next three lemmas we prove the first assumption from Assumption D.1.

**Lemma G.1.** *Suppose that $d \geqslant 100$. Then*

$$\mathbb{P}\left[\left\|\boldsymbol{R}\Sigma^{1/2}\boldsymbol{u}\right\| \leqslant h/2\right] \geqslant 0.8\alpha,$$

*where $h = 20\rho\sqrt{d}$.*

*Proof.* Let $\boldsymbol{g} \sim N(0, 4\rho^2\Sigma)$ be a $d$-dimensional Gaussian vector such that $\boldsymbol{g} = \boldsymbol{R}'\Sigma^{1/2}\boldsymbol{u}$, where $\boldsymbol{R}'$ is independent of $\boldsymbol{R}$ and has distribution of the norm of Gaussian vector $N(0, 2\rho^2\mathrm{Id})$. Note that $\mathrm{Tr}\left(\mathbb{E}\,\boldsymbol{g}\boldsymbol{g}^\top\right) = 4\rho^2 d$. Applying Fact J.6 with $t = 2$, we get

$$\mathbb{P}\left[\|\boldsymbol{g}\| \geqslant 10\rho\sqrt{d}\right] \leqslant \exp(-2) \leqslant 0.15.$$

Since

$$\mathbb{P}\left[\boldsymbol{R}' \geqslant \rho\sqrt{d}\right] \geqslant 0.99,$$

we get

$$\mathbb{P}\left[\|\boldsymbol{g}\| \leqslant 10\rho\sqrt{d}, \boldsymbol{R}' \geqslant \rho\sqrt{d}\right] \geqslant 0.8.$$

Since $\boldsymbol{R}$ is independent of $\boldsymbol{g}$ and $\boldsymbol{R}'$, we get

$$\begin{aligned}
\mathbb{P}\left[\left\|\boldsymbol{R}\Sigma^{1/2}\boldsymbol{u}\right\| \leqslant 10\sqrt{d}\right] &\geqslant \mathbb{P}\left[\left\|\boldsymbol{R}\Sigma^{1/2}\boldsymbol{u}\right\| \leqslant \|\boldsymbol{g}\|, \|\boldsymbol{g}\| \leqslant 10\rho\sqrt{d}\right] \\
&= \mathbb{P}\left[\boldsymbol{R} \leqslant \boldsymbol{R}', \|\boldsymbol{g}\| \leqslant 10\rho\sqrt{d}\right] \\
&\geqslant \mathbb{P}\left[\boldsymbol{R} \leqslant \rho\sqrt{d}, \boldsymbol{R}' \geqslant \rho\sqrt{d}, \|\boldsymbol{g}\| \leqslant 10\rho\sqrt{d}\right] \\
&\geqslant 0.8\alpha.
\end{aligned}$$

$\square$

**Lemma G.2.** *Let $w \in \mathcal{W}_{2\varepsilon}$ for $\varepsilon \lesssim \alpha$. Then with probability $1 - \exp\left(-\Omega\left(\alpha n\right)\right)$ for all $u \in \mathbb{R}^d$ such that $\|u\| \leqslant \rho\sqrt{d}$,*

$$\mathcal{L}^w\left(\mu^* + u\right) - \mathcal{L}^w\left(\mu^*\right) - \left\langle\nabla\mathcal{L}^w\left(\mu^*\right), u\right\rangle \geqslant \frac{\alpha}{4} \cdot \|u\|^2.$$

*Proof.* Let $\zeta_i = \mu - y_i$.

$$\mathcal{L}^w\left(\mu^* + u\right) - \mathcal{L}^w\left(\mu^*\right) - \left\langle\mathcal{L}^w\left(\mu^*\right), u\right\rangle = \sum_{i=1}^{n} w_i\left[f\left(\|\zeta_i + u\|\right) - f\left(\|\zeta_i\|\right) - \left\langle f\left(\zeta_i\right), u\right\rangle\right]$$

Let $M$ be the set of uncorrupted samples such that $\|\zeta_i\| = \|\boldsymbol{\eta}_i\| \leqslant h/2$. Note that for $i \in M$, $f\left(\|\zeta_i + u\|\right) = \frac{1}{2}\|\zeta_i + u\|^2$ and $f\left(\|\zeta_i\|\right) = \frac{1}{2}\|\zeta_i\|^2$. Hence

$$\sum_{i \in M} w_i\left[f\left(\|\zeta_i + u\|\right) - f\left(\|\zeta_i\|\right) - \left\langle f\left(\zeta_i\right), u\right\rangle\right] = \frac{1}{2}\sum_{i \in M} w_i\|u\|^2.$$

And by convexity,

$$\sum_{i \notin M} w_i\left[f\left(\|\zeta_i + u\|\right) - f\left(\|\zeta_i\|\right) - \left\langle f\left(\zeta_i\right), u\right\rangle\right] \geqslant 0.$$

Since $w \in \mathcal{W}_{2\varepsilon}$,

$$\sum_{i \in M} \left| w_i - \frac{1}{n} \right| \leqslant 2\varepsilon \,.$$

Hence

$$\sum_{i \in M} w_i \geqslant \frac{|M|}{n} - 2\varepsilon \,.$$

By Lemma G.1 and a Chernoff bound,

$$|M| \geqslant 0.6\alpha - \varepsilon \geqslant \alpha/2$$

with probability $1 - \exp(-\alpha n)$. $\qquad\square$

**Lemma G.3.** *Let $w \in \mathcal{W}_{2\varepsilon}$ for $\varepsilon \lesssim \alpha$, and suppose that $n \gtrsim \log(d)/\alpha^2$. Then with probability $1 - \exp\left(-\Omega\left(\alpha^2 n\right)\right)$,*

$$\langle \nabla \mathcal{L}^w(\mu^*), \mu^* - \hat{\mu}^w \rangle \geqslant \frac{\alpha}{4} \|\hat{\mu}^w - \mu^*\|^2 \,.$$

*Proof.* Let $u = \mu^* - \hat{\mu}(w)$. Let $u'$ and $t$ be such that

$$\|u'\| \leqslant \rho\sqrt{d} \,,$$
$$\hat{\mu}(w) = \mu^* + t \cdot u' \,.$$

It follows that $t = \max\{1, \|u\|\}$. Clearly, $\hat{\mu}(w)$ is the (unique) minimizer of $\mathcal{L}^w$. Let $\hat{\mu}' = \mu^* + u'$, by convexity of $\mathcal{L}^w$ it holds that $\mathcal{L}^w(\hat{\mu}') \leqslant \mathcal{L}^w(\mu^*)$. Since $\|u'\| \leqslant \rho\sqrt{d}$ it follows by Lemma G.2 that

$$\mathcal{L}^w(\mu^*) \geqslant \mathcal{L}^w(\hat{\mu}') \geqslant \mathcal{L}^w(\mu^*) - \langle \nabla \mathcal{L}^w(\mu^*), u \rangle + \frac{\alpha}{4}\|u'\|^2$$

with probability at least $1 - \exp(-\Omega(\alpha n))$. Rearranging yields

$$\|u'\| \leqslant \frac{4}{\alpha} \cdot \|\nabla \mathcal{L}^w(\mu^*)\| \,.$$

We next examine

$$\begin{aligned}
\nabla \mathcal{L}^w(\mu^*) &= \sum_{i=1}^{n} w_i f((y_i) - \mu^*) \\
&= \sum_{i=1}^{n} w_i f(\boldsymbol{\eta}_i) + \sum_{i \in S_b} w_i \left[ f((y_i) - \mu^*) - f((\boldsymbol{y}_i^*) - \mu^*) \right] \,.
\end{aligned}$$

By vector Bernstein inequality Fact J.2,

$$\left\| \sum_{i=1}^{n} w_i f(\boldsymbol{\eta}_i) \right\| \leqslant \left\| \frac{1}{n} \sum_{i=1}^{n} f(\boldsymbol{\eta}_i) \right\| + 40\varepsilon\rho\sqrt{d} \leqslant 200\rho\sqrt{d} \cdot (\tau/\sqrt{n} + \tau^2/n) + 40\varepsilon\rho\sqrt{d}$$

with probability at least $1 - \exp(-\tau^2/2)$.

The second term can be bounded as follows:

$$\sum_{i \in S_b} w_i \cdot \left[ f(\mu^* - (y_i)) - f(\mu^* - (\boldsymbol{y}_i^*)) \right] \leqslant 40\varepsilon\rho\sqrt{d} \,,$$

where we used that for all $x, y \in \mathbb{R}$ it holds that $|f(x) - f(y)| \leqslant 20\rho\sqrt{d}$ and $\sum_{i \in S_b} w_i \leqslant 2\varepsilon$. Putting everything together we obtain

$$\|u'\| \leqslant 200\frac{\rho\sqrt{d}}{\alpha} \cdot (\tau/\sqrt{n} + \tau^2/n) + \frac{40\varepsilon\rho\sqrt{d}}{\alpha}$$

with probability $1 - \exp(-\tau^2/2)$. For $\tau = \alpha\sqrt{n}/1000$, $|u'| < \rho$. Hence, $t = 1$ since otherwise

$$\|u\| = \|\hat{\mu}(w) - \mu^*\| = t \cdot \|u'\| < \|u\| \,.$$

It follows that $u' = u$ and thus $\|\hat{\mu}(w) - \mu^*\| < \rho\sqrt{d}$.

By Lemma G.2

$$\mathcal{L}^w (\mu^*) \geqslant \mathcal{L}^w (\hat{\mu}) \geqslant \mathcal{L}^w (\mu^*) + \langle \nabla\mathcal{L}^w (\mu^*), -u \rangle + \frac{\alpha}{4} \|u\|^2 ,$$

Hence

$$\langle \nabla\mathcal{L}^w (\mu^*), u \rangle \geqslant \frac{\alpha}{4} \|u\|^2 .$$

$\square$

In the following lemma we show that Assumption 2 of Assumption D.1 is satisfied for $\sigma = O(\rho\|\Sigma\|)$ with probability $1 - \delta$ as long as $n \gtrsim \frac{d\log(d/\delta)}{\alpha^2}$.

**Lemma G.4.** *Let $\delta \in (0, 1)$ and suppose that $n \geqslant \log(1/\delta)$. Then with probability at least $1 - \delta$,*

$$\left\| \frac{1}{n} \sum_{i=1}^n f(\boldsymbol{\eta}_i) \right\| \leqslant O\left( \rho\sqrt{\frac{d\log(d/\delta)}{n}} \right) .$$

*Proof.* The lemma is a direct consequence of vector Bernstein inequality Fact J.2. $\square$

In the following lemma we show that Assumption 3 of Assumption D.1 is satisfied for $\sigma = O(\rho)$ with probability $1 - \delta$ as long as $n \gtrsim d\log(d/\delta)$.

**Lemma G.5.** *Let $\delta \in (0, 1)$ and suppose that $n \geqslant r(\Sigma)\log(d/\delta)$. Then*

$$\left\| \frac{1}{n} \sum_{i=1}^n f(\boldsymbol{\eta}_i) (f(\boldsymbol{\eta}_i))^\top \right\| \leqslant O\left( \rho^2 \|\Sigma\| \right)$$

*with probability at least $1 - \delta$.*

*Proof.* Let $s(\boldsymbol{\eta}) = \frac{h}{\|\boldsymbol{\eta}\|}\boldsymbol{\eta} = \frac{h}{\|f(\boldsymbol{\eta})\|}f(\boldsymbol{\eta})$, where $h = 20\rho\sqrt{d}$. Note that $\|s(\boldsymbol{\eta})\| \geqslant \|f(\boldsymbol{\eta})\|$. By Fact J.5 and since by our parametrization $\operatorname{Tr}\Sigma = d$, it follows that

$$\sup_{\|u\|\leqslant 1} \mathbb{E} \langle f(\boldsymbol{\eta}), u \rangle^2 \leqslant \sup_{\|u\|\leqslant 1} \mathbb{E} \langle s(\boldsymbol{\eta}), u \rangle^2 \leqslant (20\rho)^2 \|\Sigma\| .$$

Let $t = \rho^2\|\Sigma\| \cdot \sqrt{\frac{r(\Sigma)\log(d/\delta)}{n}} \leqslant \rho^2\|\Sigma\|$. It follows by Fact J.3 that

$$\left\| \frac{1}{n} \sum_{i=1}^n f(\boldsymbol{\eta}_i) (f(\boldsymbol{\eta}_i))^\top \right\| \leqslant O\left( \rho^2 \|\Sigma\| + t \right) \leqslant O\left( \rho^2 \|\Sigma\| \right) ,$$

with probability at least $1 - \delta$, where we also used that $t = \rho^2\|\Sigma\| \cdot \sqrt{\frac{r(\Sigma)\log(d/\delta)}{n}} \leqslant \rho^2\|\Sigma\|$ when applying Fact J.3. $\square$

In the following lemma we show that Assumption 3 of Assumption E.2 is satisfied for $\sigma = O(\rho\|\Sigma\|)$ with probability $1 - \delta$ as long as $n \gtrsim r(\Sigma)^k\log(d/\delta)$.

**Lemma G.6.** *Let $k$ be a positive integer and suppose that $n \geqslant 10 \cdot r(\Sigma)^k \cdot \log(d/\delta)$ and*

$$\|\Sigma\|_F \leqslant \frac{\operatorname{Tr}(\Sigma)}{10\sqrt{k\log d}} .$$

*Then with probability at least $1 - \delta$, uniform distribution over the set $\{\boldsymbol{\eta}_1, \ldots, \boldsymbol{\eta}_n\}$ has $(2k, 2k)$-certifiable $O\left( \rho\sqrt{k\|\Sigma\|} \right)$-bounded $f$-moments.*

*Proof.* By Lemma K.4, $\boldsymbol{\eta}$ has $(2k, 2k)$-certifiable $O\left( \rho\sqrt{k\|\Sigma\|} \right)$-bounded $f$-moments. Since $\forall x \in \mathbb{R}^d$, $\|f(x)\| \leqslant 20\rho\sqrt{d}$ the result follows from Lemma J.4. $\square$

## G.1 Unknown $\rho$

If $\rho$ is not known and if $\alpha = 0.01$ (or arbitrary constant), we can first divide the $n$ samples into 2 subsamples of size at least $n' = \lfloor n/2 \rfloor$, and use the first half to learn $\rho$, and the second half for mean estimation.

To do this, let us take the first $n'$ samples and again divide them into two subsamples of size at least $n'' = \lfloor n'/2 \rfloor$, and subtract the second half from the first half (if $n'$ is odd, let us ignore the last sample). Then we get $n''$ ($\varepsilon$-corrupted) iid copies $\boldsymbol{\xi}_1, \ldots \boldsymbol{\xi}_{n''}$ of an elliptically distributed vector with location 0 and the scatter matrix $\Sigma$ as the original distribution. Moreover, $\boldsymbol{R}$ for this distribution satisfies $\mathbb{P}\left[\boldsymbol{R} \leqslant 2\rho\sqrt{d}\right] \geqslant \alpha^2$. Let us compute and sort the norms of the samples, and let us take the maximal value $\boldsymbol{b}$ among the smallest $\alpha^2/2$ norms. If $\varepsilon \leqslant 0.01\alpha$, then by Chernoff bound with probability at least $1 - \exp(-\Omega(\alpha n))$, $\boldsymbol{b} \leqslant 2\rho\sqrt{d}$ and $\mathbb{P}\left[\boldsymbol{R} \leqslant \boldsymbol{b} \mid \boldsymbol{b}\right] \geqslant \alpha^2/4$.

Now, we take the second $n'$ samples, divide them into two subsamples of size at least $n'' = \lfloor n'/2 \rfloor$, and add the second half to the first half (if $n'$ is odd, let us ignore the last sample). Since the initial noise distribution is symmetric about zero, we get ($\varepsilon$-corrupted) iid samples $2\mu + \boldsymbol{\zeta}_1, \ldots, 2\mu + \boldsymbol{\zeta}_{n''}$, where $\boldsymbol{\zeta}_i$ have the same distribution as $\boldsymbol{\xi}_i$ described above. Hence we can use $h = 20\boldsymbol{b}$ in our proofs and get the same guarantees as if we used $h = 20\rho\sqrt{d}$ (up to a constant factor $O(1/\alpha)$).

# H   Proof of Identifiability for Near-Optimal Error

In this section, we will prove the statement of identifiability underlying Theorem 1.4. We first introduce some notation. We use similar notation as in [Li19]. Let $D$ be an $(\alpha, \rho)$-semi-product distribution with location $\mu^* \in \mathbb{R}^d$. Let $\boldsymbol{y}_1^*, \ldots, \boldsymbol{y}_n^*$ be $n$ i.i.d.samples from $D$ and let $y_1, \ldots, y_n$ be an $\varepsilon$ corruption thereof. Further, let $S_g = \{i \mid y_i = \boldsymbol{y}_i^*\}$ be the set of uncorrupted indices and $S_b = [n] \setminus S_g$ be the set of corrupted ones. Let

$$\mathcal{W}_\varepsilon = \left\{ w \in \mathbb{R}^n \mid 0 \leqslant w \leqslant \tfrac{1}{n}\mathbb{1}, \left\| w - \tfrac{1}{n}\mathbb{1} \right\|_1 \leqslant \varepsilon \right\} ,$$

where $\mathbb{1}$ is the all-ones vector and the inequalities between vectors are entry-wise. Throughout the note we assume that $\varepsilon \lesssim \alpha$. Note, that otherwise faithfully recovering $\mu^*$ is impossible: For each coordinate, with high probability there is only an $\Theta(\alpha)$-fraction of the samples where the noise is small. If $\varepsilon \gtrsim \alpha$, the adversary could corrupt precisely this fraction and erase all information about this coordinate.

Further, for $h > 0$, let $F_h \colon \mathbb{R}^d \to \mathbb{R}$ be defined as $x \mapsto F_h(x) = \sum_{j=1}^d \Phi_h(x_j)$, where $\Phi_h$ is the Huber loss

$$\Phi(t) = \begin{cases} \frac{1}{2}t^2 , & \text{if } |t| \leqslant h , \\ h \cdot (|x| - h) , & \text{otherwise.} \end{cases}$$

and denote by $f = \nabla F$ its gradient. Note that in our case $f$ is the derivative of the Huberloss applied entry-wise. The derivative of the Huber loss is $\phi \colon \mathbb{R} \to \mathbb{R}$,

$$x \mapsto \begin{cases} t , & \text{if } |t| \leqslant h , \\ h \cdot \text{sign}(t) , & \text{otherwise.} \end{cases}$$

For $w \in \mathbb{R}^n$ with non-negative entries, we define $\mathcal{L}^w \colon \mathbb{R}^d \to \mathbb{R}$ as

$$\mathcal{L}^w(\mu) := \sum_{i=1}^n w_i F(\mu - y_i) .$$

It follows that $\nabla \mathcal{L}^w(\mu) = \sum_{i=1}^n w_i f(\mu - y_i)$. We let

$$\hat{\mu}^w = \arg\min_{\mu \in \mathbb{R}^d} \mathcal{L}^w(\mu) .$$

and $\hat{\Sigma}_f^w = \sum_{i=1}^n w_i f(y_i - \hat{\mu}^w) f(y_i - \hat{\mu}^w)^\top$.

Similar as in Appendix G.1, by looking at differences of pairs of samples, we can assume that we have access to the ($\varepsilon$-corrupted) samples from the noise distribution, by replacing $\alpha$ by $\alpha^2$ in the definition of the semi-product distribution. This allows us to compute the following relevant

parameters: Note that $\mathbb{E}_{\boldsymbol{\eta} \sim D} f(\boldsymbol{\eta}^*) f(\boldsymbol{\eta}^*)^\top$ is diagonal. We can estimate each diagonal entry $\sigma_j^2$ up to a factor $(1 \pm O(\varepsilon + \sqrt{\log(1/\delta)/n}))$ with probability $1/\delta$ by using 1-dimensional robust mean estimation. We will assume that $\mathbb{E}_{\boldsymbol{\eta} \sim D} f(\boldsymbol{\eta}^*) f(\boldsymbol{\eta}^*)^\top$ is known, the error $O(\varepsilon)$ does not affect our results, since our error bounds will be larger than this. Similarly, when $\alpha$ is known, we can also compute $\rho$ as in Appendix G.1.

**Identifiability**   Our aim will be to prove the following theorem

**Theorem H.1.** *Let $w \in \mathcal{W}_\varepsilon$, $\delta \in (0,1)$, and assume that $\varepsilon \leqslant \alpha/100$, $\rho = 1$ and $n \geqslant \frac{d \log(d)}{\alpha^4} + \log(1/\delta)$. In the above let $h = 2$. Then with probability at least $1 - \delta$ it holds that*

$$\|\hat{\mu}^w - \mu^*\| \leqslant O\left(\sqrt{\frac{d \log(d) + \log(1/\delta)}{\alpha^2 n}} + \frac{1}{\alpha^{3/2}} \cdot \sqrt{\varepsilon \left(\left\|\Sigma_f^w - \mathbb{E} f(\boldsymbol{\eta}^*) f(\boldsymbol{\eta}^*)^\top\right\| + \varepsilon \log(1/\varepsilon)\right)}\right).$$

In our proof, we will use the following properties analogous to *stability-conditions* used before in the literature (cf. for example [DK22]).

**Lemma H.2.** *Suppose that the conditions of Theorem H.1 hold. Then with probability at least $1 - \delta$ the following events hold*

*1.* $\|\sum_{i=1}^n w_i f(\boldsymbol{\eta}_i^*)\| \leqslant O\left(\varepsilon \sqrt{\log(1/\varepsilon)} + \sqrt{\frac{d + \log(1/\delta)}{n}}\right)$,

*2.* $\left\|\sum_{i \in S_g} w_i f(\boldsymbol{\eta}^*) f(\boldsymbol{\eta}^*)^\top - \mathbb{E} f(\boldsymbol{\eta}_i^*) f(\boldsymbol{\eta}_i^*)^\top\right\| \leqslant O\left(\varepsilon \log(1/\varepsilon) + \sqrt{\frac{d + \log(1/\delta)}{n}}\right)$,

*3.* $\sum_{i \in S_g} w_i f(\hat{\mu}^w - y_i^*) f(\hat{\mu}^w - y_i^*)^\top \succeq \mathbb{E} f(\boldsymbol{\eta}^*) f(\boldsymbol{\eta}^*)^\top - O\left(\varepsilon \log(1/\varepsilon)/\alpha - \sqrt{\frac{d \log d + \log(1/\delta)}{n}}\right) \cdot \mathrm{Id}_d$.

*Proof.* By Lemma F.4, $f(\boldsymbol{\eta}^*)$ is $O(1)$-sub-Gaussian. Hence the first two inequalities follow from Fact J.7.

Let us prove the third one. First, let $\Delta \in [-10\varepsilon/\alpha, 10\varepsilon/\alpha]^d$ be an arbitrary fixed (non-random) vector. Let $\boldsymbol{A} = \{|\boldsymbol{\eta}_{ij}| \mid i \in [n], j \in [d]\}$. Consider the random matrix

$$\boldsymbol{M}(i) = f(\boldsymbol{\eta}_i^* + \Delta) f(\boldsymbol{\eta}_i^* + \Delta)^\top - \mathbb{E}[f(\boldsymbol{\eta}_i^* + \Delta) \mid \boldsymbol{A}] \mathbb{E}[f(\boldsymbol{\eta}_i^* + \Delta) \mid \boldsymbol{A}]^\top$$

Since for $l, j \in [d]$ such that $l \neq j$, $\mathrm{sign}(\boldsymbol{\eta}_{ij})$ and $\mathrm{sign}(\boldsymbol{\eta}_{il})$ are independent given $\boldsymbol{A}$, $\mathbb{E}[\boldsymbol{M}(i)_{lj} \mid \boldsymbol{A}] = 0$. Note that for all $j \in [d]$, $\mathbb{E}[\boldsymbol{M}(i)_{jj} \mid \boldsymbol{A}]$ is equal to

$$\mathbb{E}\left[f\left((\boldsymbol{\eta}_i^*)_j\right)^2 - f\left((\boldsymbol{\eta}_i^*)_j\right)\left[f\left((\boldsymbol{\eta}^*)_j + \Delta_j\right) - f\left((\boldsymbol{\eta}_i^*)_j\right)\right] + \left[f\left((\boldsymbol{\eta}^*)_j + \Delta_j\right) - f\left((\boldsymbol{\eta}_i^*)_j\right)\right]^2 \,\middle|\, \boldsymbol{A}\right].$$

Since $f$ is 1-Lipschitz and is bounded by 2,

$$\left|\mathbb{E}\left[f\left((\boldsymbol{\eta}_i^*)_j\right)\left(f\left((\boldsymbol{\eta}_i^*)_j + \Delta_j\right) - f\left((\boldsymbol{\eta}_i^*)_j\right)\right) \,\middle|\, \boldsymbol{A}\right]\right| \leqslant 2|\Delta_j| \leqslant 20\varepsilon/\alpha.$$

Hence we get

$$\mathbb{E}[\boldsymbol{M}(i) \mid \boldsymbol{A}] \succeq \mathbb{E}\left[f(\boldsymbol{\eta}_i^*) f(\boldsymbol{\eta}_i^*)^\top \,\middle|\, \boldsymbol{A}\right] - O(\varepsilon/\alpha) \cdot \mathrm{Id}_d.$$

Denote $\boldsymbol{D} = \frac{1}{n} \sum_{i=1}^n \mathbb{E}\left[f(\boldsymbol{\eta}_i^*) f(\boldsymbol{\eta}_i^*)^\top \,\middle|\, \boldsymbol{A}\right]$. Since the entries of $\mathrm{sign}(\boldsymbol{\eta}^*)$ are mutually independent given $\boldsymbol{A}$, the entries of $f(\boldsymbol{\eta}^* + \Delta)$ are also mutually independent given $\boldsymbol{A}$. Since the entries are bounded by $O(1)$, by Lemma J.8, for all $\delta \in (0,1)$,

$$\mathbb{P}\left[\sum_{i=1}^n w_i' f(\boldsymbol{\eta}_i^* + \Delta) f(\boldsymbol{\eta}_i^* + \Delta)^\top \succeq \boldsymbol{D} - O\left(\frac{\varepsilon}{\alpha} + \varepsilon \log(1/\varepsilon) + \sqrt{\frac{d + \log\left(\frac{1}{\delta}\right)}{n}}\right) \cdot \mathrm{Id}_d \,\middle|\, \boldsymbol{A}\right] \geqslant 1 - \delta/10,$$

where $w_i' = w_i \mathbf{1}_{[i \in S_g]}$. Taking expectations from both sides, we get

$$\mathbb{P}\left[\sum_{i=1}^{n} w_i' f\left(\boldsymbol{\eta}_i^* + \Delta\right) f\left(\boldsymbol{\eta}_i^* + \Delta\right)^\top \succeq \boldsymbol{D} - O\left(\frac{\varepsilon}{\alpha} + \varepsilon \log(1/\varepsilon) + \sqrt{\frac{d + \log\left(\frac{1}{\delta}\right)}{n}}\right) \cdot \mathrm{Id}_d\right] \geqslant 1 - \delta/10 \,.$$

Let $\mathcal{N}_{\varepsilon'}$ be an $\varepsilon'$-net in $[-10\varepsilon/\alpha, 10\varepsilon/\alpha]^d$ of size at most $\left(\frac{20\varepsilon}{\varepsilon'\alpha}\right)^d$. Since $f$ is 1-Lipschitz and is bounded by 1,

$$\left\|f\left(\boldsymbol{\eta}_i^* + \Delta\right) w_i f\left(\boldsymbol{\eta}_i^* + \Delta\right)^\top - f\left(\boldsymbol{\eta}_i^* + \Delta'\right) f\left(\boldsymbol{\eta}_i^* + \Delta'\right)^\top\right\|_{\max} \leqslant 2\left\|\Delta - \Delta'\right\|_{\max} \,.$$

SInce the spectral norm can be only by at most $\sqrt{d}$ factor larger than the infinity norm, by taking $\varepsilon' = \varepsilon\sqrt{d}/(100\alpha)$ and using a union bound, we get that with probability $1 - \delta/10$, for all $\Delta \in [-10\varepsilon/\alpha, 10\varepsilon/\alpha]^d$ (including $\hat{\mu}^w$),

$$\sum_{i=1}^{n} w_i' f\left(\boldsymbol{\eta}_i^* + \Delta\right) f\left(\boldsymbol{\eta}_i^* + \Delta\right)^\top \succeq \boldsymbol{D} - O\left(\frac{\varepsilon}{\alpha} + \sqrt{\frac{d\log d + \log\left(\frac{1}{\delta}\right)}{n}} + \frac{d\log d + \log\left(\frac{1}{\delta}\right)}{n}\right) \cdot \mathrm{Id}_d \,.$$

Note that $\boldsymbol{D}$ is a diagonal matrix, and its diagonal entries are

$$\boldsymbol{D}_{jj} = \frac{1}{n}\sum_{i=1}^{n} f\left(\boldsymbol{\eta}_{ij}^*\right)^2 \,.$$

Since $f(\boldsymbol{\eta}_j^*)$ are $O(1)$-sub-Gaussian, with probability at least $1 - \delta/10$,

$$\left\|\boldsymbol{D} - \mathbb{E}\, f\left(\boldsymbol{\eta}^*\right) f\left(\boldsymbol{\eta}^*\right)^\top\right\| \leqslant O\left(\sqrt{\frac{\log(d/\delta)}{n}}\right) \,.$$

Hence we get the desired bound. $\qquad\square$

*Proof of Theorem H.1.* By Lemma F.1, $\frac{\alpha}{4}\left\|\hat{\mu}^w - \mu^*\right\|^2 \leqslant \langle\nabla\mathcal{L}^w(\mu^*), \mu^* - \hat{\mu}^w\rangle$. Hence, it is enough to bound $\langle\nabla\mathcal{L}^w(\mu^*), \mu^* - \hat{\mu}^w\rangle$. Let $u = \hat{\mu}^w - \mu^*$. Observe that using the Cauchy-Schwarz Inequality, it follows that

$$\begin{aligned}
\langle\nabla\mathcal{L}^w(\mu^*), u\rangle &= \sum_{i=1}^{n} w_i \langle f(\mu^* - y_i), u\rangle \\
&= \sum_{i=1}^{n} w_i \langle f(\mu^* - \boldsymbol{y}_i^*), u\rangle + \sum_{i \in S_b} w_i \langle f(\mu^* - y_i) - f(\mu^* - \boldsymbol{y}_i^*), u\rangle \\
&\leqslant \left\|\sum_{i=1}^{n} w_i f(\mu^* - \boldsymbol{y}_i^*)\right\| \cdot \|u\| + \sqrt{\varepsilon} \cdot \sqrt{\sum_{i \in S_b} w_i \langle f(\mu^* - y_i) - f(\mu^* - \boldsymbol{y}_i^*), u\rangle^2} \,.
\end{aligned}$$

Note that by the first point of Lemma H.2 it holds that $\left\|\sum_{i=1}^{n} w_i f(\mu^* - \boldsymbol{y}_i^*)\right\| \leqslant O(\varepsilon\sqrt{\log(1/\varepsilon)} + \sqrt{\frac{d+\log(1/\delta)}{n}})$. For the second term, we can bound

$$\sum_{i \in S_b} w_i \langle f(\mu^* - y_i) - f(\mu^* - \boldsymbol{y}_i^*), u\rangle^2$$
$$\lesssim \underbrace{\sum_{i \in S_b} w_i \langle f(\mu^* - \boldsymbol{y}_i^*), u\rangle^2}_{\text{Term A}} + \underbrace{\sum_{i \in S_b} w_i \langle f(\hat{\mu}^w - y_i), u\rangle^2}_{\text{Term B}} + \underbrace{\sum_{i \in S_b} w_i \langle f(\mu^* - y_i) - f(\hat{\mu}^w - y_i), u\rangle^2}_{\text{Term C}} \,.$$

Term A is at most $O\left(\varepsilon \log(1/\varepsilon) + \sqrt{\frac{d+\log(1/\delta)}{n}}\right) \cdot \|u\|^2$ by combining the second point of Lemma H.2 with Fact J.7. To bound Term C, we use that $f$ is 1-Lipschitz to observe

$$\sum_{i \in S_b} w_i \langle f(\mu^* - y_i) - f(\hat{\mu}^w - y_i), u\rangle^2$$

$$\leqslant \|u\|^2 \cdot \sum_{i \in S_b} w_i \|f(\mu^* - y_i) - f(\hat{\mu}^w - y_i^*)\|^2 \leqslant \|u\|^2 \cdot \sum_{i \in S_b} w_i \|\mu^* - \hat{\mu}^w\|^2 \leqslant \varepsilon \cdot \|u\|^4 \ .$$

It remains to bound Term B. To this end, we use the third point of Lemma H.2 and obtain

$$\sum_{i \in S_b} w_i \langle f(\hat{\mu}^w - y_i), u \rangle^2 = \sum_{i=1}^n w_i \langle f(\hat{\mu}^w - y_i), u \rangle^2 - \sum_{i \in S_g} w_i \langle f(\hat{\mu}^w - \boldsymbol{y}_i^*), u \rangle^2$$

$$\leqslant \left( \left\| \hat{\Sigma}_f - \mathbb{E} f(\boldsymbol{\eta}^*) f(\boldsymbol{\eta}^*)^\top \right\| + O\left( \varepsilon \log(1/\varepsilon)/\alpha \right) + \sqrt{\frac{d \log d + \log\left(\frac{1}{\delta}\right)}{n}} \right) \cdot \|u\|^2 \ .$$

Putting the above bounds together, we have shown that

$$\langle \nabla \mathcal{L}^w(\mu^*), u \rangle \leqslant \|u\| \cdot O\left( \frac{\varepsilon \sqrt{\log(1/\varepsilon)}}{\sqrt{\alpha}} + \sqrt{\frac{d \log(d) + \log(1/\delta)}{n}} + \varepsilon \|u\| + \sqrt{\varepsilon \left\| \hat{\Sigma}_f - \mathbb{E} f(\boldsymbol{\eta}^*) f(\boldsymbol{\eta}^*)^\top \right\|} \right) \ .$$

(Note that we also use $\sqrt{\varepsilon} \cdot \sqrt{\varepsilon \log(1/\varepsilon) + \sqrt{\frac{d + \log(1/\delta)}{n}}} \leqslant \varepsilon \sqrt{\log(1/\varepsilon)} + \sqrt{\frac{d + \log(1/\delta)}{n}}$.) Since $\langle \nabla \mathcal{L}^w(\mu^*), u \rangle \geqslant \frac{\alpha}{4} \|u\|^2$ and $\varepsilon \lesssim \alpha$, rearranging and dividing by $\|u\|$ yields that

$$\|u\| \leqslant O\left( \sqrt{\frac{d \log(d) + \log(1/\delta)}{\alpha^2 n}} + \frac{1}{\alpha^{3/2}} \cdot \sqrt{\varepsilon \left( \left\| \hat{\Sigma}_f - \mathbb{E} f(\boldsymbol{\eta}^*) f(\boldsymbol{\eta}^*)^\top \right\| + \varepsilon \log(1/\varepsilon) \right)} \right) \ .$$

$\square$

# I  Filtering Algorithm for Near-Optimal Error

In this section, we will use the proof of identifiability from Appendix H to obtain an algorithm which efficiently recovers $\mu^*$ up to (nearly) optimal error. In particular, we show that

**Theorem I.1.** *Let $C > 0$ be a large enough absolute constant. Let $\varepsilon, \alpha > 0$ be such that $\sqrt[3]{\varepsilon} \leqslant \frac{\alpha}{C}$. Let $D$ be an $(\alpha, \rho)$-product distribution with location $\mu^* \in \mathbb{R}^d$. Let $n \geqslant C \cdot \frac{d \log(d) + \log(1/\delta)}{\alpha^6 \varepsilon^2 \log(1/\varepsilon)}$. There exists an algorithm running im time $n^{O(1)} d^{O(1)}$, such that given an $\varepsilon$-corrupted sample of $D$ of size $n$ and $\rho$[9], with probability at least $1 - \delta$ it outputs $\hat{\mu}$ satisfying*

$$\|\mu^* - \hat{\mu}\| \leqslant O\left( \rho \cdot \left[ \sqrt{\frac{d \log(d) + \log(1/\delta)}{\alpha^4 n}} + \frac{\varepsilon \sqrt{\log(1/\varepsilon)}}{\alpha^3} \right] \right) \ .$$

The algorithm we propose is based on the by-now-standard filtering approach. We closely follow the exposition in [Li19]. To describe the algorithm, let $C > 0$ be some universal constant and for $\mu \in \mathbb{R}^d, w \in \mathcal{W}_\varepsilon$ let

$$\hat{\mu}(w) = \min_{\mu \in \mathbb{R}^d} \mathcal{L}^w(\mu) \ , \qquad\qquad \Sigma_f(w) = \sum_{i=1}^n w_i f(\hat{\mu}(w) - y_i) f(\hat{\mu}(w) - y_i)^\top \ .$$

Note that since we assume $\rho$ to be known, we can without loss of generality assume that $\rho = 1$ by scaling. Our algorithm is the following.

---

[9]We remark that in the case of unknown $\rho$ but known $\alpha$, we can first estimate $\rho$ from the corrupted samples similar as in Appendix G.1, at the cost of a worse dependence on $\alpha$.

---

**Algorithm I.2** (Filtering Algorithm).
**Input:** $\varepsilon$-corrupted sample $y_1, \ldots, y_n$ and $\rho, \gamma > 0$.
**Output:** Location estimate $\hat{\mu}$.

- Let $w^{(0)} = \frac{1}{n} \mathbb{1}_n$.

- Compute $\hat{\mu}^{(0)} = \min_{\mu \in \mathbb{R}^d} \mathcal{L}^{w^{(0)}}(\mu)$ and $\Sigma_f^{(0)} = \Sigma_f\left(w^{(0)}\right)$.

- Let $t = 0$.

- **while** $\|\Sigma_f^{(t)} - \mathbb{E}\, f\left(\boldsymbol{\eta}^*\right) f\left(\boldsymbol{\eta}^*\right)^\top\| > C\varepsilon \log(1/\varepsilon)/\alpha^3$ **do**

  - Compute $v^{(t)}$ the top eigenvector of $\Sigma_f^{(t)} - \mathbb{E}\, f\left(\boldsymbol{\eta}^*\right) f\left(\boldsymbol{\eta}^*\right)^\top$.

  - For $i \in [n]$, compute $\tau_i^{(t)} = \left\langle v^{(t)}, f\left(\hat{\mu}^{(t)} - y_i\right)\right\rangle^2$.
  - Sort the $\tau_i$. Assume that $\tau_1 \geqslant \ldots \geqslant \tau_n$.
  - Let $N$ be the smallest index such that $\sum_{i \leqslant N} w_i^{(t)} > 2\varepsilon$ and let $\tau_{\max} = \max_{i \in [n]} \tau_i$.
  - For $1 \leqslant i \leqslant N$, set $w_i^{(t+1)} = \left(1 - \frac{\tau_i}{\tau_{\max}}\right) \cdot w_i^{(t)}$ and for $N < i \leqslant n$, set $w_i^{(t+1)} = w_i^{(t)}$.
  - Compute $\hat{\mu}^{(t+1)} = \min_{\mu \in \mathbb{R}^d} f^{w^{(t+1)}}(\mu)$ and $\Sigma_f^{(t+1)} = \Sigma_f\left(w^{(t+1)}\right)$.
  - $t \leftarrow t + 1$.

- Output $\hat{\mu}^{(t)}$.

---

We will use the following lemma whose proof we will give at the end of this section

**Lemma I.3.** *Assume that* $\|\Sigma_f^{(t)} - \mathbb{E}\, f\left(\boldsymbol{\eta}^*\right) f\left(\boldsymbol{\eta}^*\right)^\top\| > C\varepsilon \log(1/\varepsilon)/\alpha^3$ *and*

$$\sum_{i \in S_g} \left(\frac{1}{n} - w_i^{(t)}\right) < \sum_{i \in S_b} \left(\frac{1}{n} - w_i^{(t)}\right).$$

*Then*

$$\sum_{i \in S_g} \left(\frac{1}{n} - w_i^{(t+1)}\right) < \sum_{i \in S_b} \left(\frac{1}{n} - w_i^{(t+1)}\right).$$

With this in hand, we will prove Theorem I.1.

*Proof of Theorem I.1.* First note, that every iteration can clearly be implented to run in time $n^{O(1)} d^{O(1)}$. We will show that the algorithm terminates after at most $\lceil 2\varepsilon n \rceil$ iterations. Assume towards a contradiction that the algorithm does not terminate after $T = \lceil 2\varepsilon n \rceil$ iterations. Note that the number entries of $w^{(t)}$ that are equal to 0 increases by at least 1 in every iteration. Hence, after $T$ iterations we have set at least $\varepsilon n$ entries of $w$ to zero whose index lies in $S_g$. By assumption that the algorithm didn't terminate and Lemma I.3 it holds that

$$\varepsilon \leqslant \sum_{i \in S_g} \left(\frac{1}{n} - w_i^{(T)}\right) < \sum_{i \in S_b} \left(\frac{1}{n} - w_i^{(T)}\right) \leqslant \frac{|S_b|}{n} \leqslant \varepsilon.$$

which is a contradiction.

Next, we prove the correctness of the algorithm. Let $T$ be the index of the last iteration of the algorithm before termination. Note that by our invariant

$$\left\|\frac{1}{n} - w^{(T)}\right\|_1 = \sum_{i \in S_g} \frac{1}{n} - w_i^{(T)} + \sum_{i \in S_b} \frac{1}{n} - w_i^{(T)} < 2\sum_{i \in S_b} \frac{1}{n} - w_i^{(T)} \leqslant 2\varepsilon.$$

Since also $0 \leqslant w^{(T)} \leqslant \frac{1}{n}$, it follows that $w^{(T)} \in \mathcal{W}_{2\varepsilon}$. By Theorem H.1 and since $\|\Sigma_f^{(T)} - \mathbb{E}\, f\left(\boldsymbol{\eta}^*\right) f\left(\boldsymbol{\eta}^*\right)^\top\| \leqslant C\varepsilon \log(1/\varepsilon)/\alpha^3$ it follows that

$$\|\hat{\mu}^T - \mu^*\| \leqslant O\left(\rho \cdot \left[\sqrt{\frac{d\log(d) + \log(1/\delta)}{n}} + \frac{\varepsilon\sqrt{\log(1/\varepsilon)}}{\alpha^3}\right]\right).$$

$\square$

Lastly, we will prove Lemma I.3.

*Proof of Lemma I.3.* On a high level, we will show that each iteration of the filtering algorithm removes relatively more weight from corrupted samples than from uncorrupted ones. For simplicity, let $w = w^{(t)}$ and $w' = w^{(t+1)}$. Also, let $\hat{\mu} = \hat{\mu}^{(t)}$, $\Sigma_f = \Sigma_f^{(t)}$, and $v = v^{(t)}$. Note, that it is enough to show that

$$\sum_{i \in S_g} w_i - w_i' < \sum_{i \in S_b} w_i - w_i' .$$

Let $N$ be the smallest index such that $\sum_{i \leqslant N} w_i > \varepsilon$ and denote by $T = \{1, \ldots, N\}$ Further, recall that $w_i' = \left(1 - \frac{\tau_i}{\tau_{\max}}\right) w_i$ if $i \in T$ and $w_i' = w_i$ otherwise. Thus, for $i \in T$, it holds that $w_i - w_i' = \frac{1}{\tau_{\max}} \tau_i w_i$, while for $i \notin T$ it holds that $w_i - w_i' = 0$. Hence, the above condition is equivalent to

$$\sum_{i \in S_g \cap T} w_i \tau_i < \sum_{i \in S_b \cap T} w_i \tau_i .$$

We will show that the above is indeed true in two steps. We will show that

1. $\sum_{i \in S_g \cap T} w_i \tau_i \leqslant O\left(\varepsilon \log(1/\varepsilon) + \frac{\varepsilon^2}{\alpha^3} \cdot \left\|\Sigma_f - \mathbb{E} f(\boldsymbol{\eta}^*) f(\boldsymbol{\eta}^*)^\top\right\|\right)$,

2. $\sum_{i \in S_b \cap T} w_i \tau_i \geqslant \Omega\left(\left\|\Sigma_f - \mathbb{E} f(\boldsymbol{\eta}^*) f(\boldsymbol{\eta}^*)^\top\right\|\right)$.

This implies our claim since by assumption $\left\|\Sigma_f - \mathbb{E} f(\boldsymbol{\eta}^*) f(\boldsymbol{\eta}^*)^\top\right\| > C\varepsilon \log(1/\varepsilon)/\alpha^3 > C\varepsilon \log(1/\varepsilon)$ and $\frac{\varepsilon^2}{\alpha^3} \leqslant \frac{\varepsilon}{\alpha^3}$ is sufficiently small.

**Upper Bounding the Contribution of $S_g \cap T$** Recall that $n \geqslant \frac{d \log(d) + \log(1/\delta)}{\alpha^6 \varepsilon^2 \log(1/\varepsilon)}$. Thus, by definition of $\tau_i$ and since $f$ is 1-Lipschitz it follows that

$$\sum_{i \in S_g \cap T} w_i \tau_i = \sum_{i \in S_g \cap T} w_i \langle f(\hat{\mu} - \boldsymbol{y}_i^*), v\rangle^2$$

$$\leqslant O\left(\sum_{i \in S_g \cap T} w_i \langle f(\mu^* - \boldsymbol{y}_i^*), v\rangle^2 + \sum_{i \in S_g \cap T} w_i \langle f(\hat{\mu} - \boldsymbol{y}_i^*) - f(\mu^* - \boldsymbol{y}_i^*), v\rangle^2\right)$$

$$\leqslant O\left(\varepsilon \log(1/\varepsilon) + \sum_{i \in S_g \cap T} w_i \|\hat{\mu} - \mu^*\|^2\right)$$

$$\leqslant O\left(\varepsilon \log(1/\varepsilon) + \varepsilon \cdot \left(\frac{1}{\alpha^3} \cdot \left[\varepsilon \left\|\Sigma_f - \mathbb{E} f(\boldsymbol{\eta}^*) f(\boldsymbol{\eta}^*)^\top\right\| + \varepsilon^2 \log(1/\varepsilon)\right]\right)\right)$$

$$\leqslant O\left(\varepsilon \log(1/\varepsilon) + \frac{\varepsilon^2}{\alpha^3} \cdot \left\|\Sigma_f - \mathbb{E} f(\boldsymbol{\eta}^*) f(\boldsymbol{\eta}^*)^\top\right\|\right) ,$$

where in the second inequality we used the last property of Fact J.7 and in the third inequality we used Theorem H.1.

**Lower Bounding the Contribution of $S_b \cap T$** Our strategy will be to first show that the contribution of all points in $S_b$, also those not necessarily in $T$, must be large and then show that the contribution of points in $S_b \setminus T$ is in fact small. Together they will yield our claim. Note that by definition of $v$, it holds that

$$\sum_{i=1}^n w_i \tau_i = v^\top \left(\Sigma_f - \mathbb{E} f(\boldsymbol{\eta}^*) f(\boldsymbol{\eta}^*)^\top\right) v + \mathbb{E} \langle f(\boldsymbol{\eta}^*), v\rangle^2 = \mathbb{E} \langle f(\boldsymbol{\eta}^*), v\rangle^2 + \left\|\Sigma_f - \mathbb{E} f(\boldsymbol{\eta}^*) f(\boldsymbol{\eta}^*)^\top\right\| .$$

Let $\gamma := \mathbb{E}\langle f(\boldsymbol{\eta}^*), v\rangle^2$. Note that $\gamma = O(1)$. By the second point in Lemma H.2, Cauchy–Schwarz inequality and Theorem H.1,

$$\left|\sum_{i\in S_g} w_i\tau_i - \gamma\right| = \left|\sum_{i\in S_g} w_i\langle f(\hat{\mu} - y_i), v\rangle^2 - \gamma\right|$$

$$\leqslant \left|\sum_{i\in S_g} w_i\langle f(\mu^* - \boldsymbol{y}_i^*), v\rangle^2 - \gamma\right| + \left|\sum_{i\in S_g} w_i\langle f(\hat{\mu} - \boldsymbol{y}_i^*) - f(\mu^* - \boldsymbol{y}_i^*), v\rangle^2\right|$$

$$+ 2\left|\sum_{i\in S_g} w_i\langle f(\mu^* - \boldsymbol{y}_i^*), v\rangle\left(\langle f(\hat{\mu} - \boldsymbol{y}_i^*), v\rangle - \langle f(\mu^* - \boldsymbol{y}_i^*), v\rangle\right)\right|$$

$$\leqslant O\left(\varepsilon\log(1/\varepsilon) + \|\hat{\mu} - \mu^*\|^2 + \sqrt{\gamma}\cdot\|\hat{\mu} - \mu^*\|\right)$$

$$\leqslant O\left(\varepsilon\log(1/\varepsilon) + \frac{\varepsilon}{\alpha^3}\left\|\Sigma_f - \mathbb{E}f(\boldsymbol{\eta}^*)f(\boldsymbol{\eta}^*)^\top\right\| + \frac{\varepsilon^2\log(1/\varepsilon)}{\alpha^3}\right)$$

$$\leqslant O\left(\frac{\varepsilon}{\alpha^3}\left\|\Sigma_f - \mathbb{E}f(\boldsymbol{\eta}^*)f(\boldsymbol{\eta}^*)^\top\right\| + \frac{\varepsilon^2\log(1/\varepsilon)}{\alpha^3} + \|\hat{\mu} - \mu^*\|\right).$$

Hence, since $\left\|\Sigma_f - \mathbb{E}f(\boldsymbol{\eta}^*)f(\boldsymbol{\eta}^*)^\top\right\|$ is at least a sufficiently large multiple of $\varepsilon\log(1/\varepsilon)/\alpha^3$ and $\frac{\varepsilon}{\alpha^2}$ is a sufficiently small constant, it follows that the first two terms are at most $0.0001\left\|\Sigma_f - \mathbb{E}f(\boldsymbol{\eta}^*)f(\boldsymbol{\eta}^*)^\top\right\|$. The second term is at most

$$O\left(\sqrt{\frac{\varepsilon}{\alpha^3}\left\|\Sigma_f - \mathbb{E}f(\boldsymbol{\eta}^*)f(\boldsymbol{\eta}^*)^\top\right\|}\right).$$

We claim that this is at most $0.0001\left\|\Sigma_f - \mathbb{E}f(\boldsymbol{\eta}^*)f(\boldsymbol{\eta}^*)^\top\right\|$ as well. Indeed, if it were larger, we would have that $\left\|\Sigma_f - \mathbb{E}f(\boldsymbol{\eta}^*)f(\boldsymbol{\eta}^*)^\top\right\| = O(\frac{\varepsilon}{\alpha^3})$, a contradiction. Thus, we get that

$$\sum_{i\in S_b} w_i\tau_i \geqslant 0.99\left\|\Sigma_f - \mathbb{E}f(\boldsymbol{\eta}^*)f(\boldsymbol{\eta}^*)^\top\right\|.$$

It remains to upper bound the contribution of indices $i$ in $S_b \setminus T$. Note that

$$\sum_{i\in S_b\cap T} w_i\tau_i = \sum_{i\in S_b} w_i\tau_i - \sum_{i\in S_b\setminus T} w_i\tau_i.$$

We claim that the scores for $i \notin T$ cannot be too large. Indeed, since $\sum_{i\in T} w_i > 2\varepsilon$ and $\sum_{i\in S_b\cap T} w_i \leqslant \varepsilon$ it follows that $\sum_{i\in S_g\cap T} w_i \geqslant \varepsilon$. Thus, since we assume the $\tau_i$ are sorted in decreasing order it follows that for $i \notin T$ it holds that

$$\tau_i \leqslant \frac{1}{\sum_{i\in S_g\cap T} w_i}\sum_{i\in S_g\cap T} w_i\tau_i \leqslant O\left(\log(1/\varepsilon) + \frac{\varepsilon}{\alpha^3}\cdot\left\|\Sigma_f - \mathbb{E}f(\boldsymbol{\eta}^*)f(\boldsymbol{\eta}^*)^\top\right\|\right).$$

It follows that

$$\sum_{i\in S_b\setminus T} w_i\tau_i \leqslant O\left(\varepsilon\log(1/\varepsilon) + \frac{\varepsilon^2}{\alpha^3}\cdot\left\|\Sigma_f - \mathbb{E}f(\boldsymbol{\eta}^*)f(\boldsymbol{\eta}^*)^\top\right\|\right)$$

Putting everything together we have shown that

$$\sum_{i\in S_b\cap T} w_i\tau_i \geqslant 0.9\left\|\Sigma_f - \mathbb{E}f(\boldsymbol{\eta}^*)f(\boldsymbol{\eta}^*)^\top\right\|$$

as desired.

Finally, it is easy to see that if we can only compute an estimation $\hat{v}$ of $v$ such that

$$\hat{v}^\top\left(\Sigma_f - \mathbb{E}f(\boldsymbol{\eta}^*)f(\boldsymbol{\eta}^*)^\top\right)\hat{v} = \left\|\Sigma_f - \mathbb{E}f(\boldsymbol{\eta}^*)f(\boldsymbol{\eta}^*)^\top\right\| \pm O(\varepsilon),$$

the proof still works, so we can use the $O(\varepsilon)$-close estimation $\hat{\Sigma}$ of $\mathbb{E}f(\boldsymbol{\eta}^*)f(\boldsymbol{\eta}^*)^\top$ and compute the top eigenvector of $\Sigma_f - \hat{\Sigma}$. $\qquad\square$

## J Concentration Bounds

Throughout this section we will use the following versions of the Matrix Bernstein Inequality. A proof of a slightly more general version can be found in [Tro15, Corollary 6.2.1].

**Fact J.1.** *Let $L, B > 0$, $M \in \mathbb{R}^{d \times d}$ be a symmetric random matrix and $M_1, \ldots, M_n$ i.i.d. copies of $M$. Suppose that $\|M\| \leqslant L$. Further, assume that $\|\mathbb{E}\, M^2\| \leqslant B$. Then the estimator $\bar{M} = \frac{1}{n} \sum_{i=1}^n M_i$ satisfies for all $t > 0$*

$$\mathbb{P}\left(\|\bar{M} - \mathbb{E}\, M\| \geqslant t\right) \leqslant 2d \cdot \exp\left(-\frac{t^2 n}{2B + 2Lt}\right).$$

**Fact J.2.** *Let $x_1, \ldots, x_n$ be iid $d$-dimensional random vectors such that $\|x_1\| \leqslant L$ with probability 1. Then*

$$\mathbb{P}\left(\|\bar{v} - \mathbb{E}\, v\| \geqslant t\right) \leqslant 2d \cdot \exp\left(-\frac{t^2 n}{2L^2 + 2Lt}\right).$$

**Fact J.3.** *Let $L > 0$, $v \in \mathbb{R}^d$ be a random vector and $v_1, \ldots, v_n$ i.i.d. copies of $v$. Suppose that $\|v\| \leqslant L$. Then the estimator $\bar{\Sigma} = \frac{1}{n} \sum_{i=1}^n x_i x_i^\top$ satisfies for all $t > 0$*

$$\mathbb{P}\left(\|\bar{\Sigma} - \mathbb{E}\, x_1 x_1^\top\| \geqslant t\right) \leqslant 2d \cdot \exp\left(-\frac{t^2 n}{2L\|\Sigma\| + 2Lt}\right).$$

We will use this to prove that sampling preserves bounded $f$-moments.

**Lemma J.4.** *Let $k, \ell, n, d \in \mathbb{N}_{\geqslant 1}$, $\sigma, \rho > 0$, $\delta \in (0, 1)$. Let $\nu$ be a distribution with $(2k, \ell)$-certifiable $\sigma$-bounded $f$-moments, and $\eta_1, \ldots, \eta_n \sim_{i.i.d.} \nu$. Suppose that $n \geqslant 10 \cdot d^k \cdot \log(d/\delta)$ and that $\forall x \in \mathbb{R}^d$, $\|f(x)\| \leqslant \rho\sqrt{d}$.*

*Then with probability at least $1 - \delta$, uniform distribution over the set $\{\eta_1, \ldots, \eta_n\}$ has $(2k, \ell)$-certifiable $(\sigma + \rho)$-bounded $f$-moments. In other words, with probability at least $1 - \delta$,*

$$\left\lvert\frac{v}{\ell}\right. \left(v^{\otimes k}\right)^\top \left(\frac{1}{n}\sum_{i=1}^n f(\eta_i)^{\otimes k}\left(f(\eta_i)^{\otimes k}\right)^\top\right) v^{\otimes k} \leqslant ((\sigma + \rho) \cdot \|v\|)^{2k}.$$

*If $\eta_1, \ldots, \eta_n$ follow an elliptical distribution with scatter matrix $\Sigma$ and $\sigma = O(\sqrt{k\|\Sigma\|})$, we only need $n \gtrsim (\frac{r(\Sigma)}{k})^k \cdot \log(d/\delta)$.[10] Further, the uniform distribution has $2\sigma$-bounded $f$-moments.*

*Proof.* Let $T_k = \mathbb{E}\, f(\eta)^{\otimes k}\left(f(\eta)^{\otimes k}\right)^\top$. Note that $\{\eta\}$ having $(2k, \ell)$-certifiable $\sigma$-bounded $f$-moments is equivalent to the fact that the following SoS proof exists

$$\left\lvert\frac{v}{\ell}\right. \left(v^{\otimes k}\right)^\top T_k v^{\otimes k} \leqslant (\sigma \cdot \|v\|)^{2k}.$$

Denote $\bar{T}_k = \frac{1}{n}\sum_{i=1}^n f(\eta_i)^{\otimes k}\left(f(\eta_i)^{\otimes k}\right)^\top$. Notice that by Fact K.3

$$\left\lvert\frac{v}{\ell}\right. \left(v^{\otimes k}\right)^\top \bar{T}_k v^{\otimes k} = \left(v^{\otimes k}\right)^\top \left[T_k + \bar{T}_k - T_k\right] v^{\otimes k} \leqslant \|v\|^{2k} \cdot \left(\sigma^{2k} + \|\bar{T}_k - T_k\|\right).$$

Hence, it is enough to bound $\|\bar{T}_k - T_k\|$. Our strategy will be to use Fact J.1. To this end, let $M = f(\eta)^{\otimes k}\left(f(\eta)^{\otimes k}\right)^\top$ and for $i \in [n]$, let $M_i = f(\eta_i)^{\otimes k}\left(f(\eta_i)^{\otimes k}\right)^\top$. This implies that $T_k = \mathbb{E}\, M$ and $\bar{T}_k = \frac{1}{n}\sum_{i=1}^n M_i$. Note that since the entries of $f$ are bounded by $h$ in magnitude it follows that for each $i$

$$\|M_i\| = \|f(\eta_i)\|^{2k} \leqslant \rho^{2k} d^k.$$

Further,

$$M^2 = \|f(\eta)\|^{2k} \cdot f(\eta)^{\otimes k}\left(f(\eta)^{\otimes k}\right)^\top \preceq \rho^{2k} d^k \cdot f(\eta)^{\otimes k}\left(f(\eta)^{\otimes k}\right)^\top.$$

Since $\left\|\mathbb{E}\, f(\eta)^{\otimes k}\left(f(\eta)^{\otimes k}\right)^\top\right\| \leqslant \sigma^{2k}$, it holds that

$$\left\|\mathbb{E}\, M^2\right\| \leqslant \rho^{2k} d^k \cdot \left\|\mathbb{E}\, f(\eta)^{\otimes k}\left(f(\eta)^{\otimes k}\right)^\top\right\| \leqslant (\rho\sigma)^{2k} d^k.$$

---

[10] Here we use the parametrization from Appendix G, where $\mathrm{Tr}\,\Sigma = d$

By Fact J.1,

$$\mathbb{P}\left(\left\|\bar{\boldsymbol{T}}_k - T_k\right\| \geqslant \frac{10\rho^{2k}d^k \cdot \log(d/\delta)}{n} + \sigma^k\sqrt{\frac{10\rho^{2k}d^k \cdot \log(d/\delta)}{n}}\right) \leqslant \delta. \tag{J.1}$$

Hence with probability at least $1 - \delta$,

$$\left|\frac{v}{\ell}\left(v^{\otimes k}\right)^\top \bar{\boldsymbol{T}}_k v^{\otimes k}\right| \leqslant \|v\|^{2k} \cdot \left(\sigma^{2k} + \rho^{2k} + \rho^k\sigma^k\right) \leqslant \|v\|^{2k} \cdot (\sigma + \rho)^{2k}$$

To see the improved sample complexity for elliptical distributions, note that since $\mathrm{r}(\Sigma) = \frac{\mathrm{Tr}(\Sigma)}{\|\Sigma\|} = \frac{d}{\|\Sigma\|}$ and $\sigma = O(\rho\sqrt{k\|\Sigma\|})$, it follows that

$$\rho^{2k}d^k = \rho^{2k}\|\Sigma\|^k\mathrm{r}(\Sigma)^k \cdot \frac{k^k}{k^k} = \sigma^{2k} \cdot \left(\frac{\mathrm{r}(\Sigma)}{k}\right)^k.$$

Thus, if $n \gtrsim (\frac{\mathrm{r}(\Sigma)}{k})^k \cdot \log(d/\delta)$, then $\left\|\bar{\boldsymbol{T}}_k - T_k\right\|$ in Eq. (J.1) is at most $\sigma^{2k}$ with probability at least $1 - \delta$. $\qquad\square$

**Fact J.5.** *[DTV16] Let $\boldsymbol{\eta}$ be an elliptically distributed vector with location $0$ and scatter matrix $\Sigma$. Let $s(\boldsymbol{\eta}) = \frac{\sqrt{\mathrm{Tr}(\Sigma)}}{\|\boldsymbol{\eta}\|}\boldsymbol{\eta}$. Then*

$$\left\|\mathbb{E}\, s(\boldsymbol{\eta})\, s(\boldsymbol{\eta})^\top\right\| \leqslant \|\Sigma\|.$$

**Fact J.6.** *[Bec09] Let $\boldsymbol{x} \sim N(0, \Sigma)$. Then for all $t > 0$,*

$$\mathbb{P}\left[\|\boldsymbol{x}\|^2 \leqslant \mathrm{Tr}(\Sigma) - \sqrt{2t}\,\|\Sigma\|_{\mathrm{F}}\right] \leqslant \exp(-t)$$

*and*

$$\mathbb{P}\left[\|\boldsymbol{x}\|^2 \geqslant \mathrm{Tr}(\Sigma) + \sqrt{2t}\,\|\Sigma\|_{\mathrm{F}} + t\,\|\Sigma\|\right] \leqslant \exp(-t).$$

**Fact J.7.** *[DK22] Let $\boldsymbol{\xi}_1, \ldots, \boldsymbol{\xi}_n$ be independent zero mean $1$-sub-Gaussian $d$-dimensional vectors. Let $\varepsilon, \delta \in (0, 1)$, and suppose that $n \geqslant d + \log(1/\delta)$. Let $\mathcal{W}_\varepsilon = \left\{w \in \mathbb{R}^d \mid 0 \leqslant w_i \leqslant 1/n, \sum_{i=1}^n |w_i - 1/n| \leqslant \varepsilon\right\}$. Then, with probability $1 - \delta$, for every $w \in \mathcal{W}_\varepsilon$,*

$$\left\|\sum_{i=1}^n w_i\boldsymbol{\xi}_i\right\| \leqslant O\left(\varepsilon\sqrt{\log(1/\varepsilon)} + \sqrt{\frac{d + \log(1/\delta)}{n}}\right),$$

*and*

$$\left\|\sum_{i=1}^n w_i\boldsymbol{\xi}_i\boldsymbol{\xi}_i^\top - \frac{1}{n}\sum_{i=1}^n \mathbb{E}\,\boldsymbol{\xi}\boldsymbol{\xi}^\top\right\| \leqslant O\left(\varepsilon\log(1/\varepsilon) + \sqrt{\frac{d + \log(1/\delta)}{n}}\right).$$

*Similarly, with probability $1 - \delta$ it holds that for every set $T \subseteq [n]$ with $|T| \leqslant \varepsilon n$ and every $w \in \mathcal{W}_e$ it holds that*

$$\left\|\sum_{i \in T} w_i\xi_i\xi_i^\top\right\| \leqslant O\left(\varepsilon\log(1/\varepsilon) + \sqrt{\frac{d + \log(1/\delta)}{n}}\right).$$

**Lemma J.8.** *Let $\varepsilon, \delta \in (0, 1)$ and suppose that $n \geqslant d\log d + \log(1/\delta)$. Let $\boldsymbol{\xi}_1, \ldots, \boldsymbol{\xi}_n$ be independent zero mean $1$-sub-Gaussian $d$-dimensional vectors, and let $a_1, \ldots, a_n \in \mathbb{R}^d$ be fixed (non-random) vectors such that $\|a_i\| \leqslant 100\varepsilon\sqrt{d}/\alpha$ for some $\alpha \in (0, 1)$. Suppose in addition that $\|\boldsymbol{\xi}_i\| \leqslant 10\sqrt{d}$ with probability 1. Let $\mathcal{W}_\varepsilon = \left\{w \in \mathbb{R}^d \mid 0 \leqslant w_i \leqslant 1/n, \sum_{i=1}^n |w_i - 1/n| \leqslant \varepsilon\right\}$. Then, with probability $1 - \delta$, for every $w \in \mathcal{W}_\varepsilon$,*

$$\sum_{i=1}^n w_i\left(\boldsymbol{\xi}_i + a_i\right)\left(\boldsymbol{\xi}_i + a_i\right)^\top \succeq \frac{1}{n}\sum_{i=1}^n \mathbb{E}\,\boldsymbol{\xi}\boldsymbol{\xi}^\top - O\left(\varepsilon\log(1/\varepsilon) + \varepsilon/\alpha + \sqrt{\frac{d\log(d) + \log(1/\delta)}{n}}\right).$$

*Proof.* Note that

$$\sum_{i=1}^n w_i\left(\boldsymbol{\xi}_i + a_i\right)\left(\boldsymbol{\xi}_i + a_i\right)^\top = \sum_{i=1}^n w_i\boldsymbol{\xi}_i\boldsymbol{\xi}_i^\top + \sum_{i=1}^n w_i\left(\boldsymbol{\xi}_i a_i^\top + a_i\boldsymbol{\xi}_i^\top + a_i a_i^\top\right).$$

Let us bound $u\sum_{i=1}^{n} w_i \left(\boldsymbol{\xi}_i a_i^\top + a_i \boldsymbol{\xi}_i^\top + a_i a_i^\top\right) u^\top$ for all unit vectors $u \in \mathbb{R}^d$. Note that since $\mathcal{W}_\varepsilon$ is a polytope, it is enough to bound this value only for $w$ that are its vertices, that is, the indicators of sets of size at least $(1 - \varepsilon)\, n$, normalized by $1/n$. Let $S \subset [n]$ be an arbitrary fixed (non-random) set, and let $u \in \mathbb{R}^d$ be an arbitrary fixed (non-random) unit vector. Then

$$u^\top \sum_{i \in S} \tfrac{1}{n} \left(\boldsymbol{\xi}_i a_i^\top + a_i \boldsymbol{\xi}_i^\top + a_i a_i^\top\right) u = \tfrac{1}{n} \sum_{i \in S} 2\langle \boldsymbol{\xi}_i, u\rangle \langle \boldsymbol{a}_i, u\rangle + \langle \boldsymbol{a}_i, u\rangle^2 \,.$$

By Hoeffding's inequality, with probability at least $1 - \delta$,

$$\sum_{i \in S} 2\langle \boldsymbol{\xi}_i, u\rangle \langle \boldsymbol{a}_i, u\rangle + \langle \boldsymbol{a}_i, u\rangle^2 \geqslant -O\left(\|A_S u\|\, \sqrt{\log(1/\delta)}\right) + \|A_S u\|^2 \geqslant -O\left(\log(1/\delta)\right) \,.$$

where $A_S \in \mathbb{R}^{|S| \times d}$ is the matrix with rows $a_i$ for $i \in S$.

By union bound over all sets of size at least $(1 - \varepsilon)\, n$, with probability $1 - \delta$, for every set $S$ of size at least $(1 - \varepsilon)\, n$,

$$\sum_{i \in S} 2\langle \boldsymbol{\xi}_i, u\rangle \langle \boldsymbol{a}_i, u\rangle + \langle \boldsymbol{a}_i, u\rangle^2 \geqslant -O\left(\varepsilon n \log(1/\varepsilon) + \log(1/\delta)\right) \,.$$

Let $\mathcal{N}$ be an $(0.01/d)$-net in the $d$-dimensional unit ball of size $|\mathcal{N}| \leqslant (300d)^d$. Then by union bound over $\mathcal{N}$, we get that with with probability $1 - \delta$, for every set $S$ of size at least $(1 - \varepsilon)\, n$, and for every $u \in \mathcal{N}$,

$$\sum_{i \in S} 2\langle \boldsymbol{\xi}_i, u\rangle \langle \boldsymbol{a}_i, u\rangle + \langle \boldsymbol{a}_i, u\rangle^2 \geqslant -O\left(\varepsilon n \log(1/\varepsilon) + d\log(d) + \log(1/\delta)\right) \,.$$

Now, if some unit $u$ is $(0.01/d)$-close to $u' \in \mathcal{N}$, we get

$$\tfrac{1}{n} \sum_{i \in S} 2\langle \boldsymbol{\xi}_i, u\rangle \langle \boldsymbol{a}_i, u\rangle \geqslant \tfrac{1}{n} \sum_{i \in S} 2\langle \boldsymbol{\xi}_i, u'\rangle \langle \boldsymbol{a}_i, u'\rangle - O\left(\max_i \|\boldsymbol{\xi}_i\| \cdot r \cdot \|u - u'\|\right)$$

$$\geqslant -O\left(\varepsilon \log(1/\varepsilon) + \varepsilon/\alpha + \frac{d\log(d) + \log(1/\delta)}{n}\right) \,.$$

Using the concentration of $\sum_{i=1}^{n} w_i \boldsymbol{\xi}_i \boldsymbol{\xi}_i^\top$ from Fact J.7, we get the desired bound. $\qquad\square$

# K   Sum-of-Squares Toolkit

The following fact can be found in [KS17a, Lemma A.2]

**Fact K.1.** *For all $k \in \mathbb{N}_{\geqslant 1}$ it holds that*

$$\left|\frac{X,Y}{2k}\right| (X + Y)^{2k} \leqslant 2^{2k-1} \cdot \left(X^{2k} + Y^{2K}\right) \,.$$

The next fact shows that sum-of-squares captures the Cauchy-Schwarz Inequality, a proof can be found in [MSS16, Lemma A.1]

**Fact K.2.** *For all $n \in \mathbb{N}_{\geqslant 1}$ it holds that*

$$\left|\frac{X_1, Y_1, \ldots, X_n, Y_n}{2}\right| \left(\sum_{i=1}^{n} X_i Y_i\right)^2 \leqslant \left(\sum_{i=1}^{n} X_i^2\right)\left(\sum_{j=1}^{n} Y_i^2\right)$$

**Fact K.3.** *Let $M \in \mathbb{R}^{n \times n}$ be a symmetric matrix and $X$ be an $n$-vector of formal variables. Then it holds that*

$$\left|\frac{X}{2}\right| \langle X, MX\rangle \leqslant \|M\|\, \|X\|^2 \,.$$

*Proof.* We can rewrite the inequality as $X^\top \left(\|M\| \cdot I_n - M\right) X \geqslant 0$. Since $\|M\| \cdot I_n - M$ is positive semi-definite, it follows that there exists a matrix $L \in \mathbb{R}^{n \times n}$ such that $\|M\| \cdot I_n - M = LL^\top$. Hence,

$$X^\top \left(\|M\| \cdot I_n - M\right) X = \|LX\|^2 \,,$$

which is a sum of squares in $X$. $\qquad\square$

**Lemma K.4.** *Let $k$ be a positive integer, and let $\boldsymbol{\eta}$ be elliptical $d$-dimensional vector with location $0$ and scatter matrix $\Sigma$ that satisfies*

$$\|\Sigma\|_F \leqslant \frac{\operatorname{Tr}(\Sigma)}{10\sqrt{k \log d}}.$$

*Let $f : \mathbb{R}^d \to \mathbb{R}^d$ be a projection onto the Euclidean ball of radius $R$ centered at $0$.*

*Then $\boldsymbol{\eta}$ has $(2k, 2k)$-certifiable $q$-bounded $f$-moments, where*

$$q = 2R\sqrt{\frac{k\|\Sigma\|}{\operatorname{Tr}(\Sigma)}}.$$

*Proof.* Let $v_1, \ldots, v_d$ be variables and consider the polynomial $\mathbb{E}\langle f(\boldsymbol{\eta}), v\rangle^{2k}$. Let $s(\boldsymbol{\eta}) = \frac{R}{\|\boldsymbol{\eta}\|}\boldsymbol{\eta} = \frac{R}{\|f(\boldsymbol{\eta})\|}f(\boldsymbol{\eta})$. Since spherical projection of elliptical distributions depends only on $\Sigma$ (see Theorem 35 in [Fra04]), $s(\boldsymbol{\eta})$ has the same distribution as $s(\boldsymbol{w})$, where $\boldsymbol{w} \sim N(0, \Sigma)$. Hence

$$\left|\frac{v}{2k}\langle f(\boldsymbol{\eta}), v\rangle^{2k} \leqslant \langle s(\boldsymbol{\eta}), v\rangle^{2k}\right.$$

$$= R^{2k}\left\langle \frac{\boldsymbol{w}}{\|\boldsymbol{w}\|}, v\right\rangle^{2k}$$

$$= R^{2k} \cdot \mathbf{1}_{\left[\|\boldsymbol{w}\|^2 > \frac{\operatorname{Tr}(\Sigma)}{2}\right]}\left\langle \frac{\boldsymbol{w}}{\|\boldsymbol{w}\|}, v\right\rangle^{2k} + R^{2k} \cdot \mathbf{1}_{\left[\|\boldsymbol{w}\|^2 \leqslant \frac{\operatorname{Tr}(\Sigma)}{2}\right]}\left\langle \frac{\boldsymbol{w}}{\|\boldsymbol{w}\|}, v\right\rangle^{2k}$$

$$\leqslant \left(\frac{2R^2}{\operatorname{Tr}(\Sigma)}\right)^k \langle \boldsymbol{w}, v\rangle^{2k} + R^{2k} \cdot \mathbf{1}_{\left[\|\boldsymbol{w}\|^2 \leqslant \frac{\operatorname{Tr}(\Sigma)}{2}\right]} \cdot \|v\|^{2k}.$$

By Fact J.6,

$$\mathbb{P}\left[\|\boldsymbol{w}\|^2 \leqslant \operatorname{Tr}(\Sigma)/2\right] \leqslant d^{-2k}.$$

Hence

$$\left|\frac{v}{2k}\mathbb{E}\langle f(\boldsymbol{\eta}), v\rangle^{2k} \leqslant \left(\frac{2R^2}{\operatorname{Tr}(\Sigma)}\right)^k \mathbb{E}\langle \boldsymbol{w}, v\rangle^{2k} + (R/d)^{2k}\|v\|^{2k}.\right.$$

Since $\boldsymbol{w}$ is $(2k, 2k)$-certifiably 1-subgaussian (see Lemma 5.1 in [KS17b]), and since $\|\Sigma\| / \operatorname{Tr}(\Sigma) \geqslant 1/d$,

$$\left|\frac{v}{2k}\mathbb{E}\langle f(\boldsymbol{\eta}), v\rangle^{2k} \leqslant 2 \cdot \left(\frac{2R^2 k\|\Sigma\|}{\operatorname{Tr}(\Sigma)}\right)^k \|v\|^{2k}.\right.$$

$\square$

