# OpenReview forum: "Robust Mean Estimation Without Moments for Symmetric Distributions"
_NeurIPS.cc/2023/Conference — NeurIPS 2023 poster_

### Official Review · Reviewer_DMaa · 2023-06-19

**Soundness:** 3 good
**Presentation:** 3 good
**Contribution:** 2 fair
**Rating:** 5
**Confidence:** 3

**Summary:**

This paper studies outlier-robust location estimation for symmetric(-like) distributions. For "semi-product" distributions, the authors show that one can achieve $O(\epsilon \sqrt{\log 1/\epsilon})$ asymptotic error with a polynomial number of samples (and time), and $O(\epsilon)$ asymptotic error when given quasipolynomially many samples (and time poly in the sample set size). For elliptical distributions, the authors show that as long as the scatter matrix has $\Omega(\log d)$ effective rank (hiding $\epsilon$-dependence here), then in quasipolynomial time we can yield asymptotic error $O(\epsilon \sqrt{\log 1/\epsilon})$. Crucially, for the last result, the scatter matrix is not known to the algorithm.

**Strengths:**

Prior works in (algorithmic) robust statistics get "stuck" at the $\sqrt{\epsilon}$ asymptotic error, when the covariance of the distribution is not known, even when the underlying distribution is guaranteed to have higher moments or might even be sub-Gaussian. This paper identifies semi-product distributions and elliptical distributions as special classes for which further progress can be made. The author(s) adapts the filtering framework, and proposes using variants of the Huber loss as score as opposed to the basic quadratic score in the filtering step. This allows the author(s) to beat the $\sqrt{\epsilon}$ error even when covariance/scatter matrices are unknown to the algorithm.

**Weaknesses:**

I find some parts of the writing clarity can be improved. In particular, I'm still a bit confused about how the guarantees of this work compare with prior works, re: knowledge assumptions on the algorithms. Some of the technical claims are also a bit over-sold (unless I misread or am misunderstanding). See the "Questions" section for more details.

Another weakness, for me, is the motivation: the practical relevance of semi-product and elliptical distributions seems to be rooted in mathematical finance. However, at least from the cited works, the relevance of these distributions appears to more or less be "because we can write down theorems". I hope the authors will consider adding a more self-contained discussion on why we should care about these distributions; I think it will make the paper more convincing.

**Questions:**

More important:

- Knowledge assumption on algorithm in Theorems 1.4 -- 1.6: one of the selling points from the introduction is that the algorithm doesn't need to know the covariance or scatter matrix. However, it was unclear to me whether this means the algorithm knows absolutely nothing about the scatter, or if it only knows very little (such as an upper bound). The theorems can be a lot more explicit in saying exactly what the algorithm needs to know (or explicitly say that it needs to know absolutely nothing about the scatter matrix). For example, does the algorithm in Theorem 1.4 need to know the parameter $\rho$, which is essentially an upper bound to the covariance, at least in the Gaussian case? This distinction seems pretty important to me.

- Result statement of Theorems 1.4 -- 1.5: the statements are of the form "if $n$ is sufficiently large, then we get this almost-sub-Gaussian error rate plus the claimed asymptotic error". However, it seems that the required $n$ lower bound is large enough that the almost-sub-Gaussian error is already dominated by the asymptotic error. Yet the prose following the theorems seem to sell how the theorem gets close to the sub-Gaussian rate (for Theorem 1.4, and for constant $\delta$ and large $C$ for Theorem 1.5). Am I misunderstanding something?

- Theorem 1.6: Line 170 claims that Theorem 1.6 is only slightly sub-optimal in the no-corruption part of the error. But that error is multiplicative in terms of the effective rank and $\log d/\delta$, which is much worse than the sub-Gaussian rate of *additive* error. This error is equal to the naive analysis of entry-wise median-of-means. In fact, entry-wise median-of-means does not actually need the $d$ in $\log d/\delta$ from the naive union bound, and can get error $\sqrt{\mathrm{Tr}(\Sigma)\log\frac{1}{\delta}/n}$, so the error in Theorem 1.6 is even worse than entry-wise median-of-means. Am I missing the point here?

- It is unclear to me that even the sub-Gaussian rate is the correct error rate for the non-robust part of these theorems. For 1-d symmetric distributions, and without corruptions, it was shown by Stone (1975, Maximum Likelihood Estimators of a Location Parameter) that asymptotically one can achieve Fisher information rates, which can be much better than the variance-based sub-Gaussian rate.

- The appendix was hard to follow. There are three appendices with filtering algorithms, but Appendix I was titled only "Filtering Algorithm". It also seemed a bit weird that Appendices H and I appeared last, even though they prove Theorem 1.4, which was introduced earlier than Theorems 1.5 and 1.6. In terms of the actual content, there doesn't seem to be much discussion on how the techniques in the appendices relate to each other. I could see that Appendix E generalizes Appendix D, but how does Appendix I (and relatedly, Appendix H) relate to Appendix E (and C)?

Less important:

- Notation is slightly confusing, with the bold and capital letters. There is no distinction between random scalars and random vectors (e.g. in defining elliptical distributions).

- Page 3, it was claimed that the first 2 properties of semi-product distributions allow for accurate non-robust (i.i.d.) location estimation by using entry-wise median. This seems true only when $\rho$ is small?

- Footnote 1, does $\alpha$ need to be known by the algorithm?

- Footnote 2, there is an $O(\log 1/e)$ which I assume is a typo? Is "$e$" supposed to be "$\epsilon$"?

- CTBJ22 reference: they show how to do mean estimation robustly when only $1+\alpha$ moments exist, but (1) they do not handle the full adversarial corruption model because of sample-splitting issues and (2) they require an upper bound to the $1+\alpha$ moment in every direction.

---

> ### Author Rebuttal · Authors · 2023-08-09
>
> Dear Reviewer,
>
> Thank you for you feedback. We hope to address your concerns in this note. We believe your questions and this discussion form a valuable addition to our paper and we will add this.
>
> ### Knowledge of Problem Parameters
>
> In the setting of the introduction, e.g., Theorems 1.4-1.6, our algorithms do only require the corrupted samples as input. In particular, the parameter $\rho$ for semi-product distributions does not need to be known to the algorithm. All relevant parameters can be estimated from the corrupted samples. The case of unknown $\rho$ is described in Appendix G.1. Note that Appendix G.1 is written for elliptical distributions (it corresponds to the case of unknown trace of the scatter matrix in terms we used in the introduction), but the same argument applies to semi-product distributions (we can find a good upper bound on $\rho$ by working separately with each entry, since distributions of the entries are one-dimensional symmetric, hence elliptical). In the setting of a Gaussian distribution with covariance matrix $\rho^2 \cdot I_d$, this implies that we do not need to know an upper bound on the covariance matrix.
>
> For the more general setting of Definition A.1 (with $\alpha$), either $\alpha$ or $\rho$ needs to be known, but not both. E.g., if we assume $\alpha$ is known, we can estimate $\rho$ from the input.
>
> We agree that our writing with respect to this in some places is unfortunate and we will fix this and add a discussion to the main theorems.
>
> ### Sub-Gaussian Error Rates and Sample-Complexity (Semi-Product Distributions)
>
> You are correct, that since we have a lower bound on the number of samples, the non-robust error term in Theorems 1.4 and 1.5 is always dominated by the robust error. We wrote the error in this way to make the dependence on all problem parameters explicit. We acknowledge that this might be confusing and consider omitting it.
>
> We agree that the discussions in lines 139-140 and 155-156 are indeed confusing. We remark that for Theorem 1.4 we achieve nearly optimal (up to a log factor) sample complexity (necessary to achieve error $\varepsilon\sqrt{\log(1/\varepsilon)}$, and Theorem 1.5 matches what is known for the Gaussian case -- achieving error $O(\varepsilon)$ with quasi-polynomially many samples and in time polynomial in the number of samples.
>
> ### Non-Robust Error of Theorem 1.6 (Elliptical Distributions with Unknown Scatter Matrix)
>
> We agree that in the non-robust setting, when there are no corruptions, there exist simple algorithms achieving better error than the "non-robust error" of Theorem 1.6. The main challenge is to design algorithms which (nearly) match the error of the non-robust setting that are *also* robust against adversarial corruptions (note that the entry-wise median of means estimator suffers from error $\Omega(\varepsilon \sqrt{d})$). This turns out to be a significant challenge.
>
> As mentioned in the paper, specialised to Gaussian distributions with unknown covariance matrix, the optimal (robust) rate is $O(\sqrt{\lVert \Sigma \rVert} \cdot (\sqrt{\tfrac {r(\Sigma) + \log(1/\delta)} n} + \varepsilon))$. An (inefficient) estimator achieving this was only recently found [MZ23]. To the best of our knowledge, there is no efficient estimator matching this rate (even ignoring log-factors in the "robust error"). We would like to clarify, that it is not known how to achieve this rate for general elliptical distributions with unknown scatter matrix, even inefficiently. Note that in [CGR18] the error scales with $d$ instead of the effective rank.
>
>
> ## Sub-Gaussian Error Rate vs Fisher Information Rate
>
> For simplicity, consider the 1-D setting. Let $I_D$ denote the Fisher information of a distribution $D$. It is correct that asymptotically, the "right" error rate for a given symmetric distribution $D$ scales with its Fisher information rate $I_D$. In particular, this asymptotic result could let us hope for an "instance-optimal" algorithm, i.e., when run on distribution $D$, its error scales with $I_D$ -- instead of $\inf_{D \in C} I_D$ for some class of distributions $C$. Note that for the classes of distributions we consider the infimum above corresponds to the sub-gaussian rate.
>
> Unfortunately, it seems unlikely that such instance-optimal results can be achieved in the finite sample regime, even non-robustly. Consider the following mixture distribution: $(1-\gamma)N(0,1) + \gamma \delta_0$, where $\delta_0$ is a Dirac Delta at 0. Then $I_D = \infty$, but when the number of samples $n \ll 1/\gamma$, we only see samples from $N(0,1)$ and thus, the best rate we can hope for is $1/\sqrt{n}$ -- instead of 0.
>
> The example above is taken from the very recent work [GLP23]. They provided an algorithm (in one dimension) using finitely many samples, whose error scales with a quantity related to the Fisher Information but subject to the same constraint outlined above. It would be interesting to obtain similar results in the robust setting. We remark that their techniques rely on likelihood maximization and it is not clear how to make them robust.
>
> ## Ordering of Appendices and Other Comments
>
> Appendices H and I prove Theorem 1.4, while Appendices D-G prove Theorems 1.5 & 1.6. We chose this ordering since the proofs in H & I are technically more complex and easier to understand if one has read D-G, although this is not necessary.
>
> We will also address the minor comments raised by you and add additional motivation for our distributional assumptions.
>
> ## References
>
> [CGR18]: Chen, Gao, Ren, "Robust covariance and scatter matrix estimation under huber’s contamination model"
>
> [MZ23]: Minasyan, Zhivotovskiy, "Statistically Optimal Robust Mean and Covariance Estimation for Anisotropic Gaussians"
>
> [GLP23]: Gupta, Lee, Price, "Finite-Sample Symmetric Mean Estimation with Fisher Information Rate"

---

> > ### Comment · Reviewer_DMaa · 2023-08-10
> >
> > I thank the authors for their response. Before I consider changing my score, I have the following further remarks and questions/requests.
> >
> > - Knowledge assumptions: thank you for the clarification. This is important to add to the main paper and to discuss very clearly and explicitly.
> >
> > - Sub-Gaussian error and sample complexity: I'm ok with the error being written explicitly in terms of all the parameters, but I think the "close to sub-Gaussian error" claim does need to be dropped and changed to a weaker claim instead about almost-optimal sample complexity.
> >
> > - Theorem 1.6: I agree that's the algorithmic challenge. I'm just saying that I disagree with the claim that the result is almost optimal, and I think the claim needs to be dropped.
> >
> > - Fisher information rate: I believe this warrants a discussion in the paper (even if it's in the appendix for space reasons). [GLP23] shows that it's possible to achieve smoothed Fisher information rates for the non-robust (and 1-d) setting, which asymptotically converges to the true Fisher information rate. I'm just pointing out that we can perhaps aim for smoothed Fisher information rate for the non-robust part while retaining a good robust-error, and so the optimality claims in the paper needs to be toned down, given the symmetry assumption.
> >
> > - Motivating the distributional assumptions: I understand that space was limited in the rebuttals, but can the author respond with what they plan to say about the concrete motivations?

---

> > > ### Author Response · Authors · 2023-08-15
> > >
> > > Dear Reviewer,
> > >
> > > Thank you for your additional comments. We will add a detailed discussion about what is known to the algorithm to the paper. We will rephrase the statements regarding the "non-robust error-rate" and its optimality, in particular, (a) changing them to statements about sample complexity and (b) mentioning that it is an exciting open question to go beyond the "Gaussian sample complexity/error rate", and hence, also beyond the guarantees our results achieve, and to aim for a sample complexity/error rate depending on the (smoothed) Fisher Information (similar to [GLP23]). We will also include a discussion about Fisher information similar to the one we had above.
> > >
> > > Regarding motivations for our distributional assumptions: We would like to recall that the motivation for our paper is mostly theoretical. In particular, our work is guided by the question in lines 66-67: "Do there exist classes of heavy-tailed distributions for which we can (efficiently) achieve the same robust error as for the Gaussian distribution?" A weaker version of this question is: "For what classes of distributions can we (efficiently) achieve robust error $o(\sqrt{\varepsilon})$?" We believe characterizing these distributions is an important foundational question in algorithmic robust statistics.
> > >
> > > We believe a natural approach to this question (and the first) is considering classes of distributions satisfying the following criteria
> > > - they generalize the Gaussian distribution
> > > - in the non-robust setting, they behave as nicely as the Gaussian distribution
> > > - they are general, i.e., contain many distributions for which the question is still open
> > > - they are theoretically well-studied
> > >
> > > One such class that has been studied before were sub-gaussian distributions. However, additional assumptions (identity-covariance or "certifiably" sub-gaussian) are needed to go beyond $o(\sqrt{\varepsilon})$ in this case.
> > >
> > > As stated in our submission, our work identifies two other classes satisfying the criteria above, for which error $o(\sqrt{\varepsilon})$ is achievable.
> > > 1. Semi-product distributions: This generalizes the standard Gaussian distribution. And we show how to match the robust error of the Gaussian setting with nearly optimal sample complexity. This was previously known only for sub-Gaussian distributions with covariance identity.
> > > 1. Elliptical distributions: This generalises the "unknown covariance" Gaussian distribution in which it is only known how to achieve error $o(\sqrt{\varepsilon})$  for the Gaussian distribution, due to the algebraic structure of its moments, and "certifiably" sub-Gaussian distributions.
> > >
> > > We find it very appealing that there exist classes of distributions that is well-studied in the statistical literature, for which this is possible.
> > >
> > > Kind regards,
> > > The Authors

---

### Official Review · Reviewer_QF3Z · 2023-07-03

**Soundness:** 4 excellent
**Presentation:** 3 good
**Contribution:** 3 good
**Rating:** 6
**Confidence:** 4

**Summary:**

This paper considers sample complexity of (robust) mean estimation possibly without moments. Two typical examples are product distributions and elliptical distributions. The main technical contribution of this paper is to adapt the filtering techniques to the setting with less restrict moment assumptions.

**Strengths:**

The results of the paper are interesting and new; while the idea may not be. The basic argument relies on filtering, which was previously proposed by [DKK+17] and [DKK+19]. The authors of the paper make a good observation on how the idea of filtering (and coupling) can be used with less restrictive moment assumptions (but it finally replaced by some concentration). The paper is well written, and I enjoyed every minute reading it.

**Weaknesses:**

As mentioned, the main concern is that the authors replace the moment conditions with "concentration", i.e. $P(|\eta| \le \rho) > \frac{1}{100}$ and $P(R \le \sqrt{2d}) \ge \frac{1}{100}$. Moreover, from technique viewpoint, the main components, filtering and coupling (identifiability), are not novel. The time complexity is not explicit (though polynomial). Also I think the paper is lack of (synthetic or real) experiments.




**Questions:**

I have several questions and comments.

(1) In most "useful" applications, some moment assumptions may be needed. For instance, as the authors mentioned, covariance structure may be helpful. Also I may suggest to use "location" parameter instead of "mean". One case is the product Cauchy but it does not have a mean.

(2) p.1, line 16: It may be better to say SQ refers to statistical query (if I don't miss it).

(3) p.3, Def 1.2: If I understand correctly, the variables $(|\eta_j|)_{j = 1}^d$ can be dependent. If so, this is worth commenting since it is (apparently may be slightly) stronger than the usual sense of product measure.

(4) Theorems 1.4-1.6: In all theorems, it is stated that the time complexity is "polynomial", i.e. $n^{\mathcal{O}(1)}$. However, it is not clear whether it is $n$ or $n^{100}$ (this makes difference in high dimensions). I understand that exact or a bound may be difficult. But  I hope the authors may clarify this.

---

> ### Author Rebuttal · Authors · 2023-08-09
>
> Dear Reviewer,
>
> We thank you for your feedback and are happy that you enjoyed reading our paper.
>
> ### Probability Mass Around the Location Parameter vs. Concentration
>
> While the condition that some mass is present around the location parameter, e.g., that $\mathbb{P}(\lvert \eta \rvert \leq \rho) \geq 1/100$, may suggest that some  concentration phenomena are occurring, we would like to remark that in our setting that need not be the case. In particular, the kind of concentration results needed to apply known filtering results from the bounded moment setting, *cannot* be deduced from our assumptions. As a consequence, we need new algorithmic ideas (see also the next section).
>
> As an example: In the bounded-moment setting, it is necessary the the empirical moments of the uncorrupted samples concentrate around their empirical counterpart. In our setting, such concentration may not even occur for the location (let alone for larger powers): Consider the standard Cauchy distribution. This satisfies our definition with $\rho = O(1)$, but averaging $n$ such independent variables, again results in a standard Cauchy distribution. Hence, it is unclear how to use the filtering approach for the bounded moment setting to obtain any non-trivial guarantees. We show that by carefully integrating the Huber loss into this approach, non-trivial and in some cases nearly optimal guarantees can be obtained.
>
>
>
> ### Novel Filtering and Identifiability Using the Huber loss
>
> As you correctly pointed out, the filtering paradigm is not new and has previously been used, e.g., for robust mean estimation under bounded moment assumptions. The same holds for the proof of identifiability based on moment assumptions (further, based on the square loss). Our work builds on these ideas and shows how to extend them to the setting where no moments exist, but a symmetry condition holds. In particular, we would like to emphasize, that to the best of our knowledge, the combination of these ideas with loss functions different from quadratic loss is novel to our submission. As explained in the previous section, these new ideas are necessary to achieve non-trivial guarantees under our assumptions. Using this combination, we can (among others) achieve the following:
> - Nearly achieve the same guarantees as for the standard Gaussian distribution for two natural symmetric generalizations of the standard Gaussian: Product and spherically symmetric distributions.
> - For elliptical distributions, achieve the guarantees as for certifiably sub-Gaussian distributions  (under a mild assumption on the effective rank).
> - Do this in the absence of any assumptions related to sum-of-squares on the underlying distribution.
>
> In particular, previously the filtering paradigm, as well as other algorithms, failed to achieve error better than $o(\sqrt{\varepsilon})$ without assuming the distribution has *certifiably* bounded moments in sum-of-squares (the only exceptions are the Gaussian distribution and distributions with known covariance matrix).
>
> ### Correlation and Location
>
> Regarding your first question: In the paper we mentioned that the class of elliptical distributions allows for more complex "correlation structures", which are useful in modelling, e.g., financial data. We used this term informally, to mean that the coordinates of the vectors might have complex dependencies, indicated by the scatter matrix. Note, that the covariance does not need to exist for this, nor do any other moments. We will rephrase, do make this more clear.
> Similarly, we tried our best to distinguish between mean and location (cf. Theorem 1.4 and 1.5) but will thoroughly proof-read our submission again.
>
>
> ### Other Comments
>
> We would also like to address your other comments/questions:
>
> Yes, Definition 1.2. allows for distributions with dependent coordinates, one example (mentioned in the paper) being spherically symmetric distribution. We will be happy to highlight this more.
>
> Regarding the running time and practicality: As noted in lines 146-148, we do expect our algorithm to be practical. Since it only uses one-dimensional smooth convex minimization for $O(nd)$ times and top-eigenvector computations $O(n)$ times.
>
> We remark that for the filtering algorithm under bounded moment assumptions, there are versions achieving nearly linear running time for which the code is also available [DHL19]. These works use additional ideas beyond the standard filtering idea. We could imagine that similar techniques might work in our setting.
>
>
> ### References
>
> [DHL19]: Yihe Dong, Samuel Hopkins, Jerry Li, "Quantum Entropy Scoring for Fast Robust Mean Estimation and Improved Outlier Detection"

---

> > ### Comment · Reviewer_QF3Z · 2023-08-11
> >
> > Many thanks for the detailed explanations. The score remains unchanged.

---

### Official Review · Reviewer_371F · 2023-07-03

**Soundness:** 4 excellent
**Presentation:** 3 good
**Contribution:** 3 good
**Rating:** 7
**Confidence:** 3

**Summary:**

This paper studies the robust mean estimation problem without any moment assumptions. Instead, they consider a class of symmetric distributions, that is, semi-product and elliptical distributions. They develop a method based on Huber loss and the classic filtering technique that can achieve the same error rate as if the underlining distributions are sub-Gaussian. Their sample complexities are nearly optimal (with additional log factors).

**Strengths:**

- The idea to use Huber loss and develop a similar result for the classic filtering method is very smart. I think this idea can be generalized to other settings and can be of independent interest.

- The paper is well-structured. The techniques part clearly provides the motivation and is very readable.

**Weaknesses:**

- The title is a little bit overclaimed. This paper studies the robust mean estimation problem without moments, but with the constraint that the distribution should be symmetric. I think it would be better to reflect this constraint in the title.

- The notation style is not consistent. Based on my understanding, the authors use bold font for random variables. However, in some cases, like line 95, some symbols are not in bold font.

- A quick question: I can understand that Huber loss has many fantastic properties. Compared with l2-loss, it is more robust against outliers; compared with l1-loss, it is differentiable everywhere. However, l1-loss is only not differentiable at 0. Hence, my question is can we replace the Huber loss with l1-loss? If not, can you briefly mention what is the difficulty?

**Questions:**

Please see the weakness part.

**Limitations:**

Please see the weakness part.

---

> ### Author Rebuttal · Authors · 2023-08-09
>
> Dear Reviewer,
>
> Thank you for your feedback. Regarding the title of our submission: Reflecting that we exploit symmetry already in the title is a good idea. For example, a candidate for the title could be "Robust Mean Estimation Without Moments for Symmetric Distributions".
>
> ### Notation
>
> We will also make sure to check that our notation is consistent, thank you for point this out. We would like to remark that we use regular (non-bold) font for random variables, some of which have been adversarially corrupted. E.g., in line 95, in Definition 1.1. $Z_1, \ldots, Z_n$, refer to an adversarial corruption of some iid (purely random) sample $\boldsymbol X_1, \ldots, \boldsymbol X_n$. We find this distinction between adversarially corrupted and purely random (uncorrupted) samples particularly useful in our proofs of identifiability. We do acknowledge that  we did not specify this  and will add clarification to the paper.
>
> ### $\ell_1$- instead of Huber loss
>
> An approach based on the $\ell_1$-loss should also work, but it requires slightly different assumptions than ours. Concretely, in the semi-product case, one needs to assume that the density is $\Omega(1/\rho)$ at the location. For product distributions we can achieve this by adding Gaussian noise of standard deviation $\Theta(\rho)$ to each entry. For elliptical distributions similar density assumption is also required, however, the approach with adding Gaussian noise to each entry does not work: The resulting distribution might not be elliptical anymore.
>
> To illustrate why the condition on the density is necessary, consider the one-dimensional mean estimation problem for the distribution that is uniform over the (discrete) set $\{\mu-1, \mu + 1\}$. Every $x\in [\mu -1,\mu + 1]$ minimizes the $\ell_1$-loss, to find the correct value we need an "averaging". The Huber loss does this automatically. If we have $\Omega(1)$ density at $\mu$, then the $\ell_1$ loss minimizer is close to $\mu$.

---

> > ### Comment · Reviewer_371F · 2023-08-19
> > **Thanks for your reply**
> >
> > Dear authors,
> >
> > Thanks for your detailed response, which addresses all of my concerns. I especially thank your explanation of l1-loss. I will increase my score from 6 to 7.

---

> > > ### Author Response · Authors · 2023-08-20
> > >
> > > Dear Reviewer,
> > >
> > > We are happy that our response addressed all of your concerns and thank you for raising your score.

---

### Official Review · Reviewer_6FtY · 2023-07-11

**Soundness:** 4 excellent
**Presentation:** 2 fair
**Contribution:** 3 good
**Rating:** 7
**Confidence:** 4

**Summary:**

This work studies robust mean estimation for symmetric distributions. There may possibly no moments (not even the first moment, hence the term location instead of mean), so previous efficient algorithms that rely on strong distributional assumptions may not work on the heavy-tailed setting.

The main result is that similar statistical guarantees as the Gaussian setting can be obtained for $\rho$-semi-product distributions, where a semi-product distribution is a set of distributions that covers slightly more than that of product distributions. For instance, semi-product distributions include elliptical distributions, which is not a product distribution. For the case of known covariance matrix (or scatter matrix for elliptical), the guarantee obtained nearly matches best-known guarantees for Gaussians. For the unknown case, they obtain an error guarantee of $O(\epsilon^{1-\frac{1}{2k}})$ using $\tilde{O}(d^k)$ samples.

The key approach is to generalize the filtering idea for robust mean estimation with a Huber loss instead of a quadratic loss in order to be able to handle distributions without second moments. Once the Huber loss is properly incorporated into the filtering technique, the necessary certificate for filtering can be obtained algorithmically without using sum-of-squares proofs for symmetric distributions.

**Strengths:**

- This work studies a setting which (partially*) generalizes previous works to handle heavy-tailed distributions in which first or second moments may not exist. Here, they incorporate Huber loss into the common filtering technique that has been used in much of the recent robust learning literature. This in itself is novel. Furthermore, by exploiting symmetry, the filtering technique is made more algorithmically feasible without relying SoS approaches.
- Through studying heavy-tailed symmetric distributions, such as elliptical distributions, in the case of unknown covariance, the paper obtains error bounds of $o(\sqrt{\epsilon})$ that were not known for general subgaussian distributions.
- The main result obtains strong guarantees that nearly match that of Gaussians.

*refer to Weakness 1

**Weaknesses:**

1. Only a minor weakness: though the abstact motivates the work by stating that previous efficient estimation algorithms assume strong distributional conditions, the symmetry distributional assumption seems also possibly strong. While it is able to incorporate heavy-tail distributions, it also limits itself in generality (as ever-slightly altered Gaussians are not symmetric but have strong concentrations to exploit).
2. While I enjoyed the content of the paper, he main body seems abruptly cut off without a conclusion or final algorithmic overview. The presesntation would be greatly improved with a final algorithmic description along with a retrospective conclusion.

**Questions:**

- For the main theorems, there is an upper bound on $\epsilon$. What is the breakdown point here? If C < 1/2, what is the reason the proposed approach breaks down?
- In Theorem 1.6, the intuitive definition of $k$ is unclear. Some exposition on $k$ would be helpful.

**Limitations:**

No limtation addressed.

---

> ### Author Rebuttal · Authors · 2023-08-09
>
> Dear Reviewer,
>
> We thank you for your feedback. We are happy that you enjoyed reading the content of our paper. We agree that including a succinct description of our algorithm, e.g., as an algorithms box, as well as a conclusion to the end of the paper would improve the presentation. We will add this.
>
> ### Breakdown Point
>
> The breakdown point of our algorithm depends on how much probability mass is located near the location parameter. To illustrated this, let us consider semi-product distributions (Definition 1.2). We claim that the fraction of corruptions cannot exceed $1/100$. Specifically, consider the mixture distribution $(1-\tfrac 1 {100}) N(\mu,\sigma^2) + \tfrac 1 {100} N(\mu,1)$, for $\sigma^2$ very large (like $2^n$ or larger). Note that this distribution satisfies the assumptions of Definition 1.2 with $\rho = \Theta(1)$. Then by replacing all samples from $N(\mu,1)$ with new independent samples from $N(\mu,\sigma^2)$, the input distribution becomes $N(\mu,\sigma^2)$. In this case, even the information-theoretic error scales with $\sigma$ and hence, can be unbounded.
>
>
>
>
> We remark that in the appendix, we consider a more fine-grained version of of Definition 1.2 (Definition A.1). In particular, it states that a distribution is $(\alpha, \rho)$-semi-product if, Definition 1.2 holds, but with the second item replaced by $\mathbb{P}(\lvert \boldsymbol \eta_j\rvert \leq \rho) \geq \alpha$. In this case, the above discussion shows that the fraction of corruptions cannot exceed $\alpha$. For example, in this setting, the analysis of our polynomial time algorithm (Theorem A.1, generalizing Theorem 1.4) requires $\varepsilon \leq \alpha^3 / C$ for some large enough constant $C$. We did not attempt to optimize this dependence on $\alpha$.
>
> ### On the Role of $k$ in Theorem 1.6
>
> $k$ can be seen as a parameter which allows a tradeoff between (a) number and samples
> and running time and (b) accuracy guarantees. It is analogous to the number of moments used in the bounded-moment setting, and these guarantees are similar to the guarantees of the SoS-based filtering algorithm for distributions with certifiably bounded $2k$-th moment. We will clarify it in the final version of the paper.

---

> > ### Comment · Reviewer_6FtY · 2023-08-14
> > **Reply to Authors**
> >
> > My questions have been answered. Thank you for the response.

---

### Decision · Program_Chairs · 2023-09-21

**Decision:**

Accept (poster)

**Comment:**

This paper advances the state-of-the-art for the fundamental problem of robust mean/location estimation in high dimensions. The authors provide efficient algorithms with dimension-independent error guarantees (in some cases close to best possible) for a range of distribution families without moment assumptions but with additional structural properties (namely, some kind of symmetry). While the algorithmic component heavily draws from prior work in algorithmic robust statistics, this is a solid contribution that is appropriate for publication at NeurIPS.